# `ShaplEIG`: Bayesian Experimental Design for Shapley Value Estimation

David Rundel [1 2]  Fabian Fumagalli [1 2]  Maximilian Muschalik [3 2]  Bernd Bischl [1 2]  Matthias Feurer [4 5]

## Abstract

Shapley values are a principled attribution measure widely used in interpretable machine learning, but their exact computation scales exponentially with the number of players, motivating a wide range of approximation methods based on value function evaluations of sampled coalitions. This raises the question of whether approximation accuracy can be improved by *adaptively* selecting coalitions for evaluation based on previous evaluations. This is particularly relevant in settings where the value function is *costly* and the number of evaluations is severely limited, such as retraining-based feature importance, data valuation, and hyperparameter importance.

For this purpose, we propose `ShaplEIG`, a Bayesian experimental design approach that approximates the expensive value function using a Gaussian process surrogate and adaptively selects coalitions based on their expected information gain about the Shapley values. By the linearity of the Shapley values in the value function, we show that the expected information gain is available in *closed form*. Furthermore, we propose an *efficient* computation scheme that reduces the complexity from exponential to polynomial in the number of players via elementary symmetric polynomials.

In extensive experiments across diverse costly applications, our method consistently improves sample efficiency in the low-budget regime over state-of-the-art baselines.

[1]Department of Statistics, LMU Munich, Munich, Germany [2]Munich Center for Machine Learning, Munich, Germany [3]Institute for Informatics, LMU Munich, Munich, Germany [4]Department of Computer Science, TU Dortmund University, Dortmund, Germany [5]Lamarr Institute for Machine Learning and Artificial Intelligence, Dortmund, Germany. Correspondence to: David Rundel <david.rundel@stat.uni-muenchen.de>, Matthias Feurer <matthias.feurer@tu-dortmund.de>.

*Proceedings of the 43rd International Conference on Machine Learning*, Seoul, South Korea. PMLR 306, 2026. Copyright 2026 by the author(s).

## 1. Introduction

With its origin in cooperative game theory, the Shapley value (SV; Shapley et al., 1953) has emerged as a central tool in explainable AI for axiomatic *attribution* values (Rozemberczki et al., 2022). The exact computation of SVs can be computationally demanding for two main reasons: First, the number of coalitions grows exponentially with the number of players $p$; and second, the cost of evaluating the value function for each of those coalitions can be high, ranging from single model predictions (Lundberg & Lee, 2017), through the computation of conditional expectations (Frye et al., 2021), to full model retraining (Ghorbani & Zou, 2019; Tay et al., 2022), or even complete hyperparameter optimization runs (Wever et al., 2026). *Costly* value functions are also common in applications of SVs outside AI, such as in the field of expensive computer experiments and global sensitivity analysis, where, alongside Sobol indices, SVs provide a variance-based measure of the contribution of inputs to the output of a complex system within a functional ANOVA (fANOVA) framework (Owen, 2014; Benoumechiara & Elie-Dit-Cosaque, 2019).

Even for cheap value functions, the naive exact computation of SVs is typically infeasible, motivating a broad class of stochastic approximation methods (Chen et al., 2023). Early approaches primarily relied on Monte Carlo estimation (Castro et al., 2009; Kwon & Zou, 2022), while more recent methods, such as Kernel SHAP (Lundberg & Lee, 2017), Leverage SHAP (Musco & Witter, 2025) and Regression MSR (Witter et al., 2025), fit surrogate models to the value function and derive SV estimates from these surrogates. However, both types of approaches rely on collections of value function evaluations for coalitions which are typically sampled from fixed, predefined distributions. Especially for expensive value functions, for which the evaluation budget is extremely limited, this naturally raises the question of whether SV approximation quality can be improved by *adaptively selecting* coalitions for evaluation based on previous evaluations (Slack et al., 2021; Burgess & Chapman, 2021).

*Bayesian experimental design* (BED; Lindley, 1956; 1972; Chaloner & Verdinelli, 1995; Sebastiani & Wynn, 2000; Ryan et al., 2016) provides a principled framework for such sequential design of experiments (Jones et al., 1998; Santner et al., 2018). It relies on *surrogate models* and the *expected*

*information gain* (EIG), an information-theoretic criterion, to select candidates for evaluation in order to efficiently infer function properties. Several instances of BED can be found throughout statistics and ML, with the family of entropy search–based acquisition functions (Hennig & Schuler, 2012; Hernández-Lobato et al., 2014) in Bayesian optimization (BO; Garnett, 2023) being a prominent example, where the target property is the function optimum. Despite its theoretical appeal, adoption of BED in ML has been limited, primarily due to the lack of closed-form expressions for the EIG (Foster, 2021; Rainforth et al., 2024). In particular, it has received comparatively little attention in the context of SV estimation and interpretable ML in general.

*Our contributions* are as follows:

- We introduce `ShaplEIG`, a novel, BED-based method that approximates expensive value functions in SV estimation via a Gaussian process (GP) surrogate with a Hamming kernel, adaptively selects coalitions for evaluation based on the EIG, and yields a consistent SV estimator.

- We show that the EIG of a candidate coalition about the SVs admits a closed-form expression. This is achieved by exploiting the linearity of SVs in the value function and by framing sequential coalition selection as a Bayesian linear inverse problem under a GP surrogate.

- We propose an efficient computation scheme for the EIG that exploits the multiplicative structure of the Hamming kernel via a correspondence to elementary symmetric polynomials (ESPs), reducing the computational cost from exponential to polynomial in the number of players.

- Through extensive experiments on several *costly*, *small-to moderately large* games ($8 \leq p \leq 101$), we demonstrate improved estimation accuracy and sample efficiency in the low-budget regime compared with state-of-the-art baselines. These games include retraining-based tasks - *feature importance* for TabPFN, *data valuation* and *hyperparameter importance* of learning algorithms such as XGBoost - as well as *local explanations* for vision models.

## 2. Methodological Background

### 2.1. Shapley Values

For a set $P := \{1, \ldots, p\}$ of $p$ *players*, such as features or data points, the SV aggregates the *value function* $\nu : 2^P \to \mathbb{R}$ of all $2^p$ coalitions into a single attribution score for each player $i \in P$ as

$$\phi_i(\nu) := \sum_{S \subseteq P \setminus \{i\}} \frac{1}{p \cdot \binom{p-1}{|S|}} \Big( \nu(S \cup \{i\}) - \nu(S) \Big). \quad (1)$$

This is the player's marginal contribution to the game, averaged over all possible coalitions, with weights proportional to coalition sizes. Direct evaluation of Equation 1 is only computationally feasible for small to moderate values of $p$, as the number of coalitions grows exponentially in $p$, and when evaluations of the value function $\nu$ are inexpensive.

**Approximation Methods.** A straightforward approach to approximating Equation 1, or reformulations thereof, is computing a Monte Carlo estimate based on sampled coalitions $\mathcal{S} \subset 2^P$. These estimators are generally unbiased, and depending on the reformulation and coalition sampling procedure used, differ in estimator variance and capability to simultaneously use value function evaluations for the estimation of all SVs $\phi := (\phi_1, \ldots, \phi_p)^\top \in \mathbb{R}^p$ across players. Notable variants include permutation sampling (Castro et al., 2009), maximum sample reuse (MSR; Wang & Jia, 2023), and SVARM (Kolpaczki et al., 2024). A number of variance-reduction strategies have been proposed within this Monte Carlo framework, including *stratified*, *antithetic* or *paired sampling* of complementary coalitions (Mitchell et al., 2022; Covert & Lee, 2021), and biased sampling toward coalitions near the extremes of subset cardinality, referred to as the *border trick* (Fumagalli et al., 2023). While these techniques can substantially improve efficiency, they still rely on fixed, non-adaptive sampling distributions.

In contrast, surrogate-based methods reduce SV estimation to a supervised regression problem by fitting a surrogate model $\hat{\nu} : 2^P \to \mathbb{R}$ to the value function using coalitions $\mathcal{S}$ sampled according to any of the aforementioned strategies. The SVs are then approximated as $\phi(\nu) \approx \phi(\hat{\nu})$. Kernel SHAP, Leverage SHAP, and PolySHAP (Fumagalli et al., 2026) are special cases of this framework, using a linear surrogate model and a tailored regression objective. More recently, Regression MSR employs tree-based surrogates, such as XGBoost (Chen & Guestrin, 2016), in combination with TreeSHAP (Lundberg et al., 2020), and demonstrates state-of-the-art performance. A key requirement for the surrogate model is that SVs can be extracted *efficiently*, even for large player sets, as is the case for linear models where they correspond directly to the model's coefficients. Furthermore, the surrogate should yield a *consistent* estimator, i.e., the estimates converge to the true SVs once all $2^p$ coalitions are evaluated. While Kernel SHAP and Leverage SHAP employ regression objectives that guarantee consistency, tree-based surrogates do not yield consistent estimates by default and therefore require an additional "adjustment" step via MSR on the residuals $\nu(S) - \hat{\nu}(S)$.

### 2.2. Bayesian Experimental Design

BED considers an expensive-to-evaluate *black-box function* $f : \mathcal{X} \to \mathbb{R}$, defined over a bounded input space $\mathcal{X} \subseteq \mathbb{R}^d$, and leverages probabilistic surrogate models together with

information-theoretic criteria to efficiently infer a *function property* of interest $\boldsymbol{\varphi}(f) \in \mathbb{R}^m$ under a limited budget of function evaluations. We assume that a function evaluation corresponds to a noisy experiment $y(\mathbf{x}) := f(\mathbf{x}) + \epsilon$ at a design $\mathbf{x} \in \mathcal{X}$, with homoscedastic, additive Gaussian noise $\epsilon \sim \mathcal{N}(0, \sigma_\epsilon^2)$ that is i.i.d. across evaluations.

In this work, we focus on the sequential setting with the greedy Bayesian adaptive design (BAD; Cheng & Shen, 2005) algorithm, which at each iteration $t > T_0$ selects the next design $\mathbf{x}$ by maximizing the EIG:

$$
\begin{aligned}
\mathrm{EIG}^{(t)}_{\boldsymbol{\varphi}(f)}(\mathbf{x}) :=\ & I\big(\boldsymbol{\varphi}(f); y(\mathbf{x}) \mid \mathcal{D}_t\big) \\
=\ & H\big(\boldsymbol{\varphi}(f) \mid \mathcal{D}_t\big) \\
& - \mathbb{E}_{p(y(\mathbf{x})|\mathcal{D}_t)}\big[H\big(\boldsymbol{\varphi}(f) \mid y(\mathbf{x}), \mathcal{D}_t\big)\big].
\end{aligned} \tag{2}
$$

Here, $\mathcal{D}_t := \{(\mathbf{x}^{(i)}, y(\mathbf{x}^{(i)}))\}_{i=1}^{t-1}$ denotes the dataset of all previous evaluations, including an initial design of size $T_0$. The EIG expresses the mutual information $I$ between the function property $\boldsymbol{\varphi}(f)$ and the function evaluation $y(\mathbf{x})$, conditioned on $\mathcal{D}_t$. Intuitively, this criterion selects the design whose associated function evaluation is expected to maximize the reduction in uncertainty about the property. We measure uncertainty using the differential entropy $H$ based on the posterior distribution of the surrogate model given previously observed data $\mathcal{D}_t$ (Rainforth et al., 2024; Huan et al., 2024).

Bayesian surrogate models are used to regress the latent function $f$ on $\mathbf{x}$. A common choice is to place a GP prior on the latent function $f$ (Krause et al., 2008; Houlsby et al., 2011; Hennig & Schuler, 2012; Neiswanger et al., 2021). GPs are particularly suitable in this context, as they allow *exact* Bayesian inference in closed form and provide *well-calibrated* uncertainty estimates even in data-scarce regimes. We briefly introduce GPs in Appendix A.1.

**EIG Estimation.** The second term of the EIG in Equation 2 is itself an expectation of an entropy, yielding a nested expectation structure over potentially intractable posteriors. Although the posterior predictive distribution (PPD) $p(y(\mathbf{x}) \mid \mathcal{D}_t)$ is explicit for many surrogate types such as GPs, the conditional property posterior $p(\boldsymbol{\varphi}(f) \mid y(\mathbf{x}), \mathcal{D}_t)$ is generally not. It can be approximated using standard methods if $p(y(\mathbf{x}) \mid \boldsymbol{\varphi}(f), \mathcal{D}_t)$ is explicit; otherwise, likelihood-free approaches (Csilléry et al., 2010) are often employed. In both cases, however, the normalized posterior density of the property is required for entropy estimation (Foster, 2021; Rainforth et al., 2024). Finally, because this entropy term is nested within the expectation over the PPD, which may itself be intractable, nested estimation strategies are often needed.

As a consequence, estimating the EIG poses a fundamental challenge in BED, both in terms of the accuracy of the resulting estimates and the associated computational cost. While a large body of research has focused on developing

specialized techniques to enhance its practical applicability (Houlsby et al., 2011; Heinrich et al., 2020; Goda et al., 2020), there also exist special cases where the associated computations are tractable (Attia et al., 2018).

**Bayesian Linear Inverse Problems.** A Bayesian linear inverse problem consists of an *unknown parameter* $\boldsymbol{\theta} \in \mathbb{R}^s$ that is to be inferred from *experimental data* $y(\mathbf{x}) \in \mathbb{R}$.[1] This data is assumed to be generated according to a linear *observation model*, $y(\mathbf{x}) = F(\mathbf{x})^\top \boldsymbol{\theta} + \epsilon$, where $F(\mathbf{x}) \in \mathbb{R}^s$ denotes the *parameter-to-observable mapping* depending on the experimental design $\mathbf{x}$, and $\epsilon \sim \mathcal{N}(0, \sigma_\epsilon^2)$ is additive Gaussian noise. Assuming a Gaussian prior on the parameter, i.e., $\boldsymbol{\theta} \sim \mathcal{N}(\boldsymbol{\mu}_\theta, \boldsymbol{\Sigma}_\theta)$, the posterior distribution $\boldsymbol{\theta} \mid y(\mathbf{x})$ is again Gaussian, with covariance given by

$$
\boldsymbol{\Sigma}_{(\boldsymbol{\theta}|y(\mathbf{x}))} = \big(\boldsymbol{\Sigma}_\theta^{-1} + F(\mathbf{x})\sigma_\epsilon^{-2}F(\mathbf{x})^\top\big)^{-1}.
$$

In many applications, the primary quantity of interest is not the parameter itself, but an *end-goal* $\boldsymbol{\varphi}(\boldsymbol{\theta}) \in \mathbb{R}^m$ that depends on $\boldsymbol{\theta}$. For a linear *goal operator* $\mathbf{A} \in \mathbb{R}^{m \times s}$ with full row rank, the end-goal is a linear transformation of the parameter, i.e., $\boldsymbol{\varphi}(\boldsymbol{\theta}) = \mathbf{A}\boldsymbol{\theta}$. Applying BED in this setting to select experimental data with the goal of reducing uncertainty about $\boldsymbol{\varphi}(\boldsymbol{\theta})$ corresponds to a specific instance of goal-oriented optimal design of experiments (GOODE; Lieberman & Willcox, 2013), which is closely related to Bayesian $D_A$-optimality. Notably, the EIG of a design $\mathbf{x}$ with respect to the linear end-goal can be expressed in closed form as

$$
\begin{aligned}
\mathrm{EIG}_{\boldsymbol{\varphi}(\boldsymbol{\theta})}(\mathbf{x}) =\ & I\big(\boldsymbol{\varphi}(\boldsymbol{\theta}); y(\mathbf{x})\big) \\
=\ & -\tfrac{1}{2} \log \det \big(\mathbf{A}\boldsymbol{\Sigma}_{(\boldsymbol{\theta}|y(\mathbf{x}))}\mathbf{A}^\top\big) + C,
\end{aligned} \tag{3}
$$

where $C$ is constant with respect to $\mathbf{x}$ (but may depend on $\mathbf{A}$ and $\boldsymbol{\Sigma}_\theta$; see Attia et al. (2018) for a proof). Moreover, the EIG depends only on the experimental design $\mathbf{x}$ and not on the associated outcome $y(\mathbf{x})$ (Bui-Thanh et al., 2013; Alexanderian et al., 2016; Spantini et al., 2017; Zhong et al., 2026).

## 3. BED for SV Estimation

Prior work on approximating SVs typically relies on evaluated coalitions drawn from a fixed, predefined distribution. However, when value function evaluations are *costly* and only a limited number are possible, sampling without leveraging information from previous evaluations may result in wasted resources. Instead, we propose an *adaptive* approach, which we call `ShaplEIG`, that iteratively expands the collection *optimally* based on previously observed evaluations. Specifically, we propose a greedy BAD approach (see Algorithm 1) that iteratively trains a GP surrogate for the value

---

[1] Here, we restrict our attention to the case of a single experimental observation.

---

**Algorithm 1** `ShaplEIG` algorithm

---

**Require:** GP prior $\nu \sim \mathcal{GP}(m, k_\xi)$, initial design and candidate set $\mathcal{C}_0, \mathcal{C} \subseteq \{1, \ldots, 2^p\}$ with $|\mathcal{C}_0| = T_0$ and $\mathcal{C}_0 \cap \mathcal{C} = \emptyset$

1: Initialize dataset: $\mathcal{D}_{T_0+1} := \{(\mathbf{z}^{(i)}, \nu(\mathbf{z}^{(i)}))\}_{i \in \mathcal{C}_0}$
2: **for** $t = T_0 + 1$ to $T$ **do**
3:   Optimize EIG: $g(t) := \arg\max_{i \in \mathcal{C}} \operatorname{EIG}_{\phi}^{(t)}(\mathbf{z}^{(i)})$
4:   Update dataset: $\mathcal{D}_{t+1} := \mathcal{D}_t \cup \{(\mathbf{z}^{(g(t))}, \nu(\mathbf{z}^{(g(t))}))\}$
5:   Update candidate set: $\mathcal{C} := \mathcal{C} \setminus \{g(t)\}$
6:   Refit hyperparameters: $\xi := \arg\max_\xi p(\xi \mid \mathcal{D}_{t+1})$
7: **end for**
8: **return** Consistent SV estimates: $\hat{\phi} := \boldsymbol{\mu}_{\phi|\mathcal{D}_{T+1}}$

---

function (Section 3.1) which yields a consistent estimator for the SVs (Section 3.2). It operates on an initial design of coalition evaluations and at each iteration (1) optimizes the EIG about the SVs over candidate coalitions, leveraging a closed-form expression for the EIG (Section 3.3) and an efficient computation scheme (Section 3.4), (2) evaluates the newly selected coalition, and (3) updates the GP surrogate using the newly acquired data, thereby retraining the GP hyperparameters to ensure adaptivity of the subsequent experimental design (Section 3.3).

By slight abuse of notation, we define $\nu$ equivalently on binary indicator vectors $\mathbf{z} \in \{0, 1\}^p$ via their bijective correspondence with coalitions $S \subseteq P$. Furthermore, we collect all $2^p$ such vectors as rows of a matrix $\mathbf{Z} \in \{0, 1\}^{2^p \times p}$ and denote its $i$-th row as $\mathbf{z}^{(i)} := \mathbf{Z}_i$. The previously observed coalition–evaluation pairs at iteration $t$ are gathered in $\mathcal{D}_t := \{(\mathbf{z}^{(g(i))}, \nu(\mathbf{z}^{(g(i))}))\}_{i=1}^{t-1}$, with $g : \{1, \ldots, t-1\} \to \{1, \ldots, 2^p\}$ mapping each iteration index to the row index in $\mathbf{Z}$ of the coalition selected at that iteration.

### 3.1. GP Surrogate with Hamming Kernel

We model the value function using a GP surrogate (see Appendix A.1 for an introduction). While the use of a surrogate is related to existing, popular methods from SV estimation, we depart from these approaches in our choice of surrogate, as our use is primarily motivated by adaptive coalition selection based on the EIG. In this context, we require a nonlinear, fully probabilistic model capable of handling low-data regimes. GPs are better aligned with this goal and yield further advantages that will become apparent in the following sections.

As a covariance function, we employ the Hamming distance kernel (Platt et al., 2001; Qian et al., 2008; Hutter, 2009; see Appendix A.1), which quantifies similarities between coalitions in the binary input space via weighted Hamming distances. The associated weights $\xi \in \mathbb{R}^p$ are treated as learnable hyperparameters. This kernel is a common choice for categorical input spaces and is especially

advantageous in our context, as it enables efficient EIG computation schemes (Section 3.4).

At each iteration $t$ of the sequential procedure and for fixed kernel hyperparameters $\xi$, the GP surrogate coupled with the available data $\mathcal{D}_t$ induces a closed-form multivariate normal distribution (MVN) over the value function evaluated across all coalitions $\boldsymbol{\nu} := \nu(\mathbf{Z}) \in \mathbb{R}^{2^p}$,[2] i.e.,

$$\boldsymbol{\nu} \mid \mathcal{D}_t, \xi \sim \mathcal{N}_{2^p}(\boldsymbol{\mu}_{\boldsymbol{\nu}|\mathcal{D}_t, \xi}, \boldsymbol{\Sigma}_{\boldsymbol{\nu}|\mathcal{D}_t \cdot \xi}).$$

### 3.2. Consistent SV Estimation

The SVs across players, $\phi \in \mathbb{R}^p$, depend on the value function $\nu$ only through a linear transformation of $\boldsymbol{\nu}$, as implied by the linearity axiom of SVs (Shapley et al., 1953), i.e.,

$$\phi(\nu) = \mathbf{A}\boldsymbol{\nu} := \left(\frac{\mathbf{1}_S}{p \cdot \binom{p-1}{|S|-1}} - \frac{\mathbf{1} - \mathbf{1}_S}{p \cdot \binom{p-1}{|S|}}\right)_{S \subseteq P} \boldsymbol{\nu},$$

where $\mathbf{A} \in \mathbb{R}^{p \times 2^p}$ and $\mathbf{1}_S$ denotes the indicator vector of coalition $S$ over the player set $P$.[3] As a consequence, at iteration $t$ the posterior distribution over the SVs under the GP surrogate is available in closed form (Chau et al., 2023) and given by:

$$\phi(\nu) \mid \mathcal{D}_t \sim \mathcal{N}_p(\mathbf{A}\boldsymbol{\mu}_{\boldsymbol{\nu}|\mathcal{D}_t}, \mathbf{A}\boldsymbol{\Sigma}_{\boldsymbol{\nu}|\mathcal{D}_t}\mathbf{A}^\top)$$
$$= \mathcal{N}_p(\boldsymbol{\mu}_{\phi|\mathcal{D}_t}, \boldsymbol{\Sigma}_{\phi|\mathcal{D}_t}).$$

Based on this, we extract SV estimates from the surrogate via $\hat{\phi} := \phi(\hat{\nu}) = \boldsymbol{\mu}_{\phi|\mathcal{D}_t}$. Importantly, for noiseless GPs (see Appendix A.1), this estimator is consistent by construction: when the surrogate is trained on all $2^p$ coalitions, i.e., under $\mathcal{D}_{2^p+1}$, we recover the exact SVs, $\boldsymbol{\mu}_{\phi|\mathcal{D}_{2^p+1}} = \phi(\nu)$. This follows directly from the interpolation property (Stein, 1999; Williams & Rasmussen, 2006), namely $\boldsymbol{\mu}_{\nu(\mathbf{z})|\mathcal{D}_{2^p+1}} = \nu(\mathbf{z})$ for all $\mathbf{z} \in \{0, 1\}^p$. The consistency of our estimator stands in contrast to recently proposed tree-based surrogate approaches such as Regression MSR, which require additional adjustment steps. For a discussion of noisy GP alternatives, estimator bias, and debiasing schemes, see Appendix B.1.

### 3.3. Closed-form EIG

As the GP surrogate is iteratively trained on value function observations, each potential subsequent evaluation $\nu'(\mathbf{z}^{(i)}) \in \mathbb{R}$ for a candidate coalition $\mathbf{z}^{(i)}$ at iteration $t$ partially reveals information about $\boldsymbol{\nu}$ and induces an updated posterior distribution, i.e., $\boldsymbol{\nu} \mid \nu'(\mathbf{z}^{(i)}), \mathcal{D}_t$. Note that

---

[2]For notational convenience, we omit explicit conditioning on $\xi$ in the following, and denote $\boldsymbol{\mu}_{\boldsymbol{\nu}|\mathcal{D}_t} := \boldsymbol{\mu}_{\boldsymbol{\nu}|\mathcal{D}_t, \xi}$ and $\boldsymbol{\Sigma}_{\boldsymbol{\nu}|\mathcal{D}_t \cdot \xi} := \boldsymbol{\Sigma}_{\boldsymbol{\nu}|\mathcal{D}_t}$.

[3]We adopt the convention that $\binom{n}{k} = 0$ for $k > n$ and $k < 0$ together with $0/0 := 0$.

each such evaluation constitutes experimental data generated by a linear observation model, i.e., $\nu'(\mathbf{z}^{(i)}) = \mathbf{e}_i^\top \boldsymbol{\nu} + \epsilon$. Here, $\mathbf{e}_i \in \mathbb{R}^{2^p}$ denotes the $i$-th standard basis vector, which simply selects the single entry of $\boldsymbol{\nu}$ corresponding to the value of $\mathbf{z}^{(i)}$, while $\epsilon$ is assumed to be zero-mean Gaussian noise with variance fixed to a small constant for numerical stability.

Consequently, adaptively selecting data for surrogate training can be viewed as a Bayesian linear inverse problem with $\boldsymbol{\nu}$ taking the role of the unknown parameter ($\boldsymbol{\theta}$ in Section 2.2) and $\mathbf{e}_i$ acting as the parameter-to-observable mapping ($F(\mathbf{x})$ in Section 2.2). Furthermore, selecting coalitions to efficiently infer the SVs constitutes an instance of GOODE (Lieberman & Willcox, 2013): The SVs $\boldsymbol{\phi}$ are given by a linear transformation of the unknown parameter $\boldsymbol{\nu}$ and can be interpreted as a linear end-goal of the corresponding Bayesian linear inverse problem ($\boldsymbol{\varphi}(f)$ in Section 2.2), with $\mathbf{A}$ acting as the linear goal operator. As a consequence, the EIG of $\mathbf{z}^{(i)}$ about $\boldsymbol{\phi}$ is available in closed form as:

$$\text{EIG}_{\boldsymbol{\phi}}^{(t)}(\mathbf{z}^{(i)}) \propto -\log\det\left(\mathbf{A}\boldsymbol{\Sigma}_{\boldsymbol{\nu}|\nu'(\mathbf{z}^{(i)}),\mathcal{D}_t}\mathbf{A}^\top\right) + C \quad (4)$$

$$= -\log\det\left(\boldsymbol{\Sigma}_{\boldsymbol{\phi}|\nu'(\mathbf{z}^{(i)}),\mathcal{D}_t}\right) + C. \quad (5)$$

This shows that SVs belong to a structural class of function properties for which the EIG admits a closed-form solution under GP surrogates.

**Adaptivity.** We note that the EIG depends only on the GP's posterior covariance, not on its mean. At iteration $t$, this covariance is determined solely by the previously selected coalitions and not directly by the associated evaluations (see Equation 9 in Appendix A.1). However, the GP hyperparameters $\xi$ are learned at each iteration from all data $\mathcal{D}_t$ acquired so far. As a result, previous evaluations have an indirect effect on the EIG, because they affect the kernel hyperparameters, which in turn influence the posterior covariance and thus the acquisition criterion. Without this retraining, the iterative procedure would collapse into a so-called *non-adaptive* experimental design, in which the designs do not depend on previous outcomes and can be fully determined before data collection (Huan et al., 2024). Similarly, when using a linear surrogate model with fixed basis functions and fixed noise variance, as in Kernel SHAP, the uncertainty of the SV estimates depends only on the previously selected coalitions, and not on the associated evaluations (Slack et al., 2021; Huan et al., 2024). Consequently, classical criteria for experimental design - including $D_A$-optimality, which is closely related to the EIG in our setting - yield designs that are not adaptive with respect to previous evaluations. This further motivates our use of a GP surrogate.

**Practical Implications.** Despite the mathematical formulation of the EIG in closed form, it may still be unclear to

readers how this covariance-based criterion affects coalition selection in practice, and when it can be expected to provide benefits over alternative coalition selection strategies. In Appendix B.2, we provide a detailed discussion of this matter together with intuitive examples of games and the behavior of different coalition selection strategies to further illustrate these concepts.

In summary, for each candidate coalition, the EIG considers how an evaluation reduces uncertainty about the value function across all coalitions, and how this propagates to the SVs according to the covariance structure induced by the GP. This stands in contrast to current state-of-the-art approaches for coalition sampling (e.g., leverage score sampling; Musco & Witter, 2025), which sample coalitions from fixed distributions and do not distinguish between different coalitions of the same size. Consequently, for asymmetric games, e.g., with interactions occurring primarily among a subset of relevant players, we expect the EIG to provide substantial benefits over traditional approaches.

### 3.4. Efficient Computation

In the following, we present an efficient computation scheme for the quantities associated with our proposed `ShaplEIG` framework, primarily the EIG. We also analyze the computational complexity of this approach and contrast it with a naive implementation.

In Appendix B.3, we derive that the EIG of $\mathbf{z}^{(i)}$ about $\boldsymbol{\phi}$ can be equivalently expressed as:

$$\text{EIG}_{\boldsymbol{\phi}}^{(t)}(\mathbf{z}^{(i)}) \propto C' + \log\left[\mathbf{e}_i^\top\left(\boldsymbol{\Sigma}_{(\boldsymbol{\nu}|\mathcal{D}_t)} + \sigma_\epsilon^2\mathbf{I}\right)\mathbf{e}_i\right] \quad (6)$$

$$- \log\left[\mathbf{e}_i^\top\left(\boldsymbol{\Sigma}_{(\boldsymbol{\nu}|\mathcal{D}_t)} + \sigma_\epsilon^2\mathbf{I} - \mathbf{Q}\right)\mathbf{e}_i\right].$$

Here, $C'$ is constant, $\mathbf{I} \in \mathbb{R}^{2^p \times 2^p}$ is the identity matrix, and $\mathbf{Q}_{i,i} = \mathbf{e}_i^\top\mathbf{Q}\mathbf{e}_i$, the $i$-th diagonal entry of $\mathbf{Q} \in \mathbb{R}^{2^p \times 2^p}$, is defined as

$$\mathbf{Q}_{i,i} = \left(\mathbf{A}\boldsymbol{\Sigma}_{(\boldsymbol{\nu}|\mathcal{D}_t)}\mathbf{e}_i\right)^\top\left(\mathbf{A}\boldsymbol{\Sigma}_{(\boldsymbol{\nu}|\mathcal{D}_t)}\mathbf{A}^\top\right)^{-1}\left(\mathbf{A}\boldsymbol{\Sigma}_{(\boldsymbol{\nu}|\mathcal{D}_t)}\mathbf{e}_i\right).$$

In particular, $\mathbf{e}_i^\top\boldsymbol{\Sigma}_{(\boldsymbol{\nu}|\mathcal{D}_t)}\mathbf{e}_i = \text{Var}(\nu(\mathbf{z}^{(i)}) \mid \mathcal{D}_t) \in \mathbb{R}$ reduces to the marginal posterior variance of $\nu(\mathbf{z}^{(i)})$, and $\boldsymbol{\Sigma}_{(\boldsymbol{\nu}|\mathcal{D}_t)}\mathbf{e}_i = \text{Cov}(\boldsymbol{\nu}; \nu(\mathbf{z}^{(i)}) \mid \mathcal{D}_t) \in \mathbb{R}^{2^p}$ corresponds to the posterior covariance between $\boldsymbol{\nu}$ and $\nu(\mathbf{z}^{(i)})$. However, $\mathbf{Q}_{i,i}$ depends on the inverse of $\mathbf{A}\boldsymbol{\Sigma}_{(\boldsymbol{\nu}|\mathcal{D}_t)}\mathbf{A}^\top \in \mathbb{R}^{p \times p}$, and therefore on the full posterior covariance matrix across all coalitions, $\boldsymbol{\Sigma}_{(\boldsymbol{\nu}|\mathcal{D}_t)} \in \mathbb{R}^{2^p \times 2^p}$.

**Naive EIG Computation.** In a naive approach to computing the EIG for a specific candidate $\mathbf{z}^{(i)}$, one could first compute the full covariance matrix $\boldsymbol{\Sigma}_{(\boldsymbol{\nu}|\mathcal{D}_t)}$ and then obtain all required terms via projections with $\mathbf{A}$ or slices using $\mathbf{e}_i$. In this case, the overall computational cost is dominated by $\mathcal{O}(4^p \cdot t)$ (see the posterior covariance matrix computation in Appendix A.1). This complexity is exponential in $p$ and therefore prohibitively expensive in many settings.

**Efficient EIG Computation.** However, we now show that the EIG can be computed much more efficiently, reducing the exponential scaling in $p$ to polynomial.

**Theorem 3.1.** *The EIG about the SVs $\phi$ for a candidate coalition $\mathbf{z}^{(i)} \in \{0,1\}^p$ is computable in $\mathcal{O}(p^4 + t^3)$.*

*Proof.* We present the (somewhat lengthy) proof in Appendix B.4. It is based on two further theorems: First, Theorem B.1, which shows how the linear term $\mathbf{A}K_\xi(\mathbf{Z}, \mathbf{z}^{(i)}) \in \mathbb{R}^p$, where $K_\xi$ denotes the kernel matrix (see Appendix A.1), required for the computation of $\mathbf{A}\boldsymbol{\Sigma}_{(\boldsymbol{\nu}|\mathcal{D}_t)}\mathbf{e}_i$, can be computed in $\mathcal{O}(p^2)$. Second, Theorem B.2, which shows how the quadratic term $\mathbf{A}K_\xi(\mathbf{Z}, \mathbf{Z})\mathbf{A}^\top \in \mathbb{R}^{p \times p}$, required for the computation of $\mathbf{A}\boldsymbol{\Sigma}_{(\boldsymbol{\nu}|\mathcal{D}_t)}\mathbf{A}^\top$, can be computed in $\mathcal{O}(p^4)$. Both of these results are achieved by rewriting the terms as weighted sums of kernel evaluations across coalitions, then recognizing that many of these kernel evaluations share the same weights, and that the sums over groups of kernel values with identical weights can be computed more efficiently by identifying them with scaled, uni- or bivariate elementary symmetric polynomials (ESPs; van Es & Helmers, 1988; Macdonald, 1998; Charalambides, 2018), thereby exploiting the multiplicative structure of the Hamming kernel. We then tie all of this together in Theorem B.3 for the complete EIG computation. □

The approach for the linear part is similar to the computation of products between Shapley weights and a kernel matrix proposed by Mohammadi et al. (2025a). However, our overall setup deviates from theirs, which leads our derivation to be based on scaled ESPs rather than the unscaled variant they use. Furthermore, we are not aware of any prior work regarding the quadratic part.

**Vectorized EIG Computation.** Consider the setting in which the EIG is evaluated for a set of candidate coalitions $\mathbf{W} \subseteq \{0,1\}^p$. We show in Appendix B.4.1 that this can be efficiently vectorized across candidates, scaling as $\mathcal{O}(p^4 + t^3 + |\mathbf{W}| \cdot t^2)$. In particular, the first two additive terms, which dominate the computational cost in many settings, are associated with operations that are independent of the specific candidate and thus scale independently of $|\mathbf{W}|$, while all remaining candidate-specific operations can be efficiently vectorized. For games with small $p$, this even enables exhaustive EIG optimization across all candidates.

**Efficient SV Computation.** Like the EIG, the SV estimates $\hat{\boldsymbol{\phi}} = \mathbf{A}\boldsymbol{\mu}_{\boldsymbol{\nu}|\mathcal{D}_t}$ (Section 3.2) can be computed efficiently. This follows immediately by applying Theorem B.1 to the expanded form of $\boldsymbol{\mu}_{\boldsymbol{\nu}|\mathcal{D}_t}$ (see Appendix A.1), resulting in a computational complexity of $\mathcal{O}(t^3)$ for SV estimation. Similarly, the posterior covariance of the SVs, $\boldsymbol{\Sigma}_{\boldsymbol{\phi}|\mathcal{D}_t} = \mathbf{A}\boldsymbol{\Sigma}_{(\boldsymbol{\nu}|\mathcal{D}_t)}\mathbf{A}^\top$, can be computed in $\mathcal{O}(p^4 + t^2 \cdot p)$ by applying Theorem B.2 as in the EIG computation.

## 4. Related Work

Our proposed method is closely related to the pool-based active learning literature (Lewis & Gale, 1994; Settles, 2009; Houlsby et al., 2011; Gal et al., 2017), where the goal is to train a model efficiently in an adaptive manner. In this context, it is closest to transductive, or similarly prediction-oriented, BED variants (Yu et al., 2006; Hübotter et al., 2024), where the quantities of interest are the model predictions at a set of input points (in our setting, these correspond to the GP predictions of the value function across all coalitions). However, our EIG criterion directly targets the SVs, which are a linear transformation of the value function. It therefore differs from commonly used approaches in this literature and yields both information-theoretic and computational advantages. In particular, information-based transductive learning (ITL; MacKay, 1992), i.e., the EIG for the untransformed value function vector $\boldsymbol{\nu}$, would collapse in our setting to the purely exploratory uncertainty sampling (US; Lewis & Catlett, 1994), i.e., selecting the coalition with the highest surrogate uncertainty (Krause et al., 2008), thereby ignoring how an evaluation reduces uncertainty at points beyond the candidate itself. The expected predictive information gain (EPIG; Smith et al., 2023), another popular variant, would incur prohibitively high computational cost when considering all coalitions in the target distribution. Lastly, although our approach can be framed as active testing (Kossen et al., 2021; 2022; see Appendix B.1), it is more general in that the quantity of interest is not restricted to a scalar-valued linear transformation of the function. We refer to Appendix C for further details.

Furthermore, several lines of prior work in SV estimation are related to our approach in that they exploit the predictive uncertainty of surrogates to guide the iterative selection of coalitions, or employ GPs as surrogate models. Slack et al. (2021) proposed BayesSHAP, which uses a Bayesian linear model as a surrogate to obtain point estimates of SVs together with uncertainty estimates. In addition, they proposed selecting queries using US. Mitchell et al. (2022) used GP surrogates with specialized kernels over permutations and Bayesian quadrature (BQ; Larkin, 1972; O'Hagan, 1991; Rasmussen & Ghahramani, 2003) to extract SVs, as well as sequential BQ (Huszár & Duvenaud, 2012) to select permutations for evaluation. Similarly, Nguyen et al. (2025) proposed to actively select coalitions using GP surrogates with kernels defined over data distributions for data valuation. However, in both of the latter two approaches, the selection criteria target uncertainty reduction for the SV of a single player rather than jointly across all players. Moreover, these methods rely on fixed kernels, which leads to non-adaptive experimental designs.

Beyond this, there exist various model-specific methods

for explaining the predictions of GP models on a given observation with respect to input features via SVs (Benoumechiara & Elie-Dit-Cosaque, 2019; Chau et al., 2022; 2023; Mohammadi et al., 2025a;b). In this line of work, several definitions of the value function associated with a feature subset have been considered, such as removal-based formulations that rely on conditional expectations of the model output with respect to a distribution over missing features. In the GP setting, such value functions can be computed efficiently using conditional mean embeddings (Chau et al., 2021a;b). In contrast, we focus on using GPs as surrogates to directly model the relationship between coalitions and an observed value function for arbitrary cooperative games.

## 5. Experiments

### 5.1. Experimental Setup

In the experiments, we run `ShaplEIG` with the greedy BAD algorithm and compare it to popular SV estimation methods, as well as relevant baselines from BED and ablations of our method. In particular, we choose an initial design of size $T_0 = p + 1$ with coalitions drawn according to leverage score sampling (Musco & Witter, 2025). Then, at each iteration, we optimize the EIG either exhaustively or over at most 1024 candidate coalitions, and refit the GP hyperparameters using the newly observed data. We run the procedure for a maximum of 512 evaluations, or until all coalitions have been evaluated. Note that our experiments focus on the low-budget regime, whereas other benchmarks typically consider larger evaluation budgets. We provide further details on the initial design and GP surrogate in Appendices D.1.1 and D.1.2.

**Estimation Methods.** We compare `ShaplEIG` against widely used and recent SV approximation methods. These are Kernel SHAP (Lundberg & Lee, 2017), Leverage SHAP (Musco & Witter, 2025), permutation sampling (Castro et al., 2009), and the state-of-the-art method Regression MSR (Witter et al., 2025) coupled with XGBoost. For all of the above, we apply paired sampling (Covert & Lee, 2021) as an advanced coalition sampling technique. For a fair comparison, at each iteration, these competitor methods are run using the same budget for value function evaluations as `ShaplEIG`.

In addition, we evaluate several ablations of our method. In all of these, the same GP surrogate type as in `ShaplEIG` is used in the same iterative procedure, including identical initial designs and SV extraction from the posterior conditioned on previous evaluations, but the strategies for selecting the coalitions on which the GP is trained differ from our EIG-based criterion. In detail, we compare to a) random

*Table 1.* Overview of explanation tasks and games considered in the experiments.

| Task | Model | Dataset | p | #Reps |
|---|---|---|---|---|
| *Feature Importance (FI)* (Rundel et al., 2024) | TabPFN | Diabetes (Reg.) | 10 | 100 |
| | TabPFN | Diabetes | 8 | 100 |
| | TabPFN | Breast Cancer | 8 | 100 |
| *Data Valuation (DV)* (Ghorbani & Zou, 2019) | RF | Bike Sharing | 10 | 30 |
| | GB | Bike Sharing | 10 | 30 |
| | GB | Cal. Housing | 10 | 30 |
| *Hyperparameter Importance (HPI)* (Wever et al., 2026) | XGBoost | Chess | 16 | 100 |
| | XGBoost | Thyroid | 16 | 100 |
| | LCBench | Jasmine | 8 | 100 |
| *Local Explanation (LE)* (Štrumbelj & Kononenko, 2010; Bifet et al., 2022) | RF | CorrGroups60 | 60 | 30 |
| | RF | NHANES | 79 | 30 |
| | RF | Crime | 101 | 30 |
| | ResNet | ImageNet | 14 | 30 |
| | ViT (9 patches) | ImageNet | 9 | 30 |
| | ViT (16 patches) | ImageNet | 16 | 30 |

coalition sampling (GP + Random); b) coalition sampling according to leverage scores (GP + Leverage Score Sampling; Musco & Witter, 2025), a state-of-the-art coalition sampling strategy in SV estimation; and c) pure uncertainty sampling (GP + US; Lewis & Catlett, 1994), as a widely used general-purpose baseline in BED (Krause et al., 2008; Gunter et al., 2014; Neiswanger et al., 2021) which has also been proposed in the context of BayesSHAP (Slack et al., 2021). This is done to disentangle the contribution of our proposed EIG-based coalition sampling strategy from the effect of using a GP surrogate, and to directly compare it to alternative coalition selection strategies.

**Games.** We consider several small to moderately large games from ML in which the value function is costly to evaluate (see Table 1 for an overview). Specifically, we study a) global feature importance (FI) for the TabPFN foundation model, which relies on in-context learning (Hollmann et al., 2025; Grinsztajn et al., 2025; Rundel et al., 2024); b) dataset valuation (DV; Jia et al., 2019; Ghorbani & Zou, 2019; Tay et al., 2022) for Random Forests (RFs; Breiman, 2001) and Gradient Boosting (GB; Friedman, 2001); c) hyperparameter importance (HPI) according to the HyperSHAP ablation game (Wever et al., 2026) for XGBoost (Binder et al., 2020) and LCBench (Zimmer et al., 2021); and d) local explanation (LE; Štrumbelj & Kononenko, 2010) for the computer vision models ViT (Dosovitskiy et al., 2021) and ResNet (He et al., 2016), which are further relevant SV estimation problems, and for RFs via the linear TreeSHAP algorithm (Bifet et al., 2022). While the latter (LE) quantifies the contribution of features to model predictions for individual data instances, the former three (FI, DV, HPI) quantify the contributions of features, subsets of training data, and hyperparameters, respectively, to the predictive performance of a learning algorithm on a test set.

Since FI, DV, and HPI each require full model retrainings for coalition evaluations, evaluation budgets are often severely

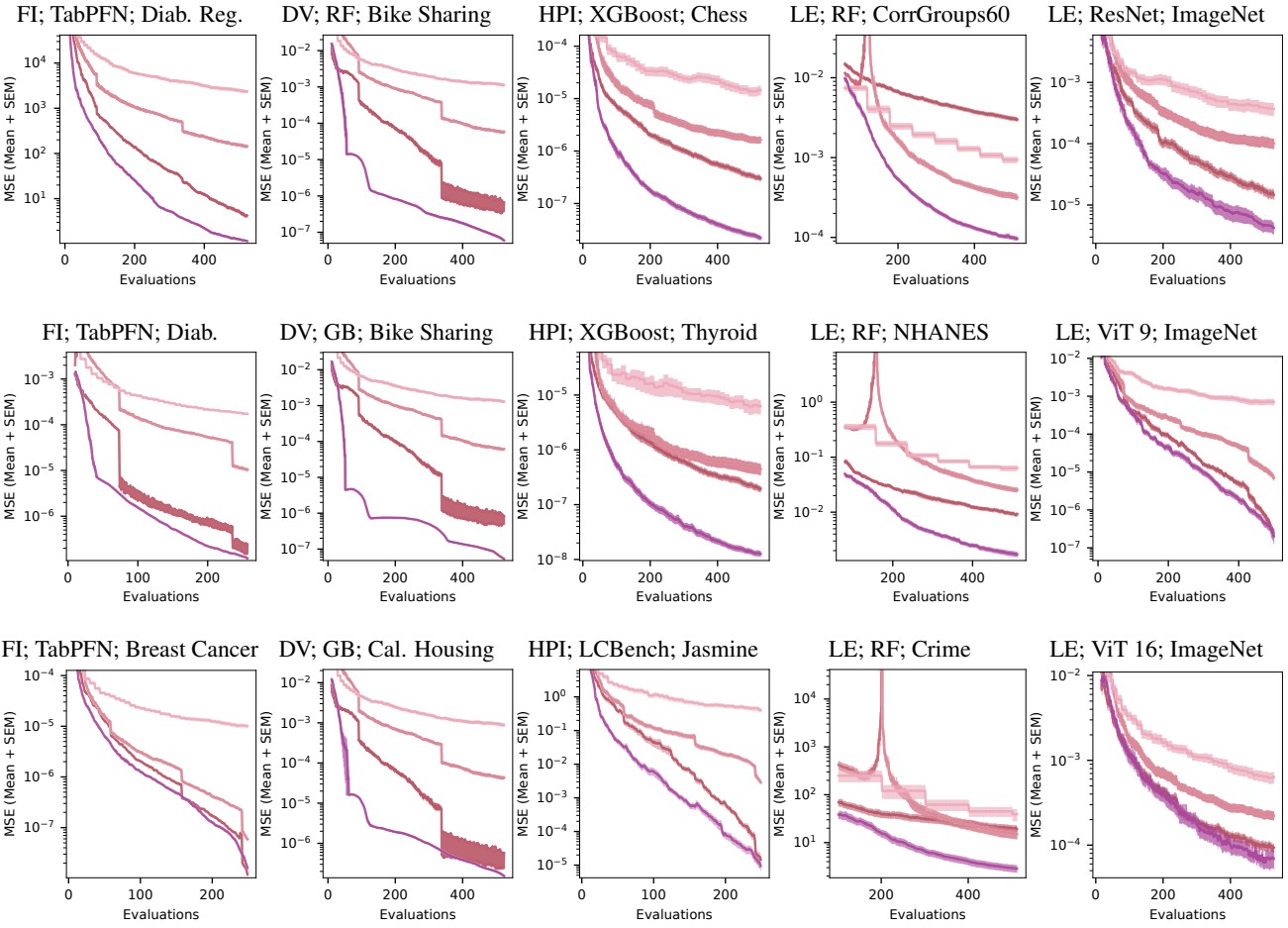

*Figure 1.* Mean squared error (MSE) between estimated and ground-truth Shapley values across all tasks and evaluation budgets, averaged over repetitions for `ShaplEIG` and the SV approximation baselines, with standard error of the mean (SEM) indicated.

`ShaplEIG` (Ours)    Regression MSR    Leverage SHAP    Kernel SHAP    Permutation Sampling

limited in these cases. Such budget constraints can also arise for repeated model evaluations in LEs, e.g., in the context of large foundation models, where each inference may incur monetary costs through APIs.

For our benchmarks, we mostly rely on pre-computed games. This avoids the computational cost associated with actual value function evaluations and thus allows us to efficiently compare estimation methods and compute ground-truth SVs. Although value function evaluations in the context of LEs for tree-based models are not particularly costly, we include these games because linear TreeSHAP enables computing ground-truth SVs even in settings with a large number of players $p$ where exhaustive enumeration of all coalitions is infeasible. We refer to Appendix D.1.3 for additional details on the games.

**Evaluation.** We compare the different methods with respect to their SV estimation accuracy across varying evaluation budgets. Estimation accuracy is measured using

the mean squared error (MSE) between the estimated and ground-truth SVs.

**Scalability.** For larger games ($p > 16$), we do not refit the GP hyperparameters in every iteration, but only according to a refit schedule. In addition to saving the cost of hyperparameter fitting in those iterations, fixed hyperparameters enable efficient updates of the EIG between iterations by allowing us to reuse intermediate quantities. This substantially reduces the computational overhead for larger games , while retaining outcome-adaptivity at the refitting iterations. See Appendix D.1.4 for details.

**Reproducibility.** Our implementation is written in Python, primarily using `BoTorch` (Balandat et al., 2020) and `GPyTorch` (Gardner et al., 2018) for GP surrogates and EIG computation. For the SV estimation baselines, we rely on the `shapiq` package (Muschalik et al., 2024). The repository containing the implementation and all experi-

ments is publicly available, ensuring full reproducibility (see Appendix D.1.5).

For statistical robustness and generalizability, we repeat all experiments with either 100 or 30 random seeds. The number of seeds depends on the size of the game and the source, with the `shapiq` package only providing 30 repetitions for pre-computed games. Generally, the seeds affect the value functions of the underlying games (e.g., data splits and training behavior for model retrainings), the SV approximation baselines, as well as the initial designs and GP hyperparameter optimization procedures used in adaptive methods. We again refer to Table 1 and Appendix D.1.3 for further details.

## 5.2. Results

The results for `ShaplEIG` and the SV approximation baselines across all tasks are presented in Figure 1. The x-axis reports the number of value function evaluations while the y-axis shows the MSE averaged over repetitions and plotted on a logarithmic scale. Error bars indicate the standard error of the mean (SEM).

Across all tasks, our proposed approach `ShaplEIG` consistently achieves the best overall accuracy and is at least as good as all competitors except over very short intervals. In the majority of tasks, it strictly dominates all established competitors from SV approximation across all evaluation budgets, while in the remaining games it is only outperformed by a competitor over short intervals and never by a substantial margin. In particular, Regression MSR is the only method that sometimes achieves competitive performance, while all other competitors are substantially outperformed by `ShaplEIG` in most settings. However, for several tasks, `ShaplEIG` outperforms all competitors - including Regression MSR - by a large margin across all phases, thus demonstrating enhanced sample efficiency in the low-budget regime. We also note that even for the larger tasks, where GP hyperparameters are refit only according to a fixed schedule, `ShaplEIG` remains effective. This indicates that simple strategies for reducing the computational overhead are sufficient to retain the benefits of our method also in larger games.

**Ablations.** Here we summarize the overall findings regarding the ablations of our method. See Appendix D.2.1 (Figure 3) for the complete results across all tasks.

`ShaplEIG` again achieves the best overall performance when compared to all (quite strong) GP-based baselines. While it frequently outperforms the variants with random and US-based coalition selection by a large margin, US often performs even worse than random sampling. This suggests that, despite being a popular approach from BED, US is not particularly effective for SV estimation compared to our approach. Moreover, although leverage score-based

coalition sampling, as a recent state-of-the-art approach from SV estimation, can sometimes come close to the performance of `ShaplEIG` when coupled with a GP, it is still consistently outperformed by our method.

Altogether, this indicates that the strong performance of our approach can only partially be attributed to the use of a GP surrogate, but is instead substantially driven by our principled EIG-based selection strategy. Note, however, that for the large LE games, our approach is outperformed by a small margin in very early stages, i.e., during the first 100 iterations, after which it starts to outperform the ablation variants.

**Computational Cost.** We also analyze the computational cost of `ShaplEIG`. In Appendix D.2.2, we present detailed results for the runtime overhead due to GP hyperparameter fitting (Figure 5) and EIG computation (Figure 6). For smaller games with up to 16 players, the overhead for hyperparameter refitting can reach up to about 2 minutes per iteration, while EIG computation always takes less than a second. For larger games with up to 100 players, the overhead for hyperparameter refitting reaches up to about 25 minutes per iteration, while the overhead for EIG computation remains below 30 seconds.

This indicates the following: 1) For smaller games the overhead is relatively low (seconds to minutes per iteration), enabling the efficient application of `ShaplEIG` even when value functions are not particularly costly; for larger games with up to 100 players, however, the overhead grows disproportionately and can be substantial (minutes to hours per iteration), rendering `ShaplEIG` only appropriate when value functions are genuinely expensive. 2) Hyperparameter training dominates the computational overhead by a large margin. Consequently, alternative GP surrogate-based approaches that do not rely on EIG, as considered in the ablation study, yield similar overhead and are therefore not more efficient than our method.

## 6. Conclusion

In this work, we demonstrated that adaptive BED with GP surrogates can substantially improve sample efficiency for SV estimation in costly games with budget constraints. Central to our approach is the observation that, for SVs, the EIG admits a closed-form expression and can be computed efficiently.

At the same time, the proposed method incurs computational overhead from repeated GP hyperparameter optimization and EIG maximization. Future work should therefore focus on additional computational improvements to further broaden the scope of `ShaplEIG` to games with even more players, to games requiring larger value function evaluation budgets, and to settings with less costly value functions.

## Acknowledgements

Maximilian Muschalik acknowledges funding by the Deutsche Forschungsgemeinschaft (DFG, German Research Foundation): TRR 318/3 2026 – 438445824.

## Impact Statement

This paper presents work whose goal is to advance the field of Machine Learning. There are many potential societal consequences of our work, none which we feel must be specifically highlighted here.

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

# A. Background

## A.1. Gaussian Processes

A Gaussian Process (GP; Williams & Rasmussen, 1995; 2006) is fully characterized by a mean function $m : \mathcal{X} \to \mathbb{R}$ and a positive definite kernel $k_\xi : \mathcal{X} \times \mathcal{X} \to \mathbb{R}$, parameterized by hyperparameters $\xi$. Formally, we write $f \sim \mathcal{GP}(m, k_\xi)$. Given noisy training data $\mathcal{D}_n := (\mathbf{X}^{(n)}, \mathbf{Y}^{(n)})$, where $\mathbf{X}^{(n)} \in \mathbb{R}^{n \times p}$ and $\mathbf{Y}^{(n)} \in \mathbb{R}^n$, with the targets $\mathbf{Y}^{(n)}$ corrupted by homoscedastic, additive Gaussian noise $\epsilon \sim \mathcal{N}(0, \sigma_\epsilon^2)$ that is i.i.d. across evaluations, updating the prior yields a posterior process that is also a GP. The posterior predictive distribution (PPD) over unseen test data $\mathbf{X}^* \in \mathbb{R}^{s \times p}$ is a multivariate Gaussian

$$f(\mathbf{X}^*) \mid \mathcal{D}_n \sim \mathcal{N}_s\big(\boldsymbol{\mu}_{f(\mathbf{X}^*)|\mathcal{D}_n}, \boldsymbol{\Sigma}_{f(\mathbf{X}^*)|\mathcal{D}_n}\big). \tag{7}$$

For a zero mean function $m = 0$, this leads to the mean prediction

$$\boldsymbol{\mu}_{f(\mathbf{X}^*)|\mathcal{D}_n} = K_\xi(\mathbf{X}^*, \mathbf{X}^{(n)}) \big[K_\xi(\mathbf{X}^{(n)}, \mathbf{X}^{(n)}) + \sigma_\epsilon^2 \mathbf{I}\big]^{-1} \mathbf{Y}^{(n)} \tag{8}$$

and covariance matrix

$$\boldsymbol{\Sigma}_{f(\mathbf{X}^*)|\mathcal{D}_n} = K_\xi(\mathbf{X}^*, \mathbf{X}^*) - K_\xi(\mathbf{X}^*, \mathbf{X}^{(n)}) \big[K_\xi(\mathbf{X}^{(n)}, \mathbf{X}^{(n)}) + \sigma_\epsilon^2 \mathbf{I}\big]^{-1} K_\xi(\mathbf{X}^{(n)}, \mathbf{X}^*) \in \mathbb{R}^{s \times s}. \tag{9}$$

Here, $K_\xi(\mathbf{A}, \mathbf{B}) \in \mathbb{R}^{a \times b}$ denotes the kernel matrix with entries $K_\xi(\mathbf{A}, \mathbf{B})_{(i,j)} = k_\xi(\mathbf{A}_{(i,\cdot)}, \mathbf{B}_{(j,\cdot)})$ for data $\mathbf{A} \in \mathbb{R}^{a \times p}$ and $\mathbf{B} \in \mathbb{R}^{b \times p}$.

**Computational Cost.** For the computation of the posterior predictive covariance $\boldsymbol{\Sigma}_{f(\mathbf{X}^*)|\mathcal{D}_n}$ over $\mathbf{X}^*$, the kernel matrix of the training data, $K_\xi(\mathbf{X}^{(n)}, \mathbf{X}^{(n)}) \in \mathbb{R}^{n \times n}$, is typically first decomposed via a Cholesky factorization, where the associated computational cost scales as $\mathcal{O}(n^3)$. The second term of Equation 9 can then be computed by solving a linear system with multiple right-hand sides, which scales as $\mathcal{O}(s \cdot n^2)$, followed by a matrix product that scales as $\mathcal{O}(s^2 \cdot n)$. Altogether, the total computational cost scales as $\mathcal{O}(n^3 + s \cdot n^2 + s^2 \cdot n)$. When only the marginal posterior predictive variances are of interest, corresponding to the diagonal of $\boldsymbol{\Sigma}_{f(\mathbf{X}^*)|\mathcal{D}_n}$, the computational cost reduces to $\mathcal{O}(n^3 + s \cdot n^2)$.

**Covariance Functions.** A popular covariance function for categorical input variables is the Hamming kernel (Platt et al., 2001; Qian et al., 2008; Hutter, 2009). It quantifies the similarity of two data points as

$$k_\xi(\mathbf{x}, \mathbf{x}') = \prod_{j=1}^{p} k_{\xi,j}(\mathbf{x}, \mathbf{x}')$$
$$= \prod_{j=1}^{p} \exp\Big(-\frac{[\mathbf{x}_j \neq \mathbf{x}'_j]}{\ell_j^2}\Big),$$

where $\xi = (\ell_1, \ldots, \ell_p)^\top \in \mathbb{R}^p$ collects the dimension-specific lengthscales. Note that this is a product kernel, meaning that the kernel value factorizes over dimensions.

**Hyperparameter Training.** The kernel hyperparameters $\xi$ (and potentially the noise variance $\sigma_\epsilon^2$) influence the approximation quality of GP models, yet are unknown in practice. Thus, they are typically learned by maximizing the log marginal likelihood (LML) of the training data $\mathcal{D}_n$, or alternatively by maximum a posteriori (MAP) estimation. This optimization balances data fit and model complexity and is usually carried out using gradient-based methods, exploiting the closed-form expression of the marginal likelihood and its derivatives with respect to the hyperparameters.

**(Quasi-) Noiseless GPs.** A noiseless GP is obtained as the limiting case of the above model when the observation noise variance is fixed to zero, i.e., $\sigma_\epsilon^2 = 0$. In this case, assuming that the kernel matrix is non-singular, the posterior mean interpolates the observed training data exactly (interpolation property; Stein, 1999; Williams & Rasmussen, 2006). In practice, however, one typically uses a quasi-noiseless GP, where $\sigma_\epsilon^2$ is fixed to a very small positive constant for numerical stability.

# B. Methodology

## B.1. Shapley Value Estimation

In the following, we discuss further methodological details on the SV estimation approach of `ShaplEIG`.

**Interpolation Property.** We propose extracting SV estimates in `ShaplEIG` via $\hat{\phi} := \mu_{\phi|\mathcal{D}_t}$ using a noiseless GP surrogate. It is important to note that one could alternatively employ a noisy GP in this context. This may lead to better generalization performance, depending on the noise level of the value function. However, in our experiments, we use quasi-noiseless GPs, which is motivated by the following considerations: First, as explained in the main paper (Section 3.2), the SV estimator is consistent when using a noiseless GP. Second, we assume deterministic games, making noiseless GPs a natural choice. Third, in the presented experiments (Section 5), `ShaplEIG` achieves state-of-the-art performance in SV estimation on real-world datasets, which may well be noisy. This suggests that the chosen approach is effective in practice.

Consequently, we leave the analysis of noisy GPs for future work. Nevertheless, we emphasize that noisy GPs may be necessary in some cases and that our current approach has limitations in certain settings.

**Estimator Bias and Debiasing.** We note that the proposed SV estimator, despite being consistent, is not unbiased (MacKay, 1992; Dasgupta & Hsu, 2008; Kossen et al., 2021). This is because the adaptive coalition selection based on the EIG breaks the assumption of independent sampling. However, for adaptive surrogate-based estimators, unbiasedness is typically not the central objective. More generally, in SV estimation, it is common to accept some bias in exchange for lower variance, and thus a reduced MSE of the estimator (as in Kernel SHAP; see the analyses of Covert & Lee (2021) and Kolpaczki et al. (2024) for further details).

Furthermore, note that there exist approaches for debiasing the SV estimator with slight changes to the acquisition strategy. In particular, `ShaplEIG` can be viewed as a special instance of active testing (Kossen et al., 2021; 2022), where the SVs correspond to expectations to be estimated in a sample-efficient way. Farquhar et al. (2021) showed that by (1) sampling coalitions according to an acquisition distribution that can be derived from the EIG scores, rather than selecting the EIG maximizer, and (2) adjusting the SV estimator with importance weights in the Levelled Unbiased Risk Estimator (LURE; Farquhar et al., 2021), the bias from active selection can be corrected. We leave such unbiased, importance-weighted variants for future work.

## B.2. Practical Implications of the EIG-based Coalition Selection

In the following, we provide a detailed discussion of how the covariance-based EIG criterion affects coalition selection in practice, and when it can be expected to provide benefits over alternative selection strategies (Section B.2.1). In addition, we present two examples of games and the behavior of different selection strategies to further illustrate these concepts (Section B.2.2).

### B.2.1. DISCUSSION OF THE EIG-BASED COALITION SELECTION

As indicated by Equation 5, the EIG about the SVs is maximized by the coalition that leads to the lowest determinant of the SV posterior covariance after the associated evaluation is added to the dataset. In particular, this is governed by the expected reduction in uncertainty about the value function across all coalitions, and how this reduction propagates to the SVs through the linear transformation defined by $\mathbf{A}$ (see Equation 4). The GP surrogate from `ShaplEIG` is able to capture this through its Hamming covariance function. It quantifies similarity between coalition pairs based on which players the two coalitions share, weighted by learnable lengthscale parameters that determine how strongly disagreements for specific players reduce covariance. As a result, two coalition pairs with the same Hamming distance (i.e., the same number of players on which they disagree) can still have very different covariance if they differ in players that are more or less influential under the surrogate. Overall, this induces a posterior covariance structure over the coalitions, through which evaluating one coalition not only reduces uncertainty locally at that coalition itself, but also globally at others. Based on this, the EIG favors coalitions that are able to reduce uncertainty jointly about the SVs, as indicated by this covariance structure, while accounting for the coalitions already observed.

This stands in contrast to current state-of-the-art approaches for coalition sampling (e.g., leverage score sampling; Musco & Witter, 2025), which sample coalitions from fixed distributions. Although these distributions typically depend on the size of candidate coalitions, they treat all player differences equally and do not distinguish between different coalitions of the same size. Even uncertainty sampling (US; Lewis & Catlett, 1994), a widely used general-purpose baseline in BED settings (Krause et al., 2008; Gunter et al., 2014; Neiswanger et al., 2021) that does distinguish between coalitions of the same size, does not fully capture this effect either. US selects the coalition with the highest marginal posterior variance under the surrogate, but ignores how its evaluation informs other coalitions.

This also clarifies the regime in which we expect the strategy to be advantageous: asymmetric games, where players have different influence on the value function - e.g., some are highly relevant while others have negligible effect - with interactions occurring primarily among the relevant players. We believe this regime is common in practice, particularly in large games, where it is unlikely that all players contribute symmetrically to the value function.

### B.2.2. ILLUSTRATIVE EXAMPLES

We consider two simple games with three players ($P := \{1, 2, 3\}$) and the initial design

$$\mathcal{D}_{T_0+1} = \{(\emptyset, \nu(\emptyset)), (\{1\}, \nu(\{1\})), (\{2\}, \nu(\{2\})), (\{3\}, \nu(\{3\})), (\{1, 2, 3\}, \nu(\{1, 2, 3\}))\},$$

containing the empty and full coalitions as well as all size-one coalitions. As the candidate set, we consider all size-two coalitions, i.e.,

$$\mathcal{C} = \{\{1, 2\}, \{1, 3\}, \{2, 3\}\}.$$

**Symmetric Games.** First, consider the symmetric value function

$$\nu_{\text{sym}}(\mathcal{S}) = |\mathcal{S}|^2,$$

which contains interactions among players but is identical across all coalitions of the same size. The initial design evaluates to

$$\mathcal{D}_{T_0+1,\text{sym}} = \{(\emptyset, 0), (\{1\}, 1), (\{2\}, 1), (\{3\}, 1), (\{1, 2, 3\}, 9)\}.$$

Since all single-player coalitions yield the same value, GP fitting in the initial `ShaplEIG` iteration produces identical lengthscales for all players, i.e., $\ell = (1.035, 1.035, 1.035)^\top$. As a result, the covariances among all unseen candidates are identical: $k_\xi(\{1, 2\}, \{1, 3\}) = k_\xi(\{1, 2\}, \{2, 3\}) = k_\xi(\{1, 3\}, \{2, 3\})$. Each pair of candidate coalitions shares exactly

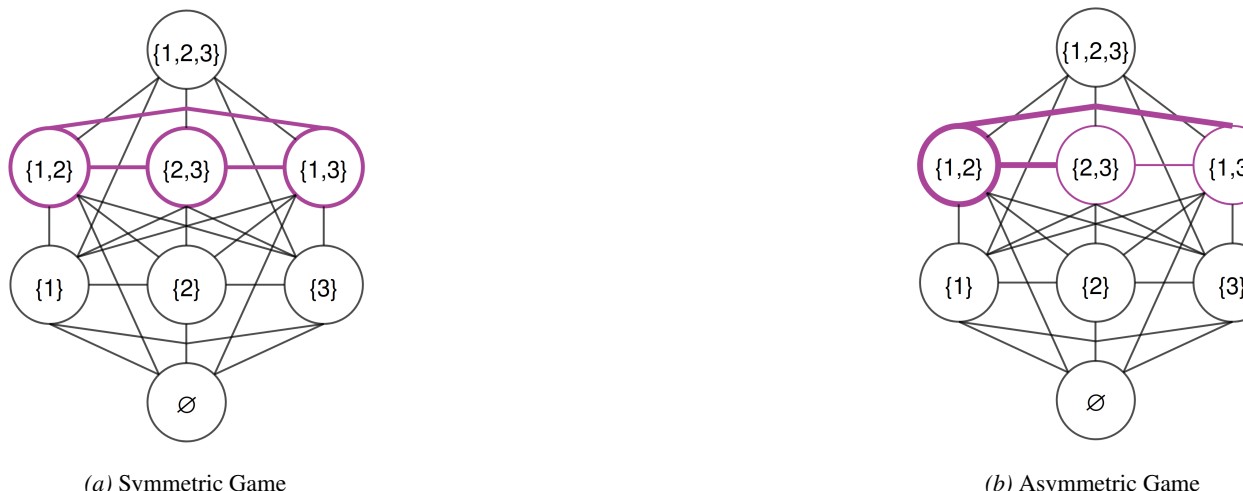

*(a)* Symmetric Game          *(b)* Asymmetric Game

*Figure 2.* Illustration of the behavior of EIG-based coalition selection given the initial design $\mathcal{D}_{T_0+1}$ in symmetric and asymmetric games. The line width of the colored edges represents the magnitude of the covariance between coalition pairs according to the GP surrogate, while the size of the colored nodes represents the EIG scores of the candidate coalitions.

one player, and due to identical lengthscales, differences in player membership do not affect the covariance differently across remaining players. Consequently, the EIG is also identical across all candidates, i.e., $\text{EIG}(\{1,2\}) = \text{EIG}(\{1,3\}) = \text{EIG}(\{2,3\})$. See Figure 2a for an illustration of the covariance structure and EIG in this setting.

Thus, in this setting, EIG provides no advantage over coalition sampling methods that depend solely on coalition size and are commonly used in classical SV estimation.

**Asymmetric Games.** We now consider an asymmetric game with value function

$$\nu_{\text{asym}}(\mathcal{S}) = 1 \cdot [1 \in \mathcal{S}] + 1 \cdot [2 \in \mathcal{S}] + 0.01 \cdot [3 \in \mathcal{S}] + 1 \cdot [\{1,2\} \subseteq \mathcal{S}].$$

Players 1 and 2 have identical main effects, player 3 is negligible, and there is an interaction between players 1 and 2.

Given the initial design

$$\mathcal{D}_{T_0+1,\text{asym}} = \{(\emptyset, 0), (\{1\}, 1), (\{2\}, 1), (\{3\}, 0.01), (\{1,2,3\}, 3.01)\},$$

the fitted GP hyperparameters $\ell = (0.807, 0.807, 3.918)^\top$ already reflect the low relevance of player 3. Consequently, the coalition containing the two relevant players, $\{1,2\}$, exhibits higher covariance with the other candidates than the other size-two coalitions:

$$k_\xi(\{1,2\},\{1,3\}) = k_\xi(\{1,2\},\{2,3\}) > k_\xi(\{1,3\},\{2,3\}).$$

This arises because players 1 and 2 have smaller lengthscales, so sharing one of them reduces variability more. Coalitions sharing only player 3 leave the relevant players in disagreement and therefore have lower covariance. This is directly reflected in $\mathbf{Q}_{i,i}$ (Equation 12), which depends on these covariances, and as a result the EIG is maximized for this coalition: $\text{EIG}(\{1,2\}) > \text{EIG}(\{1,3\}) = \text{EIG}(\{2,3\})$. Intuitively, this candidate, due to its highest correlation with the remaining coalitions, is expected to yield the largest reduction in uncertainty about the SVs. See Figure 2b for an illustration of the covariance structure and EIG in this setting.

Consequently, ShaplEIG selects $\{1,2\}$ for evaluation, revealing the key interaction. This reduces the MSE of the SV estimation from 0.024 to 5.485e-06. In contrast, classical size-based coalition samplers choose all three candidates with equal probability. As a result, it is more likely that either $\{1,3\}$ or $\{2,3\}$ is selected, neither of which reveals the relevant interaction and therefore leads to a substantially smaller reduction in MSE (from 0.024 to 8.75e-4).

This example demonstrates that in asymmetric games, where some players are highly relevant to the value function while others have negligible effect, and interactions occur primarily among the relevant players, ShaplEIG can exploit this structure to identify informative coalitions. In contrast, classical coalition sampling schemes often fail to do so, as they treat all players identically.

## B.3. Closed-form EIG for the Shapley Values

In the following, we will derive the closed-form expression for the EIG of a candidate coalition about the SVs as presented in Formula 6 in Section 3.4 of the main text.

In Section 3.3, we established that adaptively selecting data for GP surrogate training can be viewed as a Bayesian linear inverse problem and that, since the SVs are given by a linear transformation of $\boldsymbol{\nu}$, the EIG admits a closed-form expression. Consequently, according to Equation 3 in Section 2.2, the EIG about the SVs $\boldsymbol{\phi}$ for a candidate coalition $\mathbf{z}^{(i)} \in \{0,1\}^p$ at iteration $t$ is given by:

$$\begin{aligned}
\text{EIG}_{\boldsymbol{\phi}}^{(t)}(\mathbf{z}^{(i)}) &= I\big(\boldsymbol{\phi};\, \nu'(\mathbf{z}^{(i)}) \mid \mathcal{D}_t\big) \\
&\propto -\log \det\big(\mathbf{A}\boldsymbol{\Sigma}_{(\boldsymbol{\nu}|\nu'(\mathbf{z}^{(i)}),\mathcal{D}_t)}\mathbf{A}^\top\big) + C,
\end{aligned}$$

where $C$ is a constant independent of $\mathbf{z}^{(i)}$, and the posterior covariance can be expressed as

$$\boldsymbol{\Sigma}_{(\boldsymbol{\nu}|\nu'(\mathbf{z}^{(i)}),\mathcal{D}_t)} = \big(\boldsymbol{\Sigma}_{(\boldsymbol{\nu}|\mathcal{D}_t)}^{-1} + \mathbf{e}_i\sigma_\epsilon^{-2}\mathbf{e}_i^\top\big)^{-1}.$$

It can be rearranged into the following form, as presented in Formula 6 in Section 3.4 of the main text:

$$\text{EIG}_{\boldsymbol{\phi}}^{(t)}(\mathbf{z}^{(i)}) \propto C' + \log\left[\mathbf{e}_i^\top\big(\boldsymbol{\Sigma}_{(\boldsymbol{\nu}|\mathcal{D}_t)} + \sigma_\epsilon^2\mathbf{I}\big)\mathbf{e}_i\right] - \log\left[\mathbf{e}_i^\top\big(\boldsymbol{\Sigma}_{(\boldsymbol{\nu}|\mathcal{D}_t)} + \sigma_\epsilon^2\mathbf{I} - \mathbf{Q}\big)\mathbf{e}_i\right],$$

where $C'$ is constant, $\mathbf{I} \in \mathbb{R}^{2^p \times 2^p}$ is the identity matrix, and $\mathbf{Q} \in \mathbb{R}^{2^p \times 2^p}$ is defined as

$$\mathbf{Q} = \big(\mathbf{A}\boldsymbol{\Sigma}_{(\boldsymbol{\nu}|\mathcal{D}_t)}\big)^\top\big(\mathbf{A}\boldsymbol{\Sigma}_{(\boldsymbol{\nu}|\mathcal{D}_t)}\mathbf{A}^\top\big)^{-1}\big(\mathbf{A}\boldsymbol{\Sigma}_{(\boldsymbol{\nu}|\mathcal{D}_t)}\big).$$

Specifically, this follows by direct application of Sherman-Morrison:

$$\mathbf{A}\big(\boldsymbol{\Sigma}_{(\boldsymbol{\nu}|\mathcal{D}_t)}^{-1} + \mathbf{e}_i\sigma_\epsilon^{-2}\mathbf{e}_i^\top\big)^{-1}\mathbf{A}^\top = \mathbf{A}\boldsymbol{\Sigma}_{(\boldsymbol{\nu}|\mathcal{D}_t)}\mathbf{A}^\top - \mathbf{A}\boldsymbol{\Sigma}_{(\boldsymbol{\nu}|\mathcal{D}_t)}\mathbf{e}_i\big(\sigma_\epsilon^2 + \mathbf{e}_i^\top\boldsymbol{\Sigma}_{(\boldsymbol{\nu}|\mathcal{D}_t)}\mathbf{e}_i\big)^{-1}\mathbf{e}_i^\top\boldsymbol{\Sigma}_{(\boldsymbol{\nu}|\mathcal{D}_t)}\mathbf{A}^\top,$$

and by the matrix determinant lemma (see, e.g., Williams & Rasmussen (2006) A.3):

$$\begin{aligned}
\det\big(\mathbf{A}\boldsymbol{\Sigma}_{(\boldsymbol{\nu}|\nu'(\mathbf{z}^{(i)}),\mathcal{D}_t)}\mathbf{A}^\top\big) &= \det\Big(\mathbf{A}\boldsymbol{\Sigma}_{(\boldsymbol{\nu}|\mathcal{D}_t)}\mathbf{A}^\top - \mathbf{A}\boldsymbol{\Sigma}_{(\boldsymbol{\nu}|\mathcal{D}_t)}\mathbf{e}_i\big(\sigma_\epsilon^2 + \mathbf{e}_i^\top\boldsymbol{\Sigma}_{(\boldsymbol{\nu}|\mathcal{D}_t)}\mathbf{e}_i\big)^{-1}\mathbf{e}_i^\top\boldsymbol{\Sigma}_{(\boldsymbol{\nu}|\mathcal{D}_t)}\mathbf{A}^\top\Big) \\
&= \det\Big(\mathbf{A}\boldsymbol{\Sigma}_{(\boldsymbol{\nu}|\mathcal{D}_t)}\mathbf{A}^\top\Big) \cdot \det\Big(\big(\sigma_\epsilon^2 + \mathbf{e}_i^\top\boldsymbol{\Sigma}_{(\boldsymbol{\nu}|\mathcal{D}_t)}\mathbf{e}_i\big)^{-1}\Big) \\
&\quad \cdot \det\Big(\sigma_\epsilon^2 + \mathbf{e}_i^\top\boldsymbol{\Sigma}_{(\boldsymbol{\nu}|\mathcal{D}_t)}\mathbf{e}_i - \mathbf{e}_i^\top\boldsymbol{\Sigma}_{(\boldsymbol{\nu}|\mathcal{D}_t)}\mathbf{A}^\top\big(\mathbf{A}\boldsymbol{\Sigma}_{(\boldsymbol{\nu}|\mathcal{D}_t)}\mathbf{A}^\top\big)^{-1}\mathbf{A}\boldsymbol{\Sigma}_{(\boldsymbol{\nu}|\mathcal{D}_t)}\mathbf{e}_i\Big).
\end{aligned}$$

In this final formulation, the first product term does not depend on $\mathbf{z}^{(i)}$, and for the latter two terms, the $\det$ operators can be dropped, as they operate only on scalars.

## B.4. Efficient EIG Computation

In the following, we present the proof of Theorem 3.1 and thereby derive an efficient computation scheme for the EIG of a single candidate coalition $\mathbf{z}^{(i)} \in \{0,1\}^p$ about the SVs $\phi$ (Formula 6 in Section 3.4). In doing so, we exploit the multiplicative structure of the Hamming kernel via a correspondence to elementary symmetric polynomials (ESPs; van Es & Helmers, 1988; Macdonald, 1998; Charalambides, 2018). We also show how the EIG can be computed across various candidates in a vectorized manner (Subsection B.4.1).

Specifically, we will first show in Theorem B.1 how the linear term $\mathbf{A}K_\xi(\mathbf{Z}, \mathbf{z}^{(i)}) \in \mathbb{R}^p$, required for the computation of $\mathbf{A}\boldsymbol{\Sigma}_{(\boldsymbol{\nu}|\mathcal{D}_t)}\mathbf{e}_i$, can be computed in $\mathcal{O}(p^2)$ and in Theorem B.2 how the quadratic term $\mathbf{A}K_\xi(\mathbf{Z}, \mathbf{Z})\mathbf{A}^\top \in \mathbb{R}^{p \times p}$, required for the computation of $\mathbf{A}\boldsymbol{\Sigma}_{(\boldsymbol{\nu}|\mathcal{D}_t)}\mathbf{A}^\top$, can be computed in $\mathcal{O}(p^4)$. Based on these results we will then show in Theorem B.3 how the complete EIG can be computed efficiently in $\mathcal{O}(p^4 + t^3)$, thereby avoiding exponential scaling in $p$.

We assume $2^p > t > p$, which is aligned with our experiments where the initial design is of size $T_0 = p + 1$. Also, we assume a Hamming kernel with fixed length scales $\xi$ (see Section A.1):

$$k_\xi(\mathbf{x}, \mathbf{x}') = \prod_{j=1}^{p} \exp\Big( - \frac{[\mathbf{x}_j \neq \mathbf{x}'_j]}{\ell_j^2} \Big) \tag{10}$$

$$= \prod_{j=1}^{p} \big( \alpha_j[\mathbf{x}_j = \mathbf{x}'_j] + \beta_j[\mathbf{x}_j \neq \mathbf{x}'_j] \big), \tag{11}$$

where $\alpha_j = 1$ and $\beta_j = \exp(-\ell_j^{-2})$. This rewriting highlights that each multiplicative term can only take on two values, depending on whether the arguments are equal in the corresponding dimension.

Similar techniques for the linear part $\mathbf{A}K_\xi(\mathbf{Z}, \mathbf{z}^{(i)})$ have been considered for the computation of products between Shapley weights and a kernel matrix by Mohammadi et al. (2025a), which is currently only available on arXiv. However, their approach differs from ours in that it is based on unscaled ESPs, whereas we use scaled ESPs. Furthermore, we are not aware of any prior work regarding the quadratic part $\mathbf{A}K_\xi(\mathbf{Z}, \mathbf{Z})\mathbf{A}^\top$. For completeness, we present the full construction for our specific setting using a Hamming kernel with length scales.

**Theorem B.1.** *The term $\mathbf{A}K_\xi(\mathbf{Z}, \mathbf{z}^{(i)}) \in \mathbb{R}^p$, for a candidate coalition $\mathbf{z}^{(i)} \in \{0,1\}^p$, is efficiently computable in $\mathcal{O}(p^2)$ time.*

*Proof.* In the following, we will consider the matrix-vector multiplication $\mathbf{A}K_\xi(\mathbf{Z}, \mathbf{z}^{(i)}) \in \mathbb{R}^p$, where $\mathbf{A} \in \mathbb{R}^{p \times 2^p}$ and $K_\xi(\mathbf{Z}, \mathbf{z}^{(i)}) \in \mathbb{R}^{2^p}$. Initially, we will show that the entries can be rewritten as weighted sums of kernel evaluations across coalitions with weights according to the Shapley matrix $\mathbf{A}$. We then recognize that the additive kernel evaluations associated with certain groups of coalitions share the same weights and show how inner sums of kernel values over these groups can be computed more efficiently by identifying them with scaled, univariate ESPs. In detail, we will show how the sums of kernel values over all coalitions of size $k$, all coalitions of size $k$ that contain a certain player $j$, and all coalitions of size $k$ that do not contain a certain player $j$ can be computed efficiently. Lastly, we will combine everything to show how the complete term can be computed in $\mathcal{O}(p^2)$ time.

**Rewriting entries of $\mathbf{A}K_\xi(\mathbf{Z}, \mathbf{z}^{(i)})$ as sums of kernel evaluations for coalition groups of identical size.** Consider $\mathbf{A}K_\xi(\mathbf{Z}, \mathbf{z}^{(i)})$, which computes the SVs of the kernel vector $K_\xi(\mathbf{Z}, \mathbf{z}^{(i)})$. The $j$-th entry of this vector, corresponding to a player $j \in \{1, \ldots, p\}$, is given by the following summation over all coalitions:

$$(\mathbf{A}K_\xi(\mathbf{Z}, \mathbf{z}^{(i)}))_j = \sum_{S \subseteq \{1,\ldots,p\}} \mathbf{A}_{j,S}\, k_\xi(\mathbf{1}_S, \mathbf{z}^{(i)}).$$

Plugging in the definition of the Shapley matrix for the entry corresponding to player $j$ and a coalition $S$

$$\mathbf{A}_{j,S} = \frac{1}{p}\left(\frac{[j \in S]}{\binom{p-1}{|S|-1}} - \frac{[j \notin S]}{\binom{p-1}{|S|}}\right)$$

equivalently gives

$$
\begin{aligned}
(\mathbf{A}K_\xi(\mathbf{Z}, \mathbf{z}^{(i)}))_j =& \frac{1}{p} \sum_{S \subseteq \{1,\ldots,p\}} \frac{[j \in S]}{\binom{p-1}{|S|-1}} k_\xi(\mathbf{1}_S, \mathbf{z}^{(i)}) \\
& - \frac{1}{p} \sum_{S \subseteq \{1,\ldots,p\}} \frac{[j \notin S]}{\binom{p-1}{|S|}} k_\xi(\mathbf{1}_S, \mathbf{z}^{(i)}).
\end{aligned}
$$

Note that the weights according to the Shapley matrix $\mathbf{A}$, besides the total amount of players $p$, depend only on the size of a coalition $S$, and whether the player $j$ is in $S$ or not. More specifically, in the first sum, the indicator $[j \in S]$ restricts the sum to coalitions where $j$ is in $S$, and in the second sum to coalitions where $j$ is not in $S$. Consequently, both sums can be grouped by coalition sizes $k$, with identical weights for each group, and we obtain the following reformulation:

$$
\begin{aligned}
(\mathbf{A}K_\xi(\mathbf{Z}, \mathbf{z}^{(i)}))_j =& \sum_{k=1}^{p} \frac{1}{p\binom{p-1}{k-1}} \sum_{\substack{S \subseteq \{1,\ldots,p\}: \\ |S|=k,\ j \in S}} k_\xi(\mathbf{1}_S, \mathbf{z}^{(i)}) \\
& - \sum_{k=0}^{p-1} \frac{1}{p\binom{p-1}{k}} \sum_{\substack{S \subseteq \{1,\ldots,p\}: \\ |S|=k,\ j \notin S}} k_\xi(\mathbf{1}_S, \mathbf{z}^{(i)}).
\end{aligned}
$$

Thereby, for the former sum, where $j \in S$, only sizes $k \geq 1$ are possible, and for the latter sum, only sizes $k \leq p-1$ are possible.

**Lemma 1 - Computing sums over coalitions of size $k$.** Consider $k_\xi(\mathbf{1}_S, \mathbf{z}^{(i)})$ for a candidate coalition $\mathbf{z}^{(i)} \in \{0,1\}^p$. We now define the helper variables $\boldsymbol{\gamma} = (\gamma_1, \ldots, \gamma_p)^\top \in \mathbb{R}^p$ and $\boldsymbol{\delta} = (\delta_1, \ldots, \delta_p)^\top \in \mathbb{R}^p$ for each $j$ as follows:

$$\gamma_j = \begin{cases} \alpha_j & \mathbf{z}_j^{(i)} = 0 \\ \beta_j & \mathbf{z}_j^{(i)} = 1 \end{cases}, \qquad \delta_j = \begin{cases} \beta_j & \mathbf{z}_j^{(i)} = 0 \\ \alpha_j & \mathbf{z}_j^{(i)} = 1 \end{cases}.$$

Here, given a specific $\mathbf{z}^{(i)}$, $\gamma_j$ corresponds to the $j$-th multiplicative term of the kernel value $k_\xi(\mathbf{1}_S, \mathbf{z}^{(i)})$ in the case that $j \notin S$, while $\delta_j$ is the $j$-th multiplicative term when $j \in S$. For a fixed $\mathbf{z}^{(i)}$, this reduces the $j$-th multiplicative term of $k_\xi(\mathbf{1}_S, \mathbf{z}^{(i)})$ to two possible values and enables a rewriting that depends only on the coalition $S$:

$$k_{\xi, \mathbf{z}^{(i)}}(\mathbf{1}_S) = k_\xi(\mathbf{1}_S, \mathbf{z}^{(i)})$$
$$= \prod_{j \notin S} \gamma_j \prod_{j \in S} \delta_j.$$

Now consider the following generating polynomial in its factorised and expanded form respectively:

$$P(\zeta) = \prod_{j=1}^{p} (\gamma_j + \delta_j \zeta) = \sum_{k=0}^{p} c_k \zeta^k.$$

This constitutes a univariate polynomial in $\zeta$ of degree $p$, with coefficients $c_k$ for $k = 0, \ldots, p$. Notably, the coefficient $c_k$ for the $k$-th power of $\zeta$ corresponds to the sum of all kernel values $k_\xi(\mathbf{1}_S, \mathbf{z}^{(i)})$ for coalitions $S$ of size $k$:

$$c_k = \sum_{\substack{S \subseteq \{1, \ldots, p\}: \\ |S| = k}} k_\xi(\mathbf{1}_S, \mathbf{z}^{(i)}).$$

*Proof.* Expanding the factorised polynomial means that, from each factor $(\gamma_j + \delta_j \zeta)$ for $j = 1, \ldots, p$, one chooses exactly one of the two summands, multiplies all chosen summands and sums over all possible choices for such monomials. Each such choice pattern is naturally encoded by a coalition $S \subseteq \{1, \ldots, p\}$, where

$$S = \{j \in \{1, \ldots, p\} : \text{One chooses the term } \delta_j \zeta \text{ in factor } j\}.$$

Equivalently, for a given $S$, one chooses $\delta_j \zeta$ if $j \in S$, and $\gamma_j$ otherwise. This produces the monomial

$$\Big(\prod_{j \notin S} \gamma_j \prod_{j \in S} \delta_j\Big) \zeta^{|S|}.$$

Grouping all possible monomials by their degree $k$ (i.e., by the size of the corresponding coalition $S$) and summing over their coefficients gives the coefficient $c_k$ of $\zeta^k$:

$$c_k = \sum_{\substack{S \subseteq \{1, \ldots, p\}: \\ |S| = k}} \Big(\prod_{j \notin S} \gamma_j \prod_{j \in S} \delta_j\Big)$$
$$= \sum_{\substack{S \subseteq \{1, \ldots, p\}: \\ |S| = k}} k_\xi(\mathbf{1}_S, \mathbf{z}^{(i)}).$$

This is exactly the sum of all kernel values $k_\xi(\mathbf{1}_S, \mathbf{z}^{(i)})$ for coalitions $S$ of size $k$, as claimed. $\qquad\square$

Note that the polynomial $P(\zeta)$ can equivalently be rewritten as a generating polynomial whose coefficients are scaled ESPs:

$$P(\zeta) = \prod_{j=1}^{p} (\gamma_j + \delta_j \zeta)$$
$$= \Big(\prod_{j=1}^{p} \gamma_j\Big) \prod_{j=1}^{p} \Big(1 + \frac{\delta_j}{\gamma_j} \zeta\Big)$$
$$= \sum_{k=0}^{p} \Big(\prod_{j=1}^{p} \gamma_j\Big) e_k\big(\boldsymbol{\delta} \oslash \boldsymbol{\gamma}\big) \zeta^k.$$

Here $e_k(\boldsymbol{\delta} \oslash \boldsymbol{\gamma})$ denotes the $k$-th ESP in the variables $\boldsymbol{\delta} \oslash \boldsymbol{\gamma} \in \mathbb{R}^p$, with $\oslash$ denoting the element-wise division, and the product over $\boldsymbol{\gamma}$ constitutes the associated scaling.

**Lemma 2 - Computing sums over coalitions of size $k$ containing a player $j$.** Consider the following generating polynomial, which is obtained by dividing $P(\zeta)$ by the factor corresponding to a player $j \in \{1, \ldots, p\}$, in its factorised and expanded form respectively:

$$Q_j(\zeta) = \frac{P(\zeta)}{(\gamma_j + \delta_j \zeta)} = \prod_{r \in \{1,\ldots,p\} \setminus \{j\}} (\gamma_r + \delta_r \zeta) = \sum_{k=0}^{p-1} d_k^{(j)} \zeta^k.$$

This constitutes a univariate polynomial in $\zeta$ of degree $p - 1$, with coefficients $d_k^{(j)}$ for $k = 0, \ldots, p - 1$. Notably, for $k \in \{1, \ldots, p\}$, the coefficient $d_{k-1}^{(j)}$ for the $(k-1)$-th power of $\zeta$ multiplied by $\delta_j$ corresponds to the sum of all kernel values $k_\xi(\mathbf{1}_S, \mathbf{z}^{(i)})$ for coalitions $S$ of size $k$ that contain the player $j$:

$$d_{k-1}^{(j)} \delta_j = \sum_{\substack{S \subseteq \{1,\ldots,p\}: \\ |S|=k,\ j \in S}} k_\xi(\mathbf{1}_S, \mathbf{z}^{(i)}).$$

*Proof.* Similar to Lemma 1, expanding the factorised polynomial $Q_j(\zeta)$ means that, for each $r \in \{1, \ldots, p\} \setminus \{j\}$, one chooses exactly one of the two summands, multiplies all chosen summands and sums over all possible choices for such monomials. Each such choice pattern is naturally encoded by a coalition $T \subseteq \{1, \ldots, p\} \setminus \{j\}$, where

$$T = \{r \in \{1, \ldots, p\} \setminus \{j\} : \text{One chooses the term } \delta_r \zeta \text{ in factor } r\}.$$

Equivalently, for a given $T$, one chooses $\delta_r \zeta$ if $r \in T$, and $\gamma_r$ otherwise. This produces the monomial

$$\Big( \prod_{r \notin T,\ r \neq j} \gamma_r \prod_{r \in T} \delta_r \Big) \zeta^{|T|}.$$

Grouping all possible monomials by their degree $k - 1$ (i.e., by the size of the corresponding coalition $T$) and summing over their coefficients gives the coefficient $d_{k-1}^{(j)}$ of $\zeta^{k-1}$:

$$d_{k-1}^{(j)} = \sum_{\substack{T \subseteq \{1,\ldots,p\} \setminus \{j\}: \\ |T|=k-1}} \Big( \prod_{r \notin T,\ r \neq j} \gamma_r \prod_{r \in T} \delta_r \Big).$$

Now map such $T$ to a coalition $S = T \cup \{j\}$, so $|S| = k$ and $j \in S$. Summing over the coefficients of the monomials for all such $S$ (equivalently, over all $T \subseteq \{1, \ldots, p\} \setminus \{j\}$ with $|T| = k - 1$) gives:

$$\sum_{\substack{S \subseteq \{1,\ldots,p\}: \\ |S|=k,\ j \in S}} k_\xi(\mathbf{1}_S, \mathbf{z}^{(i)}) = \sum_{\substack{S \subseteq \{1,\ldots,p\}: \\ |S|=k,\ j \in S}} \Big( \prod_{r \notin S} \gamma_r \prod_{r \in S} \delta_r \Big)$$

$$= \sum_{\substack{T \subseteq \{1,\ldots,p\} \setminus \{j\}: \\ |T|=k-1}} \Big( \prod_{r \notin T,\ r \neq j} \gamma_r \prod_{r \in T} \delta_r \Big) \delta_j$$

$$= d_{k-1}^{(j)} \delta_j.$$

This is exactly the sum of all kernel values $k_\xi(\mathbf{1}_S, \mathbf{z}^{(i)})$ for coalitions $S$ of size $k$ that contain the player $j$, as claimed. $\square$

**Lemma 3 - Computing sums over coalitions of size $k$ not containing a player $j$.** It follows trivially that the sum of all kernel values $k_\xi(\mathbf{1}_S, \mathbf{z}^{(i)})$ for coalitions $S$ of size $k$ that do not contain the player $j$ can be computed as the difference between the sum of all kernel values for coalitions $S$ of size $k$ (Lemma 1) and the sum of all kernel values for coalitions $S$ of size $k$ that contain the player $j$ (Lemma 2):

$$\sum_{\substack{S \subseteq \{1,\ldots,p\}: \\ |S|=k,\ j \notin S}} k_\xi(\mathbf{1}_S, \mathbf{z}^{(i)}) = c_k - d_{k-1}^{(j)} \delta_j,$$

for $k = 0, \ldots, p - 1$. Here we define $d_{-1}^{(j)} := 0$, so that the identity also covers the case $k = 0$.

**Combining the results.** Combining the results from the previous paragraphs, we obtain the following identity for the $j$-th entry of $\mathbf{A}K_\xi(\mathbf{Z}, \mathbf{z}^{(i)})$, which reduces the expression to weighted sums of size $p$:

$$(\mathbf{A}K_\xi(\mathbf{Z}, \mathbf{z}^{(i)}))_j = \sum_{k=1}^{p} \frac{1}{p\binom{p-1}{k-1}} \sum_{\substack{S \subseteq \{1,...,p\}: \\ |S|=k, \, j \in S}} k_\xi(\mathbf{1}_S, \mathbf{z}^{(i)}) - \sum_{k=0}^{p-1} \frac{1}{p\binom{p-1}{k}} \sum_{\substack{S \subseteq \{1,...,p\}: \\ |S|=k, \, j \notin S}} k_\xi(\mathbf{1}_S, \mathbf{z}^{(i)})$$

$$= \sum_{k=1}^{p} \frac{1}{p\binom{p-1}{k-1}} \left( d_{k-1}^{(j)} \delta_j \right) - \sum_{k=0}^{p-1} \frac{1}{p\binom{p-1}{k}} \left( c_k - d_{k-1}^{(j)} \delta_j \right).$$

$\square$

**Implementation.** To compute the coefficients of $P(\zeta)$ we propose the following approach: For a given $\mathbf{z}^{(i)}$, first compute the helper variables $\boldsymbol{\gamma}$ and $\boldsymbol{\delta}$ from the kernel values $(\alpha_j, \beta_j)$ for $j = 1,...,p$. This costs $\mathcal{O}(p)$. The coefficients $c_k$ for $k = 0,...,p$ can then be obtained by the following standard degree-by-degree dynamic program: Initialize

$$c_0^{(0)} = 1, \qquad c_k^{(0)} = 0 \ (k \geq 1),$$

and in iterations $j = 1, \ldots, p$ update

$$c_k^{(j)} = \gamma_j \, c_k^{(j-1)} + \delta_j \, c_{k-1}^{(j-1)} \qquad (k = 0, \ldots, j),$$

with the convention $c_{-1}^{(j-1)} = 0$. This computes all coefficients in $\mathcal{O}(p^2)$ time. Given the coefficients $c_k$ of $P(\zeta)$, we can compute the coefficients of $Q_j(\zeta)$ for each $j = 1,...,p$ by dividing the degree-$p$ polynomial $P(\zeta)$ by the linear factor $(\gamma_j + \delta_j \zeta)$ via synthetic division. This scales as $\mathcal{O}(p)$ per $j$, and thus $\mathcal{O}(p^2)$ for all $j$. Based on $P(\zeta)$ and $Q_j(\zeta)$ for each $j = 1,...,p$, the entire term $\mathbf{A}K_\xi(\mathbf{Z}, \mathbf{z}^{(i)})$ is computable in $\mathcal{O}(p^2)$.

Alternatively, for a numerically more stable implementation, one can avoid polynomial division and obtain the required quantities by convolution of coefficients of prefix- and suffix-polynomials.

**Theorem B.2.** *The term $\mathbf{A}K_\xi(\mathbf{Z}, \mathbf{Z})\mathbf{A}^\top \in \mathbb{R}^{p \times p}$ is efficiently computable in $\mathcal{O}(p^4)$ time.*

*Proof.* In the following, we will consider the matrix-valued quadratic function $\mathbf{A}K_\xi(\mathbf{Z}, \mathbf{Z})\mathbf{A}^\top \in \mathbb{R}^{p \times p}$, where $\mathbf{A} \in \mathbb{R}^{p \times 2^p}$ and $K_\xi(\mathbf{Z}, \mathbf{Z}) \in \mathbb{R}^{2^p \times 2^p}$. Initially, we will show how each entry of this matrix can be rewritten as a weighted sum of kernel evaluations across coalition pairs with weights according to the Shapley matrix $\mathbf{A}$. We then recognize that the additive kernel evaluations associated with certain groups of coalition pairs share the same weights. We will then show how inner sums of kernel values for these groups can be computed more efficiently by identifying them with scaled, bivariate ESPs. In detail, we will show how the sums of kernel values over all coalition pairs of size $(a, b)$ can be computed efficiently, and how the sums over coalition pairs of sizes $(a, b)$ (not) containing players $i$ and $j$ can be computed for $i \neq j$ and for $i = j$ respectively. Lastly, we will combine everything and propose an implementation to compute the complete term in $\mathcal{O}(p^4)$ time.

**Rewriting entries of $\mathbf{A}K_\xi(\mathbf{Z}, \mathbf{Z})\mathbf{A}^\top$ as sums of kernel evaluations for coalition pair groups of identical size.** Consider the $(i, j)$ entry of $\mathbf{A}K_\xi(\mathbf{Z}, \mathbf{Z})\mathbf{A}^\top$, where $i, j \in \{1, \ldots, p\}$, which corresponds to the covariance between the SVs of players $i$ and $j$:

$$(\mathbf{A}K_\xi(\mathbf{Z}, \mathbf{Z})\mathbf{A}^\top)_{i,j} = \sum_{S,T \subseteq \{1,\ldots,p\}} \mathbf{A}_{i,S}\, \mathbf{A}_{j,T}\, k_\xi(\mathbf{1}_S, \mathbf{1}_T).$$

By plugging in the definition of the Shapley matrix $\mathbf{A}$ we equivalently obtain

$$\begin{aligned}
(\mathbf{A}K_\xi(\mathbf{Z}, \mathbf{Z})\mathbf{A}^\top)_{i,j} = &\sum_{S,T \subseteq \{1,\ldots,p\}} \frac{1}{p^2} \frac{[i \in S]}{\binom{p-1}{|S|-1}} \frac{[j \in T]}{\binom{p-1}{|T|-1}} k_\xi(\mathbf{1}_S, \mathbf{1}_T) \\
&- \sum_{S,T \subseteq \{1,\ldots,p\}} \frac{1}{p^2} \frac{[i \in S]}{\binom{p-1}{|S|-1}} \frac{[j \notin T]}{\binom{p-1}{|T|}} k_\xi(\mathbf{1}_S, \mathbf{1}_T) \\
&- \sum_{S,T \subseteq \{1,\ldots,p\}} \frac{1}{p^2} \frac{[i \notin S]}{\binom{p-1}{|S|}} \frac{[j \in T]}{\binom{p-1}{|T|-1}} k_\xi(\mathbf{1}_S, \mathbf{1}_T) \\
&+ \sum_{S,T \subseteq \{1,\ldots,p\}} \frac{1}{p^2} \frac{[i \notin S]}{\binom{p-1}{|S|}} \frac{[j \notin T]}{\binom{p-1}{|T|}} k_\xi(\mathbf{1}_S, \mathbf{1}_T).
\end{aligned}$$

This reflects four possible cases based on whether $i$ is in $S$ or not, and whether $j$ is in $T$ or not. Note that the weights, besides the total amount of players $p$, only depend on the sizes of the coalitions $S$ and $T$, and whether they contain the players $i$ and $j$ respectively. Consequently, the double sums can be grouped by coalition pair sizes $(a, b) = (|S|, |T|)$ and contained players $i$ and $j$, with identical weights for each group. This gives the following reformulation:

$$\begin{aligned}
(\mathbf{A}K_\xi(\mathbf{Z}, \mathbf{Z})\mathbf{A}^\top)_{i,j} = &\sum_{a=1}^{p}\sum_{b=1}^{p} \frac{1}{p^2 \binom{p-1}{a-1}\binom{p-1}{b-1}} \Bigg( \sum_{\substack{S,T \subseteq \{1,\ldots,p\}: \\ |S|=a,\, |T|=b,\, i \in S,\, j \in T}} k_\xi(\mathbf{1}_S, \mathbf{1}_T) \Bigg) \\
&- \sum_{a=1}^{p}\sum_{b=0}^{p-1} \frac{1}{p^2 \binom{p-1}{a-1}\binom{p-1}{b}} \Bigg( \sum_{\substack{S,T \subseteq \{1,\ldots,p\}: \\ |S|=a,\, |T|=b,\, i \in S,\, j \notin T}} k_\xi(\mathbf{1}_S, \mathbf{1}_T) \Bigg) \\
&- \sum_{a=0}^{p-1}\sum_{b=1}^{p} \frac{1}{p^2 \binom{p-1}{a}\binom{p-1}{b-1}} \Bigg( \sum_{\substack{S,T \subseteq \{1,\ldots,p\}: \\ |S|=a,\, |T|=b,\, i \notin S,\, j \in T}} k_\xi(\mathbf{1}_S, \mathbf{1}_T) \Bigg) \\
&+ \sum_{a=0}^{p-1}\sum_{b=0}^{p-1} \frac{1}{p^2 \binom{p-1}{a}\binom{p-1}{b}} \Bigg( \sum_{\substack{S,T \subseteq \{1,\ldots,p\}: \\ |S|=a,\, |T|=b,\, i \notin S,\, j \notin T}} k_\xi(\mathbf{1}_S, \mathbf{1}_T) \Bigg).
\end{aligned}$$

Thereby, whenever $i \in S$ (first and second sum), only sizes $a \geq 1$ are possible, and whenever $i \notin S$ (third and fourth sum), only sizes $a \leq p - 1$ are possible. Similarly, whenever $j \in T$ (first and third sum), only sizes $b \geq 1$ are possible, and whenever $j \notin T$ (second and fourth sum), only sizes $b \leq p - 1$ are possible.

**Lemma 4 - Computing sums over coalition pairs of size** $(a, b)$**.** Consider the multiplicative kernel $k_\xi$ evaluated on two coalitions $S, T \subseteq \{1, \ldots, p\}$. This gives rise to four cases, where the $r$-th multiplicative term of $k_\xi(\mathbf{1}_S, \mathbf{1}_T)$ is $\alpha_r$ if $r$ is in both $S$ and $T$, or in neither of them, and $\beta_r$ if $r$ is in exactly one of the two coalitions:

$$k_\xi(\mathbf{1}_S, \mathbf{1}_T) = \prod_{\substack{r \in \{1,\ldots,p\}: \\ r \in S, \, r \in T}} \alpha_r \prod_{\substack{r \in \{1,\ldots,p\}: \\ r \in S, \, r \notin T}} \beta_r \prod_{\substack{r \in \{1,\ldots,p\}: \\ r \notin S, \, r \in T}} \beta_r \prod_{\substack{r \in \{1,\ldots,p\}: \\ r \notin S, \, r \notin T}} \alpha_r.$$

Now consider the following generating polynomial in its factorised and expanded form respectively, where in the factorised form, for each coordinate $r$, the local factor $f_r(\zeta_1, \zeta_2) := \alpha_r(1 + \zeta_1\zeta_2) + \beta_r(\zeta_1 + \zeta_2)$ corresponds to the four cases of whether $r$ is in $S$ and $T$ or not:

$$
\begin{aligned}
F(\zeta_1, \zeta_2) &= \prod_{r \in \{1,\ldots,p\}} f_r(\zeta_1, \zeta_2) \\
&= \prod_{r \in \{1,\ldots,p\}} \big(\alpha_r(1 + \zeta_1\zeta_2) + \beta_r(\zeta_1 + \zeta_2)\big) \\
&= \sum_{a=0}^{p} \sum_{b=0}^{p} G(a, b)\, \zeta_1^a \zeta_2^b.
\end{aligned}
$$

This constitutes a generating polynomial whose coefficients are scaled, generalized bivariate ESPs. For the cases where $r$ is in both $S$ and $T$, or in neither of them, both contribute the same multiplicative term $\alpha_r$, and in the cases where $r$ is in exactly one of the two coalitions, both contribute the same multiplicative term $\beta_r$. This is a bivariate polynomial in $\zeta_1$ and $\zeta_2$ of degree $p$ in each variable, with coefficients $G(a, b)$ for $a, b = 0, \ldots, p$. Notably, the coefficient $G(a, b)$ for the monomial $\zeta_1^a \zeta_2^b$ corresponds to the sum of all kernel values $k_\xi(\mathbf{1}_S, \mathbf{1}_T)$ for coalition pairs $(S, T)$ of size $(a, b)$:

$$G(a, b) = \sum_{\substack{S,T \subseteq \{1,\ldots,p\}: \\ |S|=a, \, |T|=b}} k_\xi(\mathbf{1}_S, \mathbf{1}_T).$$

*Proof.* Expanding the factorised polynomial $F(\zeta_1, \zeta_2)$ means that, for each $r = 1, \ldots, p$, one chooses exactly one of the four summands in the local factor $f_r(\zeta_1, \zeta_2)$, multiplies all chosen summands and sums over all possible choices for such monomials. Each such choice pattern is naturally encoded by a coalition pair $(S, T)$, where

$$S = \{r \in \{1, \ldots, p\} : \text{One chooses the term } \beta_r\zeta_1 \text{ or } \alpha_r\zeta_1\zeta_2 \text{ in factor } r\},$$

and

$$T = \{r \in \{1, \ldots, p\} : \text{One chooses the term } \beta_r\zeta_2 \text{ or } \alpha_r\zeta_1\zeta_2 \text{ in factor } r\}.$$

Grouping all possible monomials by their degree $(a, b)$ (i.e., by the size of the corresponding coalition pair $(S, T)$) and summing over their coefficients gives the coefficient $G(a, b)$ of $\zeta_1^a \zeta_2^b$. This is exactly the sum of all kernel values $k_\xi(\mathbf{1}_S, \mathbf{1}_T)$ for coalition pairs $(S, T)$ of size $(a, b)$, as claimed. $\square$

**Lemma 5 - Computing sums over coalition pairs of size** $(a, b)$ **(not) containing players** $i$ **and** $j$ **for** $i \neq j$**.** Now consider the sum of all kernel values $k_\xi(\mathbf{1}_S, \mathbf{1}_T)$ for coalition pairs $(S, T)$ of size $(a, b)$ that contain (or do not contain) the players $i$ and $j$ respectively. To this end, we introduce the following local factors, fixing membership of a player $r$ in $S$ and $T$ respectively:

$$
\begin{aligned}
f_r^{S\in}(\zeta_1, \zeta_2) &= \alpha_r\zeta_1\zeta_2 + \beta_r\zeta_1 = \zeta_1(\alpha_r\zeta_2 + \beta_r), \\
f_r^{S\notin}(\zeta_1, \zeta_2) &= \alpha_r + \beta_r\zeta_2, \\
f_r^{T\in}(\zeta_1, \zeta_2) &= \alpha_r\zeta_1\zeta_2 + \beta_r\zeta_2 = \zeta_2(\alpha_r\zeta_1 + \beta_r), \\
f_r^{T\notin}(\zeta_1, \zeta_2) &= \alpha_r + \beta_r\zeta_1.
\end{aligned}
$$

Specifically, the first local factor $f_r^{S\in}(\zeta_1, \zeta_2)$ corresponds to the case that $r$ is in $S$. This fixes $\zeta_1$ as a multiplicative term, and the remaining term reflects the two cases of whether $r$ is in $T$ or not. In the former case $\zeta_1$ is multiplied by $\alpha_r\zeta_2$, as $r$ is

contained in both $S$ and $T$, while in the latter case $\zeta_1$ is multiplied by $\beta_r$, as $r$ is contained in $S$ but not in $T$. The remaining local factors are defined analogously for the cases that $r$ is not in $S$, $r$ is in $T$, and $r$ is not in $T$.

Now, consider the following generating polynomial in its factorised and expanded form respectively, where $u, v \in \{0, 1\}$ indicate for the players $i$ and $j$ whether $i \in S$ and $j \in T$ respectively or not, and we assume the case $i \neq j$:

$$
\begin{aligned}
F_{i,j}^{(u,v)}(\zeta_1, \zeta_2) = {} & \prod_{r \in \{1,\ldots,p\} \setminus \{i,j\}} f_r(\zeta_1, \zeta_2) \\
& \cdot \left( u\, f_i^{S\in}(\zeta_1, \zeta_2) + (1-u)\, f_i^{S\notin}(\zeta_1, \zeta_2) \right) \\
& \cdot \left( v\, f_j^{T\in}(\zeta_1, \zeta_2) + (1-v)\, f_j^{T\notin}(\zeta_1, \zeta_2) \right) \\
= {} & \sum_{a=u}^{(p+u-1)} \sum_{b=v}^{(p+v-1)} G_{i,j}^{(u,v)}(a, b)\, \zeta_1^a \zeta_2^b.
\end{aligned}
$$

In the factorised form, for each player $r \in \{1, \ldots, p\} \setminus \{i, j\}$, the local factor $f_r(\zeta_1, \zeta_2)$ corresponds to the four cases of whether $r$ is in $S$ and $T$ or not, as introduced in Lemma 4. For the players $i$ and $j$, the local factors are defined according to fixed membership of $i \in S$ and $j \in T$ respectively, as indicated by the variables $u, v$. This is a bivariate polynomial in $\zeta_1$ and $\zeta_2$ of degree $p$ in each variable, with coefficients $G_{i,j}^{(u,v)}(a, b)$ for $a = u, \ldots, (p+u-1)$ and $b = v, \ldots, (p+v-1)$. Notably, the coefficient $G_{i,j}^{(u,v)}(a, b)$ for the monomial $\zeta_1^a \zeta_2^b$ corresponds to the sum of all kernel values $k_\xi(\mathbf{1}_S, \mathbf{1}_T)$ for coalition pairs $(S, T)$ of size $(a, b)$ that contain (or do not contain) the players $i$ and $j$ respectively according to $u$ and $v$:

$$
G_{i,j}^{(u,v)}(a, b) = \sum_{\substack{S,T \subseteq \{1,\ldots,p\}: \\ |S|=a,\ |T|=b}} [i \in S]^u \, [i \notin S]^{1-u} \, [j \in T]^v \, [j \notin T]^{1-v} \, k_\xi(\mathbf{1}_S, \mathbf{1}_T).
$$

*Proof.* This follows trivially from the same reasoning as in the previous lemmas. □

**Lemma 6 - Computing sums over coalition pairs of size** $(a, b)$ **(not) containing players** $i$ **and** $j$ **for** $i = j$. Consider the case that $i = j$. This gives rise to the following generating polynomial in its factorised and expanded form respectively, where the indicator variables $u, v \in \{0, 1\}$ indicate whether $i \in S$ and $i \in T$ respectively or not:

$$
\begin{aligned}
F_{i,i}^{(u,v)}(\zeta_1, \zeta_2) = {} & \prod_{r \in \{1,\ldots,p\} \setminus \{i\}} f_r(\zeta_1, \zeta_2) \\
& \cdot \Big( [uv]\, \alpha_i \zeta_1 \zeta_2 + [u(1-v)]\, \beta_i \zeta_1 + [(1-u)v]\, \beta_i \zeta_2 + [(1-u)(1-v)]\, \alpha_i \Big) \\
= {} & \sum_{a=u}^{(p+u-1)} \sum_{b=v}^{(p+v-1)} G_{i,i}^{(u,v)}(a, b)\, \zeta_1^a \zeta_2^b.
\end{aligned}
$$

This is constructed similarly to the previous case where $i \neq j$ (Lemma 5), but now the local factor for player $i$ corresponds to the four cases of whether $i$ is in $S$ and $T$ or not, as indicated by the variables $u, v$. As opposed to the previous case, for each value of $(u, v)$, this only gives rise to one possible local factor for player $i$. In contrast, in the case $i \neq j$, each value of $(u, v)$ only dictates whether $i \in S$ and whether $j \in T$, but not whether $i \in T$ and whether $j \in S$, thus giving rise to four possible local factors.

Notably, the coefficient $G_{i,i}^{(u,v)}(a, b)$ for the monomial $\zeta_1^a \zeta_2^b$ corresponds to the sum of all kernel values $k_\xi(\mathbf{1}_S, \mathbf{1}_T)$ for coalition pairs $(S, T)$ of size $(a, b)$ that contain (or do not contain) the player $i$ according to $u$ and $v$:

$$
G_{i,i}^{(u,v)}(a, b) = \sum_{\substack{S,T \subseteq \{1,\ldots,p\}: \\ |S|=a,\ |T|=b}} [i \in S]^u \, [i \notin S]^{1-u} \, [i \in T]^v \, [i \notin T]^{1-v} \, k_\xi(\mathbf{1}_S, \mathbf{1}_T).
$$

*Proof.* This follows trivially from the same reasoning as in the previous lemmas. □

**Combining the results.**   Combining the results from the previous lemmas, we obtain the following identity for the $(i, j)$ entry of $\mathbf{A} K_\xi(\mathbf{Z}, \mathbf{Z}) \mathbf{A}^\top$:

$$
\begin{aligned}
(\mathbf{A} K_\xi(\mathbf{Z}, \mathbf{Z}) \mathbf{A}^\top)_{i,j} = & \sum_{a=1}^{p} \sum_{b=1}^{p} \frac{1}{p^2 \binom{p-1}{a-1} \binom{p-1}{b-1}} \left( G_{i,j}^{(1,1)}(a, b) \right) \\
& - \sum_{a=1}^{p} \sum_{b=0}^{p-1} \frac{1}{p^2 \binom{p-1}{a-1} \binom{p-1}{b}} \left( G_{i,j}^{(1,0)}(a, b) \right) \\
& - \sum_{a=0}^{p-1} \sum_{b=1}^{p} \frac{1}{p^2 \binom{p-1}{a} \binom{p-1}{b-1}} \left( G_{i,j}^{(0,1)}(a, b) \right) \\
& + \sum_{a=0}^{p-1} \sum_{b=0}^{p-1} \frac{1}{p^2 \binom{p-1}{a} \binom{p-1}{b}} \left( G_{i,j}^{(0,0)}(a, b) \right).
\end{aligned}
$$

**Implementation.**   In the following, we will propose an efficient implementation for computing $\mathbf{A} K_\xi(\mathbf{Z}, \mathbf{Z}) \mathbf{A}^\top$. To this end, we will first show that each $(i, j)$ entry can be reformulated as a weighted sum over coefficients of the polynomial containing all remaining factors except for the players $i$ and $j$, and then identify groups of additive terms in those weighted sums with weighted bilinear forms in these coefficient tables. We will then identify the coefficients of this polynomial based on prefix and suffix tables, and extend this to identify the bilinear forms with contractions of the prefix and suffix tables. We then show that these contracted prefix and suffix tables can be efficiently updated between player pairs, which is exploited to propose an algorithm to efficiently compute the entire $\mathbf{A} K_\xi(\mathbf{Z}, \mathbf{Z}) \mathbf{A}^\top$.

*Formulating the $(i, j)$ entry as a weighted sum over coefficients of the polynomial containing the remaining factors:* Consider the polynomial containing all local factors except for the players $i$ and $j$:

$$
F_{\setminus \{i,j\}}(\zeta_1, \zeta_2) = \prod_{r \in \{1, \ldots, p\} \setminus \{i,j\}} f_r(\zeta_1, \zeta_2) = \sum_{a=0}^{p-2} \sum_{b=0}^{p-2} G_{\setminus \{i,j\}}(a, b) \, \zeta_1^a \zeta_2^b.
$$

We collect the coefficients of this polynomial in a coefficient table $G_{\setminus \{i,j\}} \in \mathbb{R}^{p-1 \times p-1}$, with the $(a, b)$-th entry of the matrix given by $G_{\setminus \{i,j\}}(a, b)$. For the case where $i = j$, this amounts to the following polynomial:

$$
F_{\setminus \{i\}}(\zeta_1, \zeta_2) = \prod_{r \in \{1, \ldots, p\} \setminus \{i\}} f_r(\zeta_1, \zeta_2) = \sum_{a=0}^{p-1} \sum_{b=0}^{p-1} G_{\setminus \{i\}}(a, b) \, \zeta_1^a \zeta_2^b,
$$

with the coefficients collected in the table $G_{\setminus \{i\}} \in \mathbb{R}^{p \times p}$. Given the coefficients, $G_{\setminus \{i,j\}}(a, b)$ and $G_{\setminus \{i\}}(a, b)$ respectively, the coefficients of the polynomials $F_{i,j}^{(u,v)}(\zeta_1, \zeta_2)$ for all $i, j \in \{1, \ldots, p\}$ and $u, v \in \{0, 1\}$, which are required to compute the entries of $\mathbf{A} K_\xi(\mathbf{Z}, \mathbf{Z}) \mathbf{A}^\top$, can easily be obtained as a weighted sum with shifts in the indices according to the local factors of players $i$ and $j$. For instance, for the case where $i \neq j$ and $u = 1$ and $v = 1$, the coefficient can be obtained as follows:

$$
\begin{aligned}
G_{i,j}^{(1,1)}(a, b) = & \, \alpha_i \alpha_j \, G_{\setminus \{i,j\}}(a - 2, b - 2) \\
& + \alpha_i \beta_j \, G_{\setminus \{i,j\}}(a - 1, b - 2) \\
& + \beta_i \alpha_j \, G_{\setminus \{i,j\}}(a - 2, b - 1) \\
& + \beta_i \beta_j \, G_{\setminus \{i,j\}}(a - 1, b - 1),
\end{aligned}
$$

where out-of-range indices are treated as 0. Plugging this into the formulation of the $(i, j)$ entry of $\mathbf{A} K_\xi(\mathbf{Z}, \mathbf{Z}) \mathbf{A}^\top$ from

above, it can be reformulated in the following way for the case that $i \neq j$:

$$
\begin{aligned}
(\mathbf{A} K_\xi(\mathbf{Z}, \mathbf{Z}) \mathbf{A}^\top)_{i,j} =& \sum_{a=1}^{p} \sum_{b=1}^{p} \frac{1}{p^2 \binom{p-1}{a-1} \binom{p-1}{b-1}} \Big( \alpha_i \alpha_j \, G_{\backslash\{i,j\}}(a-2, b-2) + \alpha_i \beta_j \, G_{\backslash\{i,j\}}(a-1, b-2) \\
&\qquad\qquad + \beta_i \alpha_j \, G_{\backslash\{i,j\}}(a-2, b-1) + \beta_i \beta_j \, G_{\backslash\{i,j\}}(a-1, b-1) \Big) \\
&- \sum_{a=1}^{p} \sum_{b=0}^{p-1} \frac{1}{p^2 \binom{p-1}{a-1} \binom{p-1}{b}} \Big( \alpha_i \alpha_j \, G_{\backslash\{i,j\}}(a-1, b-1) + \alpha_i \beta_j \, G_{\backslash\{i,j\}}(a-2, b-1) \\
&\qquad\qquad + \beta_i \alpha_j \, G_{\backslash\{i,j\}}(a-1, b) + \beta_i \beta_j \, G_{\backslash\{i,j\}}(a-2, b) \Big) \\
&- \sum_{a=0}^{p-1} \sum_{b=1}^{p} \frac{1}{p^2 \binom{p-1}{a} \binom{p-1}{b-1}} \Big( \alpha_i \alpha_j \, G_{\backslash\{i,j\}}(a-1, b-1) + \alpha_i \beta_j \, G_{\backslash\{i,j\}}(a, b-1) \\
&\qquad\qquad + \beta_i \alpha_j \, G_{\backslash\{i,j\}}(a-1, b-2) + \beta_i \beta_j \, G_{\backslash\{i,j\}}(a, b-2) \Big) \\
&+ \sum_{a=0}^{p-1} \sum_{b=0}^{p-1} \frac{1}{p^2 \binom{p-1}{a} \binom{p-1}{b}} \Big( \alpha_i \alpha_j \, G_{\backslash\{i,j\}}(a, b) + \alpha_i \beta_j \, G_{\backslash\{i,j\}}(a-1, b) \\
&\qquad\qquad + \beta_i \alpha_j \, G_{\backslash\{i,j\}}(a, b-1) + \beta_i \beta_j \, G_{\backslash\{i,j\}}(a-1, b-1) \Big)
\end{aligned}
$$

$$
\begin{aligned}
=& \; \alpha_i \alpha_j \Big( \sum_{a=1}^{p} \sum_{b=1}^{p} \frac{1}{p^2 \binom{p-1}{a-1} \binom{p-1}{b-1}} G_{\backslash\{i,j\}}(a-2, b-2) - \sum_{a=1}^{p} \sum_{b=0}^{p-1} \frac{1}{p^2 \binom{p-1}{a-1} \binom{p-1}{b}} G_{\backslash\{i,j\}}(a-1, b-1) \\
&\qquad - \sum_{a=0}^{p-1} \sum_{b=1}^{p} \frac{1}{p^2 \binom{p-1}{a} \binom{p-1}{b-1}} G_{\backslash\{i,j\}}(a-1, b-1) + \sum_{a=0}^{p-1} \sum_{b=0}^{p-1} \frac{1}{p^2 \binom{p-1}{a} \binom{p-1}{b}} G_{\backslash\{i,j\}}(a, b) \Big) \\
&+ \alpha_i \beta_j \Big( \sum_{a=1}^{p} \sum_{b=1}^{p} \frac{1}{p^2 \binom{p-1}{a-1} \binom{p-1}{b-1}} G_{\backslash\{i,j\}}(a-1, b-2) - \sum_{a=1}^{p} \sum_{b=0}^{p-1} \frac{1}{p^2 \binom{p-1}{a-1} \binom{p-1}{b}} G_{\backslash\{i,j\}}(a-2, b-1) \\
&\qquad - \sum_{a=0}^{p-1} \sum_{b=1}^{p} \frac{1}{p^2 \binom{p-1}{a} \binom{p-1}{b-1}} G_{\backslash\{i,j\}}(a, b-1) + \sum_{a=0}^{p-1} \sum_{b=0}^{p-1} \frac{1}{p^2 \binom{p-1}{a} \binom{p-1}{b}} G_{\backslash\{i,j\}}(a-1, b) \Big) \\
&+ \beta_i \alpha_j \Big( \sum_{a=1}^{p} \sum_{b=1}^{p} \frac{1}{p^2 \binom{p-1}{a-1} \binom{p-1}{b-1}} G_{\backslash\{i,j\}}(a-2, b-1) - \sum_{a=1}^{p} \sum_{b=0}^{p-1} \frac{1}{p^2 \binom{p-1}{a-1} \binom{p-1}{b}} G_{\backslash\{i,j\}}(a-1, b) \\
&\qquad - \sum_{a=0}^{p-1} \sum_{b=1}^{p} \frac{1}{p^2 \binom{p-1}{a} \binom{p-1}{b-1}} G_{\backslash\{i,j\}}(a-1, b-2) + \sum_{a=0}^{p-1} \sum_{b=0}^{p-1} \frac{1}{p^2 \binom{p-1}{a} \binom{p-1}{b}} G_{\backslash\{i,j\}}(a, b-1) \Big) \\
&+ \beta_i \beta_j \Big( \sum_{a=1}^{p} \sum_{b=1}^{p} \frac{1}{p^2 \binom{p-1}{a-1} \binom{p-1}{b-1}} G_{\backslash\{i,j\}}(a-1, b-1) - \sum_{a=1}^{p} \sum_{b=0}^{p-1} \frac{1}{p^2 \binom{p-1}{a-1} \binom{p-1}{b}} G_{\backslash\{i,j\}}(a-2, b) \\
&\qquad - \sum_{a=0}^{p-1} \sum_{b=1}^{p} \frac{1}{p^2 \binom{p-1}{a} \binom{p-1}{b-1}} G_{\backslash\{i,j\}}(a, b-2) + \sum_{a=0}^{p-1} \sum_{b=0}^{p-1} \frac{1}{p^2 \binom{p-1}{a} \binom{p-1}{b}} G_{\backslash\{i,j\}}(a-1, b-1) \Big).
\end{aligned}
$$

This reveals that the $(i, j)$ entry can be obtained as a weighted sum over the coefficient table $G_{\backslash\{i,j\}}$ containing 16 major additive terms. For the case where $i = j$, a similar formulation can be obtained as a weighted sum over the coefficients collected in $G_{\backslash\{i\}}$ containing 4 major additive terms:

$$
\begin{aligned}
(\mathbf{A} K_\xi(\mathbf{Z}, \mathbf{Z}) \mathbf{A}^\top)_{i,i} =& \; \alpha_i \Big( \sum_{a=1}^{p} \sum_{b=1}^{p} \frac{1}{p^2 \binom{p-1}{a-1} \binom{p-1}{b-1}} G_{\backslash\{i\}}(a-1, b-1) \Big) - \beta_i \Big( \sum_{a=1}^{p} \sum_{b=0}^{p-1} \frac{1}{p^2 \binom{p-1}{a-1} \binom{p-1}{b}} G_{\backslash\{i\}}(a-1, b) \Big) \\
&- \beta_i \Big( \sum_{a=0}^{p-1} \sum_{b=1}^{p} \frac{1}{p^2 \binom{p-1}{a} \binom{p-1}{b-1}} G_{\backslash\{i\}}(a, b-1) \Big) + \alpha_i \Big( \sum_{a=0}^{p-1} \sum_{b=0}^{p-1} \frac{1}{p^2 \binom{p-1}{a} \binom{p-1}{b}} G_{\backslash\{i\}}(a, b) \Big)
\end{aligned}
$$

*Identifying groups of additive terms with weighted bilinear forms:* Each of the above major additive terms corresponds to a

weighted bilinear form in $G_{\backslash\{i,j\}}$ or $G_{\backslash\{i\}}$ respectively. In particular, for the case where $i \neq j$, all 16 terms are of the form

$$\mathbf{w}_{\text{left}}^{\top}\left(G_{\backslash\{i,j\}}\right)\mathbf{w}_{\text{right}} = \sum_{a=0}^{p-1}\sum_{b=0}^{p-1}\mathbf{w}_{\text{left},a}\cdot G_{\backslash\{i,j\}}(a,b)\cdot\mathbf{w}_{\text{right},b}$$

$$= \sum_{a=0}^{p-1}\sum_{b=0}^{p-1}\mathbf{w}_{\text{left}}(a)\cdot G_{\backslash\{i,j\}}(a,b)\cdot\mathbf{w}_{\text{right}}(b),$$

with $\mathbf{w}_{\text{left}} \in \mathbb{R}^p$ and $\mathbf{w}_{\text{right}} \in \mathbb{R}^p$ as the corresponding weight vectors, which can be obtained for each of the 16 terms by collecting the $p$ associated weights, appending zeros for out-of-range indices and reindexing the sum accordingly. For instance, for the first term in the part multiplied by $\alpha_i\alpha_j$, this bilinear form can be obtained as follows:

$$\sum_{a=1}^{p}\sum_{b=1}^{p}\frac{1}{p^2\binom{p-1}{a-1}\binom{p-1}{b-1}}G_{\backslash\{i,j\}}(a-2,b-2) = \sum_{a=0}^{p-1}\sum_{b=0}^{p-1}\frac{1}{p\binom{p-1}{a+1}}G_{\backslash\{i,j\}}(a,b)\frac{1}{p\binom{p-1}{b+1}},$$

where for ease of notation, we adopt the convention that $\binom{n}{k} = 0$ for $k > n$. Similarly, for the case where $i = j$, all 4 major additive terms can be identified as weighted bilinear forms in $G_{\backslash\{i\}}$.

*Identifying the polynomial coefficients based on prefix and suffix tables:* In order to obtain the coefficient tables $G_{\backslash\{i,j\}}$ for pairs $(i,j)$, we define for all $i \in \{1, \ldots, p\}$ and $j \in \{0, 1, \ldots, p\}$ the prefix and suffix polynomials containing all local factors except for $i$ and $j$, but before and after $j$ respectively:

$$Pr'_{\backslash\{i,j\}}(\zeta_1,\zeta_2) = \prod_{r\in\{1,\ldots,j\}\backslash\{i,j\}}f_r(\zeta_1,\zeta_2) = \sum_{a=0}^{p-1}\sum_{b=0}^{p-1}Pr_{\backslash\{i,j\}}(a,b)\,\zeta_1^a\zeta_2^b,$$

$$Su'_{\backslash\{i,j\}}(\zeta_1,\zeta_2) = \prod_{r\in\{j,\ldots,p\}\backslash\{i,j\}}f_r(\zeta_1,\zeta_2) = \sum_{a=0}^{p-1}\sum_{b=0}^{p-1}Su_{\backslash\{i,j\}}(a,b)\,\zeta_1^a\zeta_2^b.$$

This yields the associated coefficient tables $Pr_{\backslash\{i,j\}} \in \mathbb{R}^{p\times p}$ and $Su_{\backslash\{i,j\}} \in \mathbb{R}^{p\times p}$. Then, using the convention that $Pr'_{\backslash\{i,1\}}(\zeta_1,\zeta_2) = Su'_{\backslash\{i,p\}}(\zeta_1,\zeta_2) = 1$ for all $i \in \{1, \ldots, p\}$, the coefficient table $G_{\backslash\{i,j\}}$ can be obtained as the convolution of the two tables $Pr_{\backslash\{i,j\}}$ and $Su_{\backslash\{i,j\}}$. In particular, the $(a,b)$-th entry of $G_{\backslash\{i,j\}}$ is defined as follows:

$$G_{\backslash\{i,j\}}(a,b) = \sum_{\substack{a_1,a_2:\\a_1+a_2=a}}\sum_{\substack{b_1,b_2:\\b_1+b_2=b}}Pr_{\backslash\{i,j\}}(a_1,b_1)\,Su_{\backslash\{i,j\}}(a_2,b_2).$$

For the case where $i = j$, the coefficient table $G_{\backslash\{i\}}$ can trivially be obtained as $Su_{\backslash\{i,0\}}$.

*Identifying the bilinear forms based on contracted prefix and suffix tables:* Plugging this expression for the coefficient table $G_{\backslash\{i,j\}}$ in the case where $i \neq j$ into the bilinear form expression derived above, it can be reformulated as follows:

$$\mathbf{w}_{\text{left}}^{\top}\left(G_{\backslash\{i,j\}}\right)\mathbf{w}_{\text{right}} = \sum_{a=0}^{p-1}\sum_{b=0}^{p-1}\mathbf{w}_{\text{left}}(a)\cdot\mathbf{w}_{\text{right}}(b)\cdot\Big(\sum_{\substack{a_1,a_2:\\a_1+a_2=a}}\sum_{\substack{b_1,b_2:\\b_1+b_2=b}}Pr_{\backslash\{i,j\}}(a_1,b_1)\cdot Su_{\backslash\{i,j\}}(a_2,b_2)\Big)$$

$$= \sum_{a_1=0}^{p-1}\sum_{a_2=0}^{p-1}\sum_{b_1=0}^{p-1}\sum_{b_2=0}^{p-1}\Big([a_1+a_2\leq p-1]\cdot[b_1+b_2\leq p-1]$$

$$\cdot\,\mathbf{w}_{\text{left}}(a_1+a_2)\cdot\mathbf{w}_{\text{right}}(b_1+b_2)\cdot Pr_{\backslash\{i,j\}}(a_1,b_1)\cdot Su_{\backslash\{i,j\}}(a_2,b_2)\Big)$$

$$= \sum_{a_2=0}^{p-1}\sum_{b_1=0}^{p-1}\Big(\sum_{a_1=0}^{p-1-a_2}\mathbf{w}_{\text{left}}(a_1+a_2)\cdot Pr_{\backslash\{i,j\}}(a_1,b_1)\Big)\cdot\Big(\sum_{b_2=0}^{p-1-b_1}\mathbf{w}_{\text{right}}(b_1+b_2)\cdot Su_{\backslash\{i,j\}}(a_2,b_2)\Big)$$

$$= \sum_{a_2=0}^{p-1}\sum_{b_1=0}^{p-1}Pr_{\text{contr.},\backslash\{i,j\}}(a_2,b_1)\cdot Su_{\text{contr.},\backslash\{i,j\}}(a_2,b_1).$$

Here we denote with $Pr_{\text{contr.},\backslash\{i,j\}}(a_2, b_1) \in \mathbb{R}$ and $Su_{\text{contr.},\backslash\{i,j\}}(a_2, b_1) \in \mathbb{R}$ the prefix and suffix entries contracted with the weight vectors $\mathbf{w}_{\text{left}}$ and $\mathbf{w}_{\text{right}}$ respectively. These are collected across values of $a_2$ and $b_1$ in the contracted coefficient tables $Pr_{\text{contr.},\backslash\{i,j\}} \in \mathbb{R}^{p \times p}$ and $Su_{\text{contr.},\backslash\{i,j\}} \in \mathbb{R}^{p \times p}$. This shows that the bilinear forms can be computed by element-wise multiplying the contracted prefix and suffix tables and summing over the entries of the resulting table. Note that the contracted prefix and suffix tables can be computed independently from each other, as they are based on different sets of local factors and weight vectors. In particular, all 16 distinct bilinear forms can be computed based on four unique contracted prefix and four unique contracted suffix tables.

For the case where $i = j$, the 4 unique bilinear forms can be obtained as follows:

$$\mathbf{w}_{\text{left}}^{\top} \left(G_{\backslash\{i\}}\right) \mathbf{w}_{\text{right}} = \sum_{a=0}^{p-1} \sum_{b=0}^{p-1} \mathbf{w}_{\text{left}}(a) \cdot \mathbf{w}_{\text{right}}(b) \cdot \left( \sum_{\substack{a_1, a_2: \\ a_1 + a_2 = a}} \sum_{\substack{b_1, b_2: \\ b_1 + b_2 = b}} Pr_{\backslash\{i,1\}}(a_1, b_1) \cdot Su_{\backslash\{i,0\}}(a_2, b_2) \right)$$

$$= \sum_{a_2=0}^{p-1} \sum_{b_1=0}^{p-1} Pr_{\text{contr.},\backslash\{i,1\}}(a_2, b_1) \cdot Su_{\text{contr.},\backslash\{i,0\}}(a_2, b_1).$$

This follows trivially as $Su_{\backslash\{i,0\}}$ is equivalent to its convolution with $Pr_{\backslash\{i,1\}}$.

*Efficiently updating the contracted prefix and suffix tables:* Furthermore, we show that the contracted prefix and suffix tables evolve as simple, linear functions of their previous and proceeding tables respectively. In particular, the entries of $Pr_{\text{contr.},\backslash\{i,j\}}$ for $j \in \{2, \dots, p\}$ can be obtained as a linear function of the entries of $Pr_{\text{contr.},\backslash\{i,j-1\}}$:

$$Pr_{\text{contr.},\backslash\{i,j\}}(a_2, b_1) = \sum_{a_1=0}^{p-1-a_2} \mathbf{w}_{\text{left}}(a_1 + a_2) \cdot Pr_{\backslash\{i,j\}}(a_1, b_1)$$

$$= \sum_{a_1=0}^{p-1-a_2} \mathbf{w}_{\text{left}}(a_1 + a_2) \cdot \Big( \alpha_{j-1} \cdot Pr_{\backslash\{i,j-1\}}(a_1, b_1)$$
$$+ \beta_{j-1} \cdot Pr_{\backslash\{i,j-1\}}(a_1 - 1, b_1)$$
$$+ \beta_{j-1} \cdot Pr_{\backslash\{i,j-1\}}(a_1, b_1 - 1)$$
$$+ \alpha_{j-1} \cdot Pr_{\backslash\{i,j-1\}}(a_1 - 1, b_1 - 1) \Big)$$

$$= \alpha_{j-1} \cdot Pr_{\text{contr.},\backslash\{i,j-1\}}(a_2, b_1)$$
$$+ \beta_{j-1} \cdot Pr_{\text{contr.},\backslash\{i,j-1\}}(a_2 + 1, b_1)$$
$$+ \beta_{j-1} \cdot Pr_{\text{contr.},\backslash\{i,j-1\}}(a_2, b_1 - 1)$$
$$+ \alpha_{j-1} \cdot Pr_{\text{contr.},\backslash\{i,j-1\}}(a_2 + 1, b_1 - 1).$$

Again, out-of-range indices are treated as 0. Similarly, the entries of $Su_{\text{contr.},\backslash\{i,j\}}$ for $j \in \{0, \dots, p-1\}$ can be obtained as a linear function of the entries of $Su_{\text{contr.},\backslash\{i,j+1\}}$:

$$Su_{\text{contr.},\backslash\{i,j\}}(a_2, b_1) = \sum_{b_2=0}^{p-1-b_1} \mathbf{w}_{\text{right}}(b_1 + b_2) \cdot Su_{\backslash\{i,j\}}(a_2, b_2)$$

$$= \sum_{b_2=0}^{p-1-b_1} \mathbf{w}_{\text{right}}(b_1 + b_2) \cdot \Big( \alpha_{j+1} \cdot Su_{\backslash\{i,j+1\}}(a_2, b_2)$$
$$+ \beta_{j+1} \cdot Su_{\backslash\{i,j+1\}}(a_2 - 1, b_2)$$
$$+ \beta_{j+1} \cdot Su_{\backslash\{i,j+1\}}(a_2, b_2 - 1)$$
$$+ \alpha_{j+1} \cdot Su_{\backslash\{i,j+1\}}(a_2 - 1, b_2 - 1) \Big)$$

$$= \alpha_{j+1} \cdot Su_{\text{contr.},\backslash\{i,j+1\}}(a_2, b_1)$$
$$+ \beta_{j+1} \cdot Su_{\text{contr.},\backslash\{i,j+1\}}(a_2 - 1, b_1)$$
$$+ \beta_{j+1} \cdot Su_{\text{contr.},\backslash\{i,j+1\}}(a_2, b_1 + 1)$$
$$+ \alpha_{j+1} \cdot Su_{\text{contr.},\backslash\{i,j+1\}}(a_2 - 1, b_1 + 1).$$

*Proposed algorithm and runtime analysis:* Based on the above findings, we propose the following algorithm for efficiently computing $\mathbf{A}K_\xi(\mathbf{Z}, \mathbf{Z})\mathbf{A}^\top$: Overall, for each $i' \in \{1, \ldots, p\}$ independently, we compute the entries of all pairs $(i', j')$ with $j' \leq i'$. Due to symmetry of $\mathbf{A}K_\xi(\mathbf{Z}, \mathbf{Z})\mathbf{A}^\top$, this gives all entries. In particular, for a certain $i'$, we conduct two phases: A backward and a forward phase. In the initial backward phase, we generate - for each of the 4 different weight vectors - a sequence of contracted suffix tables in reverse order. Specifically, after initializing $Su_{\text{contr.}, \backslash\{i', p\}}$ for each weight vector, we iteratively update the tables in backward order until we obtain $Su_{\text{contr.}, \backslash\{i', 0\}}$ across weight vectors. Thereby, we store all the intermediate tables, as they are needed in the subsequent forward phase. Afterwards, in the forward phase, we initialize the first contracted prefix table $Pr_{\text{contr.}, \backslash\{i', 1\}}$ across weight vectors and iteratively update the tables in forward order until $Pr_{\text{contr.}, \backslash\{i', i'-1\}}$. At each iteration, the current contracted prefix tables across weight vectors together with the associated contracted suffix tables from the backward phase are used to compute all 16 bilinear forms, which are then weighted and summed up to obtain the corresponding entries of $\mathbf{A}K_\xi(\mathbf{Z}, \mathbf{Z})\mathbf{A}^\top$. The diagonal entry of $\mathbf{A}K_\xi(\mathbf{Z}, \mathbf{Z})\mathbf{A}^\top$ for the current $i'$ is computed based on the $Su_{\text{contr.}, \backslash\{i', 0\}}$ and $Pr_{\text{contr.}, \backslash\{i', 1\}}$ across weight vectors. Note that in the forward phase, it is not necessary to persist the intermediate contracted prefix tables, as they are not needed for future computations.

For each of $p$ independent backward phases, where each is associated with one particular $i'$, the amount of contracted suffix tables scales as $\mathcal{O}(p)$, and the computation of each of those scales as $\mathcal{O}(p^2)$. This is as the size of the contracted suffix tables scales in $\mathcal{O}(p^2)$, and each entry can be computed as a linear function of the shifted entries of the proceeding table. Overall, this amounts to total scaling as $\mathcal{O}(p^3)$ for the backward phase for a certain $i'$. Similarly, for each of $p$ independent forward phases, where each is associated with a certain $i'$, the amount of contracted prefix table updates, based on the preceding tables, scales as $\mathcal{O}(p)$ and the cost for each update also scales as $\mathcal{O}(p^2)$. Furthermore, in each forward phase step, entries of $\mathbf{A}K_\xi(\mathbf{Z}, \mathbf{Z})\mathbf{A}^\top$ are computed based on the contracted tables by element-wise multiplication and summation, which scales as $\mathcal{O}(p^2)$. Overall, this amounts to total scaling as $\mathcal{O}(p^3)$ for the forward phase for a certain $i'$. In total, the computation of $\mathbf{A}K_\xi(\mathbf{Z}, \mathbf{Z})\mathbf{A}^\top$ thus scales as $\mathcal{O}(p^4)$.

Note that updating the non-contracted prefix and suffix coefficient tables in the forward and backward phase, as opposed to the contracted ones, would induce computational cost scaling as $\mathcal{O}(p^5)$. This is because the bilinear form computation via explicit convolution - and without efficiently updating the contracted tables - would require an additional weighted summation scaling as $\mathcal{O}(p)$.

$\square$

**Theorem B.3.** *The EIG about the SVs $\phi$ for a candidate coalition $\mathbf{z}^{(i)} \in \{0,1\}^p$ (Formula 6 in Section 3.4) is computable in $\mathcal{O}(p^4 + t^3)$.*

*Proof.* For clarity, we repeat the derived expression of the EIG about the SVs $\phi$ for a candidate coalition $\mathbf{z}^{(i)} \in \{0,1\}^p$ at iteration $t$:

$$\mathrm{EIG}_{\phi}^{(t)}(\mathbf{z}^{(i)}) \propto C' + \log\left[\mathbf{e}_i^\top \left(\mathbf{\Sigma}_{(\boldsymbol{\nu}|\mathcal{D}_t)} + \sigma_\epsilon^2 \mathbf{I}\right)\mathbf{e}_i\right]$$
$$- \log\left[\mathbf{e}_i^\top \left(\mathbf{\Sigma}_{(\boldsymbol{\nu}|\mathcal{D}_t)} + \sigma_\epsilon^2 \mathbf{I} - \mathbf{Q}\right)\mathbf{e}_i\right].$$

Here, $C'$ is constant, $\mathbf{I} \in \mathbb{R}^{2^p \times 2^p}$ is the identity matrix, and $\mathbf{Q}_{i,i} = \mathbf{e}_i^\top \mathbf{Q}\mathbf{e}_i$, the $i$-th diagonal entry of $\mathbf{Q} \in \mathbb{R}^{2^p \times 2^p}$, is defined as

$$\mathbf{Q}_{i,i} = \left(\mathbf{A}\mathbf{\Sigma}_{(\boldsymbol{\nu}|\mathcal{D}_t)}\mathbf{e}_i\right)^\top \left(\mathbf{A}\mathbf{\Sigma}_{(\boldsymbol{\nu}|\mathcal{D}_t)}\mathbf{A}^\top\right)^{-1}\left(\mathbf{A}\mathbf{\Sigma}_{(\boldsymbol{\nu}|\mathcal{D}_t)}\mathbf{e}_i\right). \tag{12}$$

The EIG depends on the marginal posterior variance $\mathbf{e}_i^\top \mathbf{\Sigma}_{(\boldsymbol{\nu}|\mathcal{D}_t)}\mathbf{e}_i = \mathrm{Var}(\nu(\mathbf{z}^{(i)}) \mid \mathcal{D}_t) \in \mathbb{R}$ and on $\mathbf{Q}_{i,i} \in \mathbb{R}$. While the computation of the former term scales as $\mathcal{O}(t^3)$ (see Subsection A.1), naive computation of the EIG is dominated by the latter, which is prohibitively expensive in many settings. We will now show how the two main components of $\mathbf{Q}_{i,i}$, namely $\mathbf{A}\mathbf{\Sigma}_{(\boldsymbol{\nu}|\mathcal{D}_t)}\mathbf{e}_i$ and $\mathbf{A}\mathbf{\Sigma}_{(\boldsymbol{\nu}|\mathcal{D}_t)}\mathbf{A}^\top$, can be computed efficiently using Theorems B.1 and B.2, and how this enables an efficient computation of $\mathbf{Q}_{i,i}$. Here, we denote by $\mathbf{X}^{(t)} \in \mathbb{R}^{(t-1)\times p}$ the matrix containing all previously evaluated coalitions at iteration $t$.

We propose initially to compute $\mathrm{Var}(\nu(\mathbf{z}^{(i)}) \mid \mathcal{D}_t)$, separately from $\mathbf{Q}_{i,i}$, as it cannot be obtained as a submatrix of the projected $\mathbf{A}\mathbf{\Sigma}_{(\boldsymbol{\nu}|\mathcal{D}_t)}$ and $\mathbf{A}\mathbf{\Sigma}_{(\boldsymbol{\nu}|\mathcal{D}_t)}\mathbf{A}^\top$. Note that this already depends on the inverse (noisy) kernel matrix of the training data $\left[K_\xi(\mathbf{X}^{(t)}, \mathbf{X}^{(t)}) + \sigma_\epsilon^2\mathbf{I}\right]^{-1} \in \mathbb{R}^{(t-1)\times(t-1)}$. Thus, the kernel matrix is already decomposed into a Cholesky factor, which takes $\mathcal{O}(t^3)$, and can be reused across further computations within the same iteration. This cost will not be repeatedly stated in the following.

**Computation of $\mathbf{A}\mathbf{\Sigma}_{(\boldsymbol{\nu}|\mathcal{D}_t)}\mathbf{e}_i$.** Consider the expanded formulation of $\mathbf{A}\mathbf{\Sigma}_{(\boldsymbol{\nu}|\mathcal{D}_t)}\mathbf{e}_i \in \mathbb{R}^p$:

$$\mathbf{A}\mathbf{\Sigma}_{(\boldsymbol{\nu}|\mathcal{D}_t)}\mathbf{e}_i = \mathbf{A}K_\xi(\mathbf{Z}, \mathbf{Z})\mathbf{e}_i - \mathbf{A}K_\xi(\mathbf{Z}, \mathbf{X}^{(t)})\left[K_\xi(\mathbf{X}^{(t)}, \mathbf{X}^{(t)}) + \sigma_\epsilon^2\mathbf{I}\right]^{-1}K_\xi(\mathbf{Z}, \mathbf{X}^{(t)})^\top \mathbf{e}_i$$
$$= \mathbf{A}K_\xi(\mathbf{Z}, \mathbf{z}^{(i)}) - \mathbf{A}K_\xi(\mathbf{Z}, \mathbf{X}^{(t)})\left[K_\xi(\mathbf{X}^{(t)}, \mathbf{X}^{(t)}) + \sigma_\epsilon^2\mathbf{I}\right]^{-1}K_\xi(\mathbf{X}^{(t)}, \mathbf{z}^{(i)}).$$

By Theorem B.1 the first term of the difference $\mathbf{A}K_\xi(\mathbf{Z}, \mathbf{z}^{(i)}) \in \mathbb{R}^p$ can be computed in $\mathcal{O}(p^2)$, and $\mathbf{A}K_\xi(\mathbf{Z}, \mathbf{X}^{(t)}) \in \mathbb{R}^{p\times(t-1)}$ from the second term can be computed in $\mathcal{O}(t \cdot p^2)$. The latter follows as the computation from Theorem B.1 can be applied to each column of the cross-kernel matrix $K_\xi(\mathbf{Z}, \mathbf{X}^{(t)}) \in \mathbb{R}^{2^p \times (t-1)}$ independently. This is admissible as for each row of $\mathbf{X}^{(t)}$, i.e. $\mathbf{X}_i^{(t)}$ for $i = 1, ..., t-1$, it holds that $\mathbf{X}_i^{(t)} \in \{0,1\}^p$. With pre-computed Cholesky factorization for the kernel matrix, the remaining components of the second term can be computed by solving a linear system, which scales as $\mathcal{O}(t^2)$, and a matrix-vector product that scales as $\mathcal{O}(t \cdot p)$. Taking all operations together, this is $\mathcal{O}(t \cdot p^2 + t^2)$.

**Computation of $\mathbf{A}\mathbf{\Sigma}_{(\boldsymbol{\nu}|\mathcal{D}_t)}\mathbf{A}^\top$.** Consider the expanded formulation of $\mathbf{A}\mathbf{\Sigma}_{(\boldsymbol{\nu}|\mathcal{D}_t)}\mathbf{A}^\top \in \mathbb{R}^{p\times p}$:

$$\mathbf{A}\mathbf{\Sigma}_{(\boldsymbol{\nu}|\mathcal{D}_t)}\mathbf{A}^\top = \mathbf{A}K_\xi(\mathbf{Z}, \mathbf{Z})\mathbf{A}^\top - \mathbf{A}K_\xi(\mathbf{Z}, \mathbf{X}^{(t)})\left[K_\xi(\mathbf{X}^{(t)}, \mathbf{X}^{(t)}) + \sigma_\epsilon^2\mathbf{I}\right]^{-1}K_\xi(\mathbf{Z}, \mathbf{X}^{(t)})^\top \mathbf{A}^\top.$$

By Theorem B.2 the first term of the difference, $\mathbf{A}K_\xi(\mathbf{Z}, \mathbf{Z})\mathbf{A}^\top \in \mathbb{R}^{p\times p}$, can be computed in $\mathcal{O}(p^4)$. $\mathbf{A}K_\xi(\mathbf{Z}, \mathbf{X}^{(t)})$ can be reused from the previous step, and thus does not require additional computations. Using the pre-computed Cholesky factorization for the kernel matrix, the remaining components of the second term can be computed by solving a linear system, which scales as $\mathcal{O}(t^2 \cdot p)$, and a matrix-matrix product that scales as $\mathcal{O}(p^2 \cdot t)$. In total, this is $\mathcal{O}(p^4 + t^2 \cdot p)$ for $\mathbf{A}\mathbf{\Sigma}_{(\boldsymbol{\nu}|\mathcal{D}_t)}\mathbf{A}^\top$.

**Computation of $\mathbf{Q}_{i,i}$.** Given the terms $\mathbf{A}\mathbf{\Sigma}_{(\boldsymbol{\nu}|\mathcal{D}_t)}\mathbf{e}_i$ and $\mathbf{A}\mathbf{\Sigma}_{(\boldsymbol{\nu}|\mathcal{D}_t)}\mathbf{A}^\top$, we propose for a stable and efficient implementation of $\mathbf{Q}_{i,i}$, applying a Cholesky decomposition to $\mathbf{A}\mathbf{\Sigma}_{(\boldsymbol{\nu}|\mathcal{D}_t)}\mathbf{A}^\top$, which scales as $\mathcal{O}(p^3)$, then solving the associated linear system with the vector $\mathbf{A}\mathbf{\Sigma}_{(\boldsymbol{\nu}|\mathcal{D}_t)}\mathbf{e}_i$, which scales as $\mathcal{O}(p^2)$, and lastly computing a dot product in $\mathcal{O}(p)$.

In summary, the EIG for a single candidate can be computed in $\mathcal{O}(p^4 + t^3)$, which is polynomial in $p$. $\qquad\square$

B.4.1. VECTORIZED COMPUTATION.

Consider the setting in which the EIG is evaluated for a set of candidate coalitions $\mathbf{W} \subseteq \{0,1\}^p$, yielding $\mathrm{EIG}_\phi^{(t)}(\mathbf{W}) := (\mathrm{EIG}_\phi^{(t)}(\mathbf{z}^{(i)}))_{\mathbf{z}^{(i)} \in \mathbf{W}} \in \mathbb{R}^{|\mathbf{W}|}$. The exhaustive evaluation over all coalitions in $\mathbf{Z}$ is recovered as the special case where $|\mathbf{W}| = 2^p$. For each candidate $\mathbf{z}^{(i)} \in \{0,1\}^p$, the EIG expression in Equation 6 consists of quadratic forms of the type $\mathbf{e}_i^\top \mathbf{M} \mathbf{e}_i$, where $\mathbf{M}$ is independent of the specific candidate index $i$; such terms simply extract the $i$-th diagonal element of the corresponding matrix. Stacking this expression over all candidate coalitions and collecting the resulting scalars therefore amounts to taking the diagonal of the respective matrices restricted to $\mathbf{W}$. This yields the following expression:

$$\mathrm{EIG}_\phi^{(t)}(\mathbf{W}) \propto C' + \log \left[ \mathrm{diag} \left( \left( \boldsymbol{\Sigma}_{(\boldsymbol{\nu}|\mathcal{D}_t)} + \sigma_\epsilon^2 \mathbf{I} \right)_{\mathbf{W},\mathbf{W}} \right) \right] \tag{13}$$
$$- \log \left[ \mathrm{diag} \left( \left( \boldsymbol{\Sigma}_{(\boldsymbol{\nu}|\mathcal{D}_t)} + \sigma_\epsilon^2 \mathbf{I} - \mathbf{Q} \right)_{\mathbf{W},\mathbf{W}} \right) \right],$$

where $(\cdot)_{\mathbf{W},\mathbf{W}}$ denotes restriction to the rows and columns indexed by candidates in $\mathbf{W}$, $\mathrm{diag}(\cdot)$ denotes the diagonal of a square matrix, and $\log(\cdot)$ represents an elementwise logarithm.

In our proposed implementation, we initially compute the marginal variances $\mathrm{diag} \left( (\boldsymbol{\Sigma}_{(\boldsymbol{\nu}|\mathcal{D}_t)})_{\mathbf{W},\mathbf{W}} \right) \in \mathbb{R}^{|\mathbf{W}|}$. This scales as $\mathcal{O}(|\mathbf{W}| \cdot t^2 + t^3)$ across candidates and already includes the cost of Cholesky-decomposing the training kernel matrix, which can thus be reused in the following and is independent of the specific candidate. All remaining operations for variance computation can be efficiently vectorized across candidates in common computational frameworks. The evaluation of $\mathrm{diag} \left( (\mathbf{Q})_{\mathbf{W},\mathbf{W}} \right) \in \mathbb{R}^{|\mathbf{W}|}$ requires the computation and Cholesky decomposition of $\mathbf{A} \boldsymbol{\Sigma}_{(\boldsymbol{\nu}|\mathcal{D}_t)} \mathbf{A}^\top$, which scales as $\mathcal{O}(p^4 + t^2 \cdot p)$ and is also independent of the specific candidate and thus reusable. Then, for each candidate, $\mathbf{A} \boldsymbol{\Sigma}_{(\boldsymbol{\nu}|\mathcal{D}_t)} \mathbf{e}_i$ must be computed, the associated triangular system of linear equations with $\mathbf{A} \boldsymbol{\Sigma}_{(\boldsymbol{\nu}|\mathcal{D}_t)} \mathbf{A}^\top$ must be solved, and a dot product must be computed. This scales as $\mathcal{O}(|\mathbf{W}| \cdot p \cdot t + p \cdot t^2)$ and can be vectorized across candidates.

Overall, the computation across candidates scales as $\mathcal{O}(p^4 + t^3 + |\mathbf{W}| \cdot t^2)$. Note that the first two additive terms, which dominate the computational cost in many settings, are associated with operations that are independent of the specific candidate and thus scale independently of $|\mathbf{W}|$. All remaining candidate-specific operations can be efficiently vectorized and scale only in $\mathcal{O}(|\mathbf{W}| \cdot t^2)$. Consequently, in many settings, vectorized EIG evaluation incurs only a manageable overhead compared to evaluating a single candidate. This even enables exhaustive EIG optimization across all candidates for small $p$.

# C. Related Work

In the following, we provide further details on the related work discussed in Section 4. In particular, we discuss the relationship to popular approaches from the transductive and prediction-oriented active learning literature.

**Information-based transductive learning.** Information-based transductive learning (ITL; MacKay, 1992) selects candidates based on the EIG for a set of target function values. In the SV setting, a direct application with the target set chosen as all coalitions corresponds to the EIG for the value function vector $\boldsymbol{\nu}$. At iteration $t$, this amounts to the following optimization problem:

$$\arg\max_{i \in \mathcal{C}} I\big(\boldsymbol{\nu}; \nu'(\mathbf{z}^{(i)}) \mid \mathcal{D}_t\big)$$
$$= \arg\max_{i \in \mathcal{C}} \mathrm{Var}(\nu'(\mathbf{z}^{(i)}) \mid \mathcal{D}_t).$$

However, maximizing this criterion collapses to purely exploratory uncertainty sampling in our setting, i.e., selecting the coalition with the highest marginal posterior variance under the GP surrogate (Krause et al., 2008; Hübotter et al., 2024). This follows from the homoscedastic, i.i.d. Gaussian noise assumption. Thus, this criterion does not explicitly account for how an evaluation reduces uncertainty at other coalitions, and hence about the SVs.

**Expected predictive information gain.** The expected predictive information gain (EPIG; Smith et al., 2023) is another popular criterion from prediction-oriented BED. A natural choice when applying it in the SV estimation setting is a uniform distribution over all coalitions as the target distribution $p_*$. At iteration $t$, this amounts to the following optimization problem:

$$\arg\max_{i \in \mathcal{C}} \mathbb{E}_{p_*(j)}\big[I\big(\nu'(\mathbf{z}^{(j)}); \nu'(\mathbf{z}^{(i)}) \mid \mathcal{D}_t\big)\big]$$
$$= \arg\max_{i \in \mathcal{C}} -\frac{1}{2^p} \sum_{j=1}^{2^p} H\big(\nu'(\mathbf{z}^{(j)}) \mid \nu'(\mathbf{z}^{(i)}), \mathcal{D}_t\big).$$

Note that the sum scales exponentially in $p$, rendering this approach prohibitively expensive in many settings. Although this expectation could in principle be approximated by Monte Carlo sampling of target coalitions, we do not pursue this direction here, since it would introduce an additional approximation and sampling-design choice for a criterion that is already not directly aligned with our final quantity of interest, the SVs.

# D. Experiments

In the following, we provide further details on the experiments reported in the main paper (Section 5).

## D.1. Experimental Setup

### D.1.1. INITIAL DESIGN

As explained in the main paper, the initial designs for ShaplEIG consist of $T_0 = p + 1$ coalitions drawn according to leverage score sampling. This size follows common practice in GP-based BED and BO, and is also a natural lower-end choice in our experiments, since several linear-model-based competitors (Kernel SHAP and Leverage SHAP) require at least $p + 1$ observations. The intuition here is to keep the initial design as small as possible, so that all remaining design points are subject to guided sequential selection. The use of leverage score sampling for the initial design is motivated by its state-of-the-art performance in recent benchmarks, while adding virtually no computational overhead.

However, we did not tune these choices and, in preliminary experiments, did not observe a strong influence of the initial design scheme on the performance of ShaplEIG. This suggests that the strong performance of ShaplEIG is not overly dependent on this particular initialization strategy.

### D.1.2. GAUSSIAN PROCESS SURROGATES

In the following, we provide further details on the GP surrogates used.

For our proposed method ShaplEIG and the other GP-based competitor variants, we use a zero-mean, unit-variance GP prior with a Hamming kernel as the covariance function (see Appendix A.1). This is consistent with standardizing the training data at each iteration. The kernel has characteristic lengthscale hyperparameters $\xi \in \mathbb{R}^p$, which are optimized via maximum a posteriori (MAP) estimation using the L-BFGS-B optimizer (Byrd et al., 1995). We use the following prior:

$$\xi \sim \text{LogNormal}\left(\mu = \sqrt{2} + 0.5 \log p, \ \sigma = \sqrt{3}\right), \tag{14}$$

which corresponds to the default setting in BoTorch (Balandat et al., 2020). We enforce a minimum value of $10^{-6}$ for each lengthscale. Hyperparameters are optimized at each iteration using random initialization and restarts in case of failed optimization runs. For numerical stability, we assume additive zero-mean Gaussian noise with fixed variance $10^{-6}$. This is the smallest value supported by BoTorch and effectively yields quasi-noiseless GPs.

### D.1.3. GAMES

In the following, we provide further details on the games considered in our experiments.

**Feature importance.** We consider global FI for the TabPFN-2.5 foundation model (Hollmann et al., 2025; Grinsztajn et al., 2025). As TabPFN relies on in-context learning, the value function for a feature coalition is defined as the performance (MSE or accuracy) on an inference set after removing absent features from both the training and inference data during a forward pass (Rundel et al., 2024). We use three datasets: Diabetes regression (Efron et al., 2004), Diabetes classification (Smith et al., 1988), and Breast Cancer (Wolberg & Mangasarian, 1990). We obtained the first two datasets from OpenML (Bischl et al., 2025) and the last from scikit-learn (Pedregosa et al., 2011; Buitinck et al., 2013). We rely on precomputed value function evaluations and provide the scripts for reproducing those in the code repository. The seeds influence the random splitting of the data into training and inference sets, which is done with a 70/30 ratio, and are also used for the TabPFN model call.

**Data valuation.** For DV (Jia et al., 2019; Ghorbani & Zou, 2019; Tay et al., 2022), the achieved test-set performance of the Random Forest (RF; Breiman, 2001) or Gradient Boosting (GB; Friedman, 2001) algorithm on the Bike Sharing (BS; Fanaee-T & Gama, 2014) or California Housing (CH; Pace & Barry, 1997) dataset serves as the payoff to be attributed across subsets of training data as players. The games are precomputed and taken from the shapiq library. Here, the seeds influence the random splitting of the data into training and inference sets and the learning algorithm. Further details can be found in the accompanying paper (Muschalik et al., 2024).

**Hyperparameter importance.** For HPI, we implement an ablation game following HyperSHAP (Wever et al., 2026, Section 5). Here, the value of a coalition of hyperparameters is defined as the performance obtained by setting all

hyperparameters in the coalition to the values of a configuration of interest (e.g., an optimal configuration), while fixing all remaining hyperparameters to a reference configuration (e.g., a default configuration). We report experiments for rbv2_xgboost (Binder et al., 2020) on the Chess (Shapiro, 1983; ID: 3) and Thyroid Disease (Quinlan, 1986; ID: 38) tasks and for LCBench (Zimmer et al., 2021) on the Jasmine task (Guyon et al., 2019; ID: 41143). All IDs refer to `openml.org` (Bischl et al., 2025). Cheap-to-evaluate surrogate models for the relationship between hyperparameter configurations and performance metrics provided by Yahpo-Gym (Pfisterer et al., 2022) are used for value function evaluations. The compared configurations are determined by the seed used.

**Local explanation.** Based on Kolpaczki et al. (2024) and `shapiq` (Muschalik et al., 2024), we use an LE game (Štrumbelj & Kononenko, 2010) to explain predictions for individual images from the ImageNet dataset (Deng et al., 2009). In this cooperative game, the players correspond to image components, defined as superpixels for the ResNet (He et al., 2016) model and patches for the vision transformer model (Dosovitskiy et al., 2021), and the value of a coalition is the predicted score for the target class when only the components of the coalition are retained in a model call, while all other components are replaced by a reference value (i.e., greyed out). Depending on the model size and accessibility, the value function evaluations may range from inexpensive forward passes with full model access to costly black-box queries via an inference API. Here, we again rely on precomputed games taken from the `shapiq` library.

Furthermore, we benchmark local explanations to attribute RF predictions for single test instances to features as players using the linear TreeSHAP algorithm (Bifet et al., 2022). This algorithm allows efficient and exact computation of ground-truth SVs for tree-based models in linear as opposed to exponential time, and thus enables benchmarking in the context of large games where exhaustive enumeration of all coalitions is infeasible. Here, we use the tabular CorrGroups60 (Lundberg & Lee, 2017), NHANES (Dinh et al., 2019), and Communities and Crime (Crime; Redmond, 2011) datasets with up to $p = 101$ features, provided by the `shap` package (Lundberg & Lee, 2017). The seeds determine the train-test splits of the data, where all except for a single test instance are used for training, and also influence the training procedure of the RF model. Due to the low computational cost of value function evaluations and the large number of players, we do not rely on precomputed games for this benchmark, but instead evaluate the games online.

### D.1.4. SCALABILITY

In the presented experiments, for games with $p > 16$, we do not refit the GP hyperparameters in every iteration, but instead follow a fixed refitting schedule. Specifically, we refit the hyperparameters in every iteration for the first 64 iterations, in every 8th iteration for the next 128 iterations, in every 16th iteration for the next 256 iterations, and in every 32nd iteration thereafter. In iterations without hyperparameter refitting, the EIG for all remaining candidates is computed using the GP posterior conditioned on all previously evaluated coalitions, while keeping the hyperparameters fixed at their most recent refitted values. In these non-refitting iterations, several quantities required for EIG computation can be updated efficiently: compared with the previous iteration, only one row and column need to be added to the training kernel matrix $K_\xi(\mathbf{X}^{(t)}, \mathbf{X}^{(t)})$, $\mathbf{A}K_\xi(\mathbf{Z}, \mathbf{X}^{(t)})$ only requires adding a single column, and $\mathbf{A}K_\xi(\mathbf{Z}, \mathbf{Z})\mathbf{A}^\top$ can be reused entirely.

### D.1.5. REPRODUCIBILITY

The code is publicly available at `https://github.com/slds-lmu/shapleig`. Please see the `README.md` file for instructions on reproducing the experimental results and on generating the precomputed games for TabPFN. All experiments were run on a CPU instance with 32 cores and 64 GB of RAM. The experiments were conducted using Python 3.11.13, Torch 2.9.1, GPyTorch 1.14, BoTorch 0.14.0, and shapiq 1.4.1.

## D.2. Experimental Results

In the following, we provide more detailed results for the experiments presented in the main paper (Section 5).

### D.2.1. ABLATIONS

We present the detailed results from the ablation study in Figure 3. For the sake of completeness, we also provide the experimental results with all baselines from SV approximation and the ablation in a single plot in Figure 4.

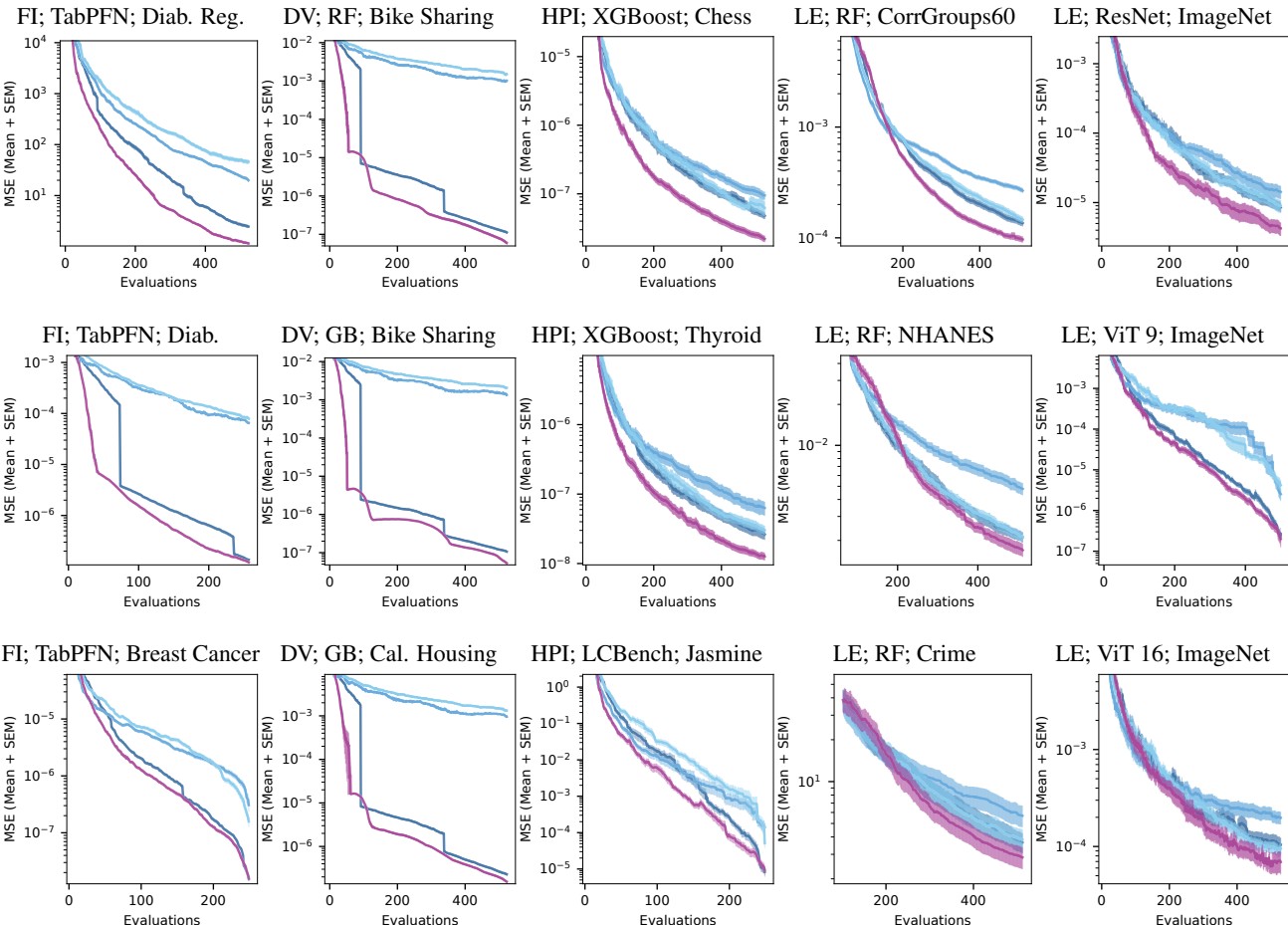

*Figure 3.* Mean squared error (MSE) between estimated and ground-truth Shapley values across all tasks and evaluation budgets, averaged over repetitions for `ShaplEIG` and the ablation baselines, with standard error of the mean (SEM) indicated.

`ShaplEIG` (Ours)    GP + Leverage Score Sampling    GP + US    GP + Random

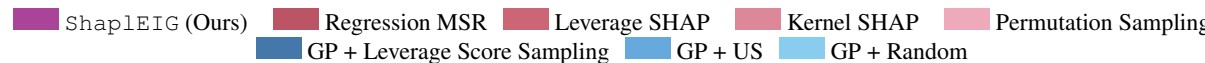

*Figure 4.* Mean squared error (MSE) between estimated and ground-truth Shapley values across all tasks and evaluation budgets, averaged over repetitions for `ShaplEIG` and the SV approximation and ablation baselines, with standard error of the mean (SEM) indicated.

D.2.2. COMPUTATIONAL COST

In the following, we present plots showing the computational cost (in seconds) of GP hyperparameter optimization and vectorized EIG evaluation for `ShaplEIG`, across all tasks and evaluation budgets.

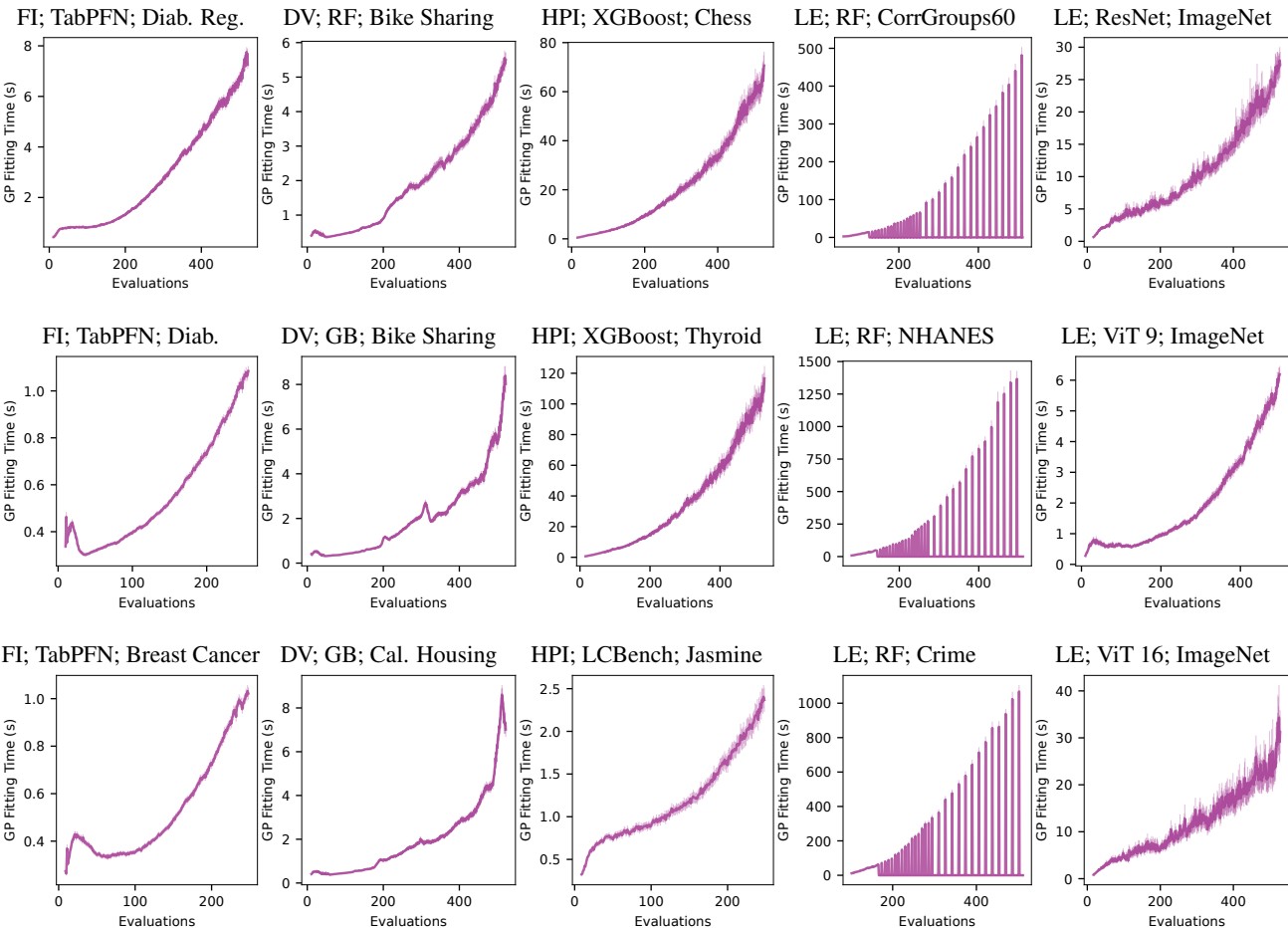

*Figure 5.* Computational cost (in seconds) of GP hyperparameter fitting across all tasks and evaluation budgets for `ShaplEIG`, averaged over repetitions and with standard error of the mean (SEM) indicated.

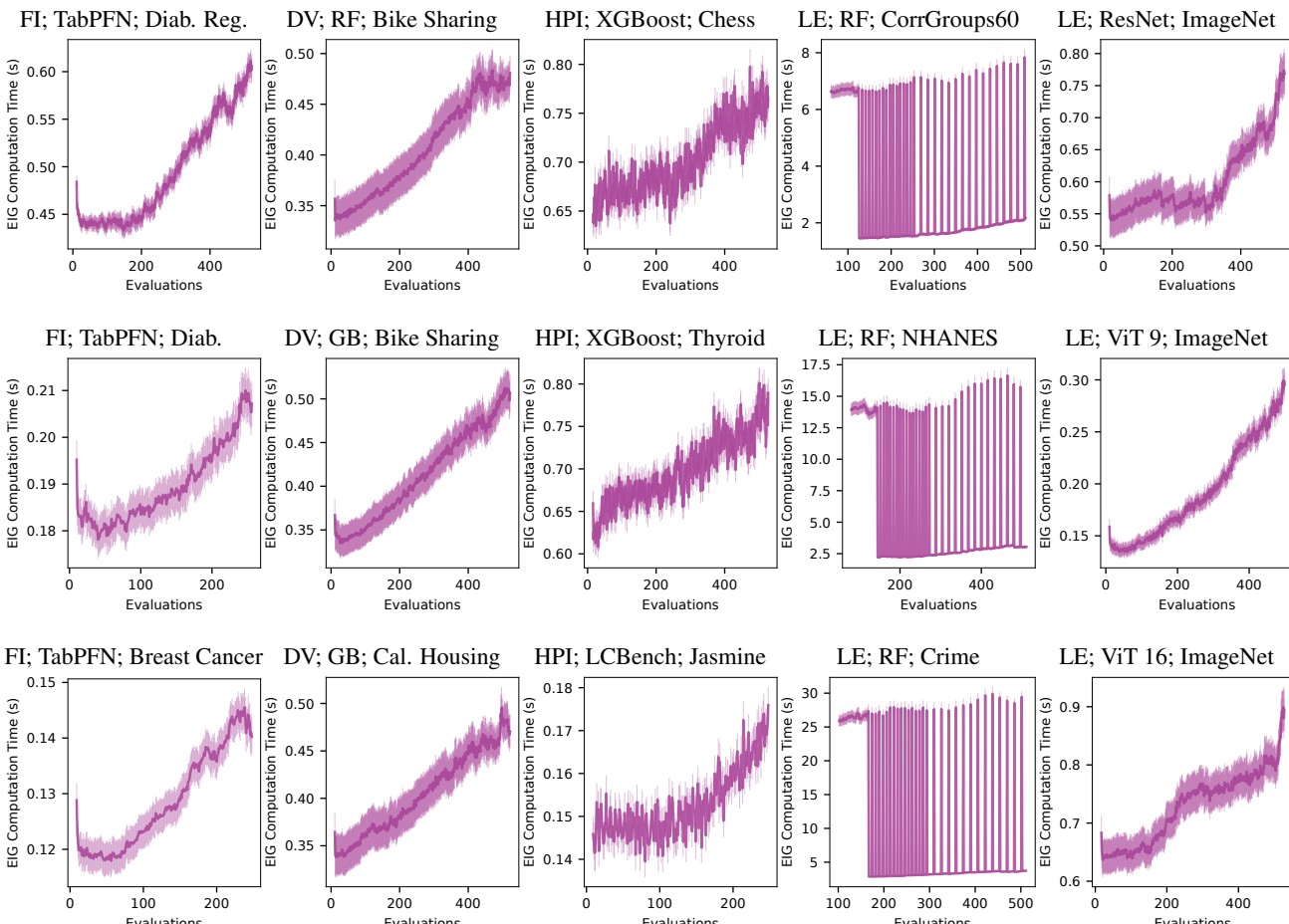

*Figure 6.* Computational cost (in seconds) of vectorized EIG evaluation across all tasks and evaluation budgets for `ShaplEIG`, averaged over repetitions and with the standard error of the mean (SEM) indicated.

