# OpenReview forum: "$\texttt{ShaplEIG}$: Bayesian Experimental Design for Shapley Value Estimation"
_ICML.cc/2026/Conference — ICML 2026 regular_

### Official Review · Reviewer_2sbH · 2026-02-24

**Soundness:** 3
**Presentation:** 3
**Significance:** 3
**Originality:** 3
**Overall Recommendation:** 4
**Confidence:** 3

**Summary:**

This paper proposes Bayesian Adaptive Experimental Design to improve Shapley Value estimation. The core ideas are to learn a Gaussian Process surrogate with a Hamming distance kernel to model the value function and to adaptively select the next coalition to evaluate in each iteration by refitting the kernel hyper parameters and maximising the Expected Information Gain, which can be computed in closed form as the Shapley value is a linear transformation of the value function. The paper proposes the efficient computation steps (and analyze the complexity) and empirically verified for up to 20 players games that their method leads to lower mean squared error of Shapley value estimates.

**Compliance With Llm Reviewing Policy:**

Affirmed.

**Final Justification:**

The authors' rebuttal has addressed the scalability concerns and better justified the surrogate model choice, thus I have updated the evaluation accordingly.

**Key Questions For Authors:**

1. How does the total computation cost and number of iterations scale with the number of players? In one iteration, what is the total complexity to evaluate the EIG for all candidates? Can the complexity of the baselines and other methods be reported?
2. What are the challenges associated with more players for the proposed method?
3. Can the authors further explain the ablation baselines a) random coalition sampling and b) leverage score sampling? Do they both use the coalition–evaluation pairs dataset and additional predicted value scores by the GP surrogate?
4. Is the GP model with Hamming distance kernel always a good enough surrogate? For example, what is the mean squared error for the value function/GP surrogate when the noise in the feature/output/dataset is increased for the feature importance/data valuation tasks?
5. App C.1.2 suggests that quasi-noiseless GPs are always used and the noise is not a learned hyper parameter. Are there other reasons beyond "consistent by construction"? Is overfitting/small resulting length scale a possible problem?

**Limitations:**

The conclusion mentions that future work should focus on computational improvements.

**Strengths And Weaknesses:**

## Strengths
* The paper is generally clearly written with clear notations. Sufficient details are provided in the main paper and the appendix. The background and related works are rather comprehensive.
*  The proposed Shapley value estimation method is novelly adaptive and applicable to many different games. The originality of the method lies in the application rather than the technical approach.
* Empirically, the method results in better Shapley value estimates with lower mean squared error.

## Weaknesses
* The method is still somewhat computationally expensive and is hence only evaluated on games with moderate number of players (<20). This may limit the significance of the work. The challenge associated with more players (e.g., computation cost, poorer approximation due to curse of dimensionality) should be discussed.
* The quality of the GP surrogate (mean and uncertainty quantification) can be further investigated. It is an important intermediate step affecting the next coalition selected and the quality of the Shapley value estimate. Intuitively, the estimate may be worse when the value function is erratic/noisy and at the start of training with few coalition-evaluations pairs.

---

> ### Author Rebuttal · Authors · 2026-03-31
>
> We gratefully thank the reviewer for their time and effort spent on reviewing our manuscript.
>
> ### **Q1 & Q2: Scalability & Computational Cost**
>
> ShaplEIG scales well beyond p>=20. Theoretically this was already shown in the paper, the experiments have now been extended, too (see response to Q2 for reviewer ogzU). We now answer your multiple questions clearly point-by-point. Apologies that for space constraints we have to refer to previous answers.
>
>  - The EIG computation for a single eval scales O(t^3 + p^3 + t^2* p). This is a major theoretical result of the paper (Theorem 3.1 and its proof).
>  - The evaluation of multiple / c candidates – as needed in EIG maximization – scales as O(t^3 + p^3 + t^2* c+ t^2* p). This is achievable via careful precomputation of all possible parts. This was shown in Appendix B.2.2. We agree that this should likely be moved to Theorem 3.1 in the main paper.
>
>  - Regarding extended practical experiments with p >= 20 and a discussion on the optimization of EIG over larger spaces see our response to Q2 for reviewer ogzU — also very much related to the above.
>  - In response of Q2 for reviewer yzSA we also benchmark the overhead of ShaplEIG and discuss the practical operating regime.
>
> - Regarding the complexity of the baselines, it is generally much lower than that of our method. In newly presented runtime benchmarks (https://anonymous.4open.science/r/shapleig_rebuttal-7C47/scalability_benchmark.md), the computational overhead of the strongest baseline (RegressionMSR) typically remains in the sub-second to seconds range, even for larger games. However, we again want to emphasize that in costly regimes, the induced overhead of ShaplEIG is worthwhile, as we obtain substantially smaller estimation errors with the same number of evaluations.
> - Lastly, regarding the concern about the curse of dimensionality (CoD) mentioned in the weaknesses: In our additional larger-scale experiments (Q2 of Reviewer ogzU above), we did not observe a degradation in estimation quality that would suggest a CoD effect within the tested regime.
>
> ### **Q3: GP-based ablation baselines**
>
> The GP-based ablation baselines all use the same GP surrogate architecture and the same type of initial design. They differ only in how additional coalitions are selected for evaluation during the sequential procedure. Specifically, “GP + Random” selects new coalitions uniformly at random, whereas “GP + Leverage Score Sampling” selects them according to leverage-scores that are analytically derived from the weighted least squares optimization in KernelSHAP (Musco & Witter, 2025). In both cases, the coalition-evaluation pairs are then used to fit the GP surrogate, and the Shapley values are extracted from the resulting GP posterior in exactly the same way as in ShaplEIG.
>
> Thus, the difference between these ablations lies entirely in the coalition-selection strategy, which in turn leads to different training datasets, different learned GP hyperparameters, and consequently different SV estimates. This clearly demonstrates that the adaptive sampling mechanism we introduce in our submission is the driver of the increased estimation quality.
>
> ### **Q4 & Q5: Surrogate quality in noisy settings**
>
> Regarding the overall quality of the GP surrogate with Hamming kernel, the strong SV-estimation performance of ShaplEIG already suggests that it is sufficiently expressive.
>
> Regarding surrogate quality in noisy settings, the quasi-noiseless GP was primarily used because the benchmarked games are deterministic, thus making them a natural choice. This choice also yields the consistency property of the resulting SV estimator. Also, in the experiments we used various real-world, and thus potentially noisy, tasks, and our proposed surrogate nevertheless showed strong performance and did not indicate signs of severe overfitting. However, in genuinely noisy settings, we agree that a noisy GP would often be more appropriate, with lower risk of overfitting. At the same time, the consistency property would then be lost, and in such settings it would generally be more principled to consider Shapley values as random variables induced by random games (Chau et al., 2023). We did not consider the latter here because it goes beyond the standard benchmark setting and would substantially broaden the scope. That said, we agree this is an important consideration for future work, and we will make this limitation more explicit in a revised version.
>
> For a broader discussion of GP surrogate quality beyond noisy settings, please refer to our response to Q1 for Reviewer ogzU, where we investigate the sensitivity of the GP surrogate to the initial design size and thus also cover its performance in the low-data regime. Furthermore, in our response to Q1 for Reviewer XdDP, we discuss the overall suitability of the Hamming kernel and extensions. Lastly, in our response to Q3 for Reviewer yzSA, we investigate the overall quality of the GP uncertainty quantification.

---

> > ### Author Rebuttal · Reviewer_2sbH · 2026-04-03
> >
> > My Q1-3 concerns have been addressed.
> > I still have concerns regarding the quality of the GP uncertainty quantification.
> >
> > The link shows that the NLPD decreases with more evaluations but the value themselves are not easy to interpret. Would it be useful to report the mean standardised log loss instead (the loss is standardised by subtracting the loss that would be obtained by predicting using the Gaussian mean and variance of the dataset)? The values should be negative for better methods. Are there evidence that the predicted coalition values are mostly within 3 standard deviation of the mean?

---

> > > ### Author Response · Authors · 2026-04-04
> > >
> > > We thank the reviewer for their follow-up. General response before we comply with your concrete requests:
> > >
> > > We note that in our framework the surrogate model serves as a means to guide the selection of coalitions. Thus, what matters for the method's effectiveness is the quality of the resulting Shapley value estimates, for which we provide extensive evidence in our paper and even more in the rebuttals here. This principle is well-established in Bayesian optimization, where it is known both theoretically [Berkenkamp et al., 2019] and empirically [Hutter et al., 2011] that surrogate model misspecification does not preclude good optimization performance, as long as the model provides useful guidance for the sequential design.
> > > In [Berkenkamp et al., 2019] even when the GP model is progressively wrong (they shrink lengthscales over time), convergence to the global optimum is guaranteed. In SMAC (and all of the many follow-up papers) papers, random forests are very successfully used and the "imperfect" modelling of the (epistemic) uncertainty of it is notoriously known. Would you agree with the above?
> > >
> > > ### References:
> > >  - Hutter, F., Hoos, H. H., & Leyton-Brown, K. (2011, January). Sequential model-based optimization for general algorithm configuration. In International conference on learning and intelligent optimization (pp. 507-523). Berlin, Heidelberg: Springer Berlin Heidelberg.
> > >  - Berkenkamp et al., 2019: No-Regret Bayesian Optimization with Unknown Hyperparameters. (https://arxiv.org/pdf/1901.03357)
> > >
> > > ---
> > >
> > > Nevertheless, we find your question quite interesting so here are the results:
> > >
> > > We expanded the additional analyses of the uncertainty estimates presented in Q3 of our rebuttal to reviewer yzSA (https://anonymous.4open.science/r/shapleig_rebuttal-7C47/uncertainty_estimation.md) to include both MSLL and calibration at the coalition-value level. More specifically, for each metric and across 128 iterations, we computed the metrics using the set of candidate coalitions that had not yet been evaluated at the respective iteration, treating them as an unseen test set. For MSLL, we used the archive of previously evaluated coalitions (i.e., the GP training data) to define the standardization baseline at each iteration.
> > >
> > > ## **Mean Standardized Log Loss**
> > > We present the MSLL results (mean and standard error of the mean computed over 100 seeds) across iterations in the following table:
> > >
> > > | Task | Blackbox | Seeds |Iteration: 1 | 16 | 64 | 128 |
> > > | --- | --- | --- | --- | --- | --- | --- |
> > > | TabPFN FI | Diabreg | 100 | -0.44 ± 0.02 | -1.08 ± 0.03 | -1.29 ± 0.05 | -1.84 ± 0.02 |
> > > | HyperSHAP | XGBoost (Chess) | 100 | -0.11 ± 0.09 | -0.77 ± 0.09 | -1.58 ± 0.08 | -2.08 ± 0.06 |
> > > | HyperSHAP | LCBench (Jasmine) | 100 | -0.03 ± 0.08 | -1.23 ± 0.09 | -2.64 ± 0.08 | -3.41 ± 0.11 |
> > >
> > > Across all tasks, the MSLL values are consistently below zero and decrease as the number of evaluations increases, indicating that the GP provides increasingly accurate predictive distributions.
> > >
> > > ## **Calibration**
> > > Furthermore, we evaluate the calibration of the GP surrogate via empirical coverage of predictive intervals at nominal confidence levels $c \in \{ 0.955, 0.997 \}$, corresponding to 2 and 3 standard deviations under the Gaussian predictive distribution. For each level $c$, we report the fraction of true coalition values that fall within the corresponding predictive interval, also indicating mean and standard error of the mean over 100 seeds:
> > >
> > > | Task | Blackbox | Seeds | Nominal level c | Iteration: 1 | 16 | 64 | 128 |
> > > | --- | --- | --- | --- | --- | --- | --- | --- |
> > > | TabPFN FI | Diabreg | 100 | 0.955 | 0.95 ± 0.00 | 0.95 ± 0.00 | 0.90 ± 0.00 | 0.93 ± 0.00 |
> > > | TabPFN FI | Diabreg | 100 | 0.997 | 0.99 ± 0.00 | 0.98 ± 0.00 | 0.96 ± 0.00 | 0.97 ± 0.00 |
> > > | HyperSHAP | XGBoost (Chess) | 100 | 0.955 | 0.85 ± 0.01 | 0.89 ± 0.01 | 0.91 ± 0.00 | 0.92 ± 0.00 |
> > > | HyperSHAP | XGBoost (Chess) | 100 | 0.997 | 0.94 ± 0.01 | 0.95 ± 0.00 | 0.97 ± 0.00 | 0.97 ± 0.00 |
> > > | HyperSHAP | LCBench (Jasmine) | 100 | 0.955 | 0.84 ± 0.01 | 0.90 ± 0.01 | 0.91 ± 0.01 | 0.90 ± 0.01 |
> > > | HyperSHAP | LCBench (Jasmine) | 100 | 0.997 | 0.94 ± 0.01 | 0.96 ± 0.00 | 0.97 ± 0.00 | 0.96 ± 0.00 |
> > >
> > > Across tasks, we observe different calibration dynamics over the course of the sequential design:
> > >  - For the TabPFN task, the empirical coverage initially matches the nominal level very closely, while in later iterations it slightly decreases below the nominal level, indicating mild overconfidence.
> > >  - In contrast, for the HyperSHAP tasks, the model is initially overconfident, but the calibration improves steadily as more data is collected, with empirical coverage approaching the nominal levels in later iterations.
> > >
> > > Overall, these results indicate that the uncertainty estimates are consistently well-behaved: while mild overconfidence can occur in certain phases, the uncertainty estimates remain consistently reasonable and close to the nominal levels.

---

### Official Review · Reviewer_XdDP · 2026-03-12

**Soundness:** 4
**Presentation:** 3
**Significance:** 4
**Originality:** 3
**Overall Recommendation:** 5
**Confidence:** 4

**Summary:**

This paper introduces a novel procedure to estimate Shapley values when evaluating the value function is expensive, based on the selection of coalitions with an adaptive design of experiments. This adaptivity is achieved via the training of a Gaussian process (GP) surrogate on observations of coalition-value function pairs, and an acquisition function chosen as the expected information gain (EIG). To avoid a very expensive computational cost due to the coalitions cardinality, they exploit the specific structure of the kernel they propose to reach a linear complexity with respect to the dimension instead of an exponential one. They illustrate the performance of their adaptive strategy for several tasks of Shapley value estimation and show faster convergence than competitors.

**Compliance With Llm Reviewing Policy:**

Affirmed.

**Final Justification:**

As I mentioned in my review, the paper is original since I was not aware of previous work on sequential design with GPs in the context of Shapley value estimation. The numerical illustrations are very convincing, and the authors replied convinclgy to my questions, as well as questions from other reviewers related to uncertainty quantification.

**Key Questions For Authors:**

I mainly have 3 questions for the authors:
- The Hamming kernel is not very expressive in the context of GPs with categorical features in general, and is not state-of-the-art currently in this setting. Although it works well in the experiments and enables access to huge computational savings, did the authors envision other kernels?
- I think an additional baseline for the experiments would consist in comparing their adaptive procedure to the recent fixed one proposed by Yang et al., which consists of an optimized design of experiments for coalitions. Did the authors evaluate this strategy? In addition, the ideas in this paper may give hints about new kernels.
- Finally, for tasks where the function value is a conditional expectation, an alternative procedure is to build a surrogate of the conditional expectation with all features, and use it to approximate all value functions (i.e. conditional expectation given the features in a coalition). When using a random forest surrogate, a selection strategy of the coalitions based on importance sampling was proposed by Bénard et al., and was shown to perform efficiently. Could this type of alternatives be considered as competitors in experiments?




References
- Liuqing Yang, Yongdao Zhou, Haoda Fu, Min-Qian Liu & Wei Zheng
(2024) Fast Approximation of the Shapley Values Based on Order-of-Addition Experimental
Designs, Journal of the American Statistical Association, 119:547, 2294-2304.
- Bénard, C., Biau, G., Da Veiga, S., & Scornet, E. (2022, May). SHAFF: Fast and consistent SHApley eFfect estimates via random Forests. In International conference on artificial intelligence and statistics (pp. 5563-5582). PMLR.

**Limitations:**

I was unable to find somewhere in the paper or the Appendix a discussion of the potential limitations of the work, although further improvements are mentioned as future work in the conclusion.

**Strengths And Weaknesses:**

The paper is clearly written, easy to follow, and the main ideas are introduced in a very pedagogical way. Importantly, I really like the author's proposal, which smartly derives an efficient acquisition function for selecting the coalitions, which is a very significant contribution for Shapley value estimation. Their derivation is technically sound (although heavy, as they mention themselves), and their claims are very well supported with a large variety of tasks and consistent improved performance over state-of-the-art competitors. Although using adaptive design of experiments with GPs is not original per se, using this strategy in the context of Shapley value estimation is however novel and yields impressive results in a small sample size regime. In addition, I would like to point out that both ablation studies and reproducible code are available, which is a very good point to support the paper.

On the weakness side, I may only mention points related to my questions below: missing discussion on alternatives related to design of experiments for coalitions (not adaptive) and surrogates.

---

> ### Author Rebuttal · Authors · 2026-03-31
>
> We gratefully thank the reviewer for their time and effort spent and appreciate that the contribution is viewed as novel and significant.
>
> ### **Q1: More expressive kernels beyond Hamming**
>
> In our experiments, the strong empirical performance of ShaplEIG suggests that the GP surrogate with the Hamming kernel is sufficiently expressive. Thus, we have not yet explored alternative kernels. Furthermore, we are not aware of other kernel choices that retain the computational advantages of the Hamming kernel, which are crucial for the efficient EIG computation presented in Theorem 3.1.
>
> We also note that recent work of Doumont et al. (NeurIPS 2025; https://openreview.net/forum?id=hdT7UC7oG6) show that, under their definition of a Hamming kernel (Def. 6), several recent kernels for categorical spaces are equivalent (and only differ in their hyperparameters): COMBO, CASMOPOLITAN and graph kernels. Our kernel (Equ. 13, Appendix A.1) is a Hamming kernel in their definition with $k_l(d) = exp(\frac{d^2}{\sum^p_{i=1}l^2_i})exp(\frac{-1}{\sum^p_{i=1}l^2_i})$. This also implies that the BODi and CoCABO kernels are different. So there are limited alternatives we can currently test, but we should consider the CASMOPOLITAN trust-region approach used by Doumont et al., which could also help choose coalition subsets in EIG point selection (c.f. answer to Reviewer ogzU). We aim to add this in the final version. We would also appreciate suggestions for more expressive kernels that preserve these computational advantages.
>
>
>
> ### **Q2: Comparison with Yang et al.**
>
> We thank the reviewer for pointing out the recent work of Yang et al. (2024), which is indeed related, as it uses a principled design-of-experiments perspective to improve Shapley estimation. At the same time, it differs from our setting in an important way: Yang et al. propose a **non-adaptive** design for permutation-based Shapley estimation, whereas our method performs adaptive coalition selection under a GP surrogate.
>
> We agree that this is a relevant non-adaptive baseline, and we will position this connection more explicitly in related work. However, we could not identify an open-source implementation and therefore could not include it in additional post-submission experiments. Permutation sampling (Castro et al., 2009), the method Yang et al. improve upon, is already included as a baseline in our experiments and is consistently outperformed by all competitors. Moreover, the permutation formulation is inherently costly: each sampled permutation requires $p$ sequential value function evaluations to yield one marginal contribution per player, whereas our method selects individual coalitions and reuses each evaluation for estimating all Shapley values jointly through the GP surrogate.
>
> Regarding your suggestion about _new kernels_ inspired by the combinatorial structures in Yang et al., we find this a promising direction. Specifically, one could **design kernels over the coalition space that incorporate prior knowledge about the Shapley weight structure**, such as coalition-size-dependent correlations, rather than relying on the generic Hamming kernel. Doumont et al. (see answer to Q1) should also come in handy here as it discusses incorporating structure into Hamming kernels in Appendix D. We will discuss this concretely as future work in the revised manuscript.
>
> ### **Q3: Comparison with SHAFF**
>
> Yes. Again, thank you for pointing us to this work, which is also relevant. SHAFF exploits that, when the value function is a _conditional expectation_, all coalition values can be derived from a single pre-fitted model via projection. In our setting, however, the value functions are not conditional expectations: they involve model retraining, hyperparameter ablation, or baseline imputation. This is precisely the setting where a _direct_ value function surrogate, as in our GP-based approach becomes necessary. Akin to Yang et al., we also note that SHAFF's coalition sampling distribution seems to be fixed by the tree structure of the initial forest and does not adapt based on value function evaluations placing it also in the same non-adaptive category as LeverageSHAP. Nevertheless, imposing further assumptions on the value function such as in the conditional-expectation setting is a natural and interesting extension of our work. Yet, in our opinion, it would potentially be an even more interesting direction for future work to make other surrogate-based methods, such as random-forests here, adaptive as well, for example by updating the coalition-sampling distribution online based on observed coalition values. We will add this discussion to the revised manuscript.
>
> ### **Limitations**
>
> We mainly discussed limitations more implicitly with respect to future work. We will do this more prominently. We think the major points are discussed in Response Q2 to ogzU.

---

> > ### Author Rebuttal · Reviewer_XdDP · 2026-04-02
> >
> > Thank you for your detailed response to my questions, which are convincing. I will keep my initial positive score.

---

### Official Review · Reviewer_ogzU · 2026-03-13

**Soundness:** 3
**Presentation:** 3
**Significance:** 2
**Originality:** 4
**Overall Recommendation:** 4
**Confidence:** 5

**Summary:**

This paper proposes ShaplEIG, a novel method for estimating Shapley values when the value function evaluation is computationally expensive and the budget is limited. The coalition selection is framed as a Bayesian Experimental Design (BED) problem. By utilizing the linearity of Shapley values, this paper formulates the selection of coalitions in sampling as a Bayesian linear inverse problem of a Gaussian Process (GP) surrogate with a Hamming kernel. This paper presents a closed-form expression for the Expected Information Gain (EIG). To overcome the computational bottleneck of EIG maximization, this paper further proposes an efficient O(p · t²) computation algorithm utilizing the multiplicative structure of the Hamming kernel. Experiments show that ShaplEIG outperforms baselines such as RegressionMSR and KernelSHAP in targeted tasks.

**Compliance With Llm Reviewing Policy:**

Affirmed.

**Final Justification:**

The rebuttal addressed my main concerns, thus changing my evaluation.

**Key Questions For Authors:**

1.	The initial design relies on p + 1 coalitions drawn via leverage score sampling. How sensitive are the GP hyperparameters and the final SV estimation to this initial sampling?
2.	Is it possible to scale the method to slightly larger feature sets (e.g., p = 50) using heuristic or approximate EIG maximization algorithms? A brief discussion on this would be helpful.

**Limitations:**

The constraints of the method, especially the strict p ≤ 20 limit, are adequately discussed by the authors. This is acceptable given the scope of the paper.

**Strengths And Weaknesses:**

Strengths:
•	The main theoretical contribution of this paper is framing adaptive SV estimation as a Bayesian linear inverse problem. This paper avoids the computationally heavy EIG estimations common in BED by utilizing the linearity of Shapley values.
•	The naive time complexity of computing EIG is O(4^p · t). The paper lowers it to O(p · t^2) by utilizing the structure of the Hamming kernel and elementary symmetric polynomials. The proofs are solid and clear.
•	The experiments are specifically designed for the proposed method in targeted tasks. (e.g., TabPFN feature importance, data valuation). The results demonstrate consistent superiority of sample efficiency compared to baselines
Weaknesses:
•	The reliance on exact EIG computation restricts the method to small-scale games (p ≤ 20) with strict limits.
•	This paper is lacking comparisons with Shapley estimation methods (not specifically for SHAP) commonly found in the AI and database literature.
•	Lack of consistency in presentation. Providing a comprehensive algorithm (pseudocode) would be helpful.
•	The adaptive sampling process breaks the assumption of independent samples, which means ShaplElG may not be unbiased. But this paper provides consistency instead.

---

> ### Author Rebuttal · Authors · 2026-03-31
>
> Thanks for your detailed review. We will include all of the below in greater detail in a revised version.
>
> ### **Q2: Scalability & Practical Scope**
>
>
> We would like to clarify that ShaplEIG is not inherently limited to games with fewer than 20 players. This was already theoretically worked out in the original submission due to the runtime scaling analysis O (t^3 + p^3 + p*t^2) (see Theorem 3.1 and its proof). But: We improved the scaling substantially shortly before submission, and could not extend the experiments until then (and maybe not every sentence in the paper reflected better scaling already). This is rectified now, including also an improved implementation.
> We now include four additional tasks with up to 101 players. They show that ShaplEIG remains tractable in this regime and consistently outperform all competitors by a large margin (More detailed results are available under https://anonymous.4open.science/r/shapleig_rebuttal-7C47/large_scale_experiments.md):
>
> **Table 1**. Scaling ShaplEIG to larger player-counts $p$ measured as Mean MSE at 256 iters.
> | Dataset | p | ShaplEIG (ours) | Closest competitor |
> |-|-|-|-|
> | CommCrime | **101** | **1.0151e+01** | 4.0460e+01  |
> | NHANESI | **79** | **6.2570e-03** | 2.2288e-02 |
> | CorrGroups | **60** | **2.926e-04** | 9.183e-04 |
> | IndepLin | **60** | **1.519e-04** | 9.422e-04 |
>
> While the EIG computation is reasonably fast, it also needs to be maximized across a space of 2^p size.
> We have implemented a very simple stochastic search here (biased sampling w.r.t. leverage scores), and note that this already enables the good results above. But we think that this can potentially be even more improved by using a more complex genetic algorithm on the binary space.
> We also found that the overhead can be reduced substantially while maintaining high performance by re-fitting GP hyperparameters only every k iterations and reusing constant EIG components in between. This suggests that techniques for approximate EIG optimization show great potential for further extending the method to even larger games or to regimes with less expensive value functions.
>
> Also see our response of Q2 to yzSA, where we benchmark the computational cost of ShaplEIG.
>
>
> ### **Q1: Sensitivity to Initial Design**
>
> Our choice of a size of p+1 follows common practice in GP-based BED and BO. It is also a natural lower-end choice in our experiments, since several linear-model-based competitors (KernelSHAP or LeverageSHAP) require at least p+1 observations. Regarding the sampling scheme for the initial design, we used the state-of-the-art leverage-score sampling (Musco & Witter, 2025) because it is a strong strategy in recent benchmarks, while adding basically no overhead.
>
> The intuition here is to have the initial design as small as possible, to have all remaining design points subject to guided sequential selection – while ensuring that the initial surrogates are not completely broken. Please note that this trade-off exists for *any* BO method from the last 20+ years and performance is usually not influenced too much by it.
> Nevertheless, as the reviewer asked, we also now assessed its sensitivity in additional benchmarks varying both the initial design size and the initialization strategy. These results show that larger initial designs can slightly improve final performance, while random initialization performs only marginally worse than leverage-score initialization for the considered task (this is in line with BO benchmarks, where the influence of the init design is usually only marginal).
> In any case, ShaplEIG, dominates all baseline.
> We provide more detailed results under https://anonymous.4open.science/r/shapleig_rebuttal-7C47/initial_design.md.
>
> ### **Weaknesses:**
>
> - **Regarding lacking comparisons**:  We focused on baselines that are strong, commonly used and directly relevant to our setting; this is in line with recent benchmarks / standard implementations such as the shapiq (Muschalik et al., 2024) and SHAP packages (Lundberg and Lee, 2017). Unfortunately, the reviewer does not explicitly name a missing baseline from the “AI and database literature.” Therefore, we think that we provide thorough evidence of the performance of ShaplEIG. However, if anything relevant is concretely mentioned, we would be happy to include it for the camera ready copy.
> - **Regarding pseudocode**: We agree and will add one.
> - **Regarding unbiasedness**: We agree with this point and note that the paper does not claim unbiasedness. For adaptive surrogate-based estimators, unbiasedness is typically not the central objective. More generally, in Shapley-value estimation it is common to accept some bias in exchange for lower variance (as in KernelSHAP, see the analyses of Covert & Lee (2021) and Kolpazki et al. (2024) for further details), to achieve a reduced MSE of the estimator. We do exactly the same, and our results show that this works. We have also shown consistency of our estimator.

---

> > ### Author Rebuttal · Reviewer_ogzU · 2026-04-03
> >
> > Thanks for your responses. Here are two specific papers of SOTA Shapley value computation methods in AI and database communities.
> > "Faster Approximation of Probabilistic and Distributional Values via Least Squares."
> > "Efficient Sampling Approaches to ShapleyValue Approximation."
> > Besides, I'd like to raise the score to 4.

---

### Official Review · Reviewer_yzSA · 2026-03-13

**Soundness:** 3
**Presentation:** 3
**Significance:** 3
**Originality:** 3
**Overall Recommendation:** 3
**Confidence:** 3

**Summary:**

The paper studies Shapley value estimation from the perspective of Bayesian experimental design in settings where coalition evaluations are expensive. It proposes ShaplEIG, a method that models the coalition value function with a GP surrogate and sequentially selects coalitions by maximizing the expected information gain about the Shapley values. Using the fact that the Shapley vector is a linear transformation of the coalition-value vector, the paper derives a closed-form EIG objective and an efficient computation scheme based on the Hamming-kernel structure. Experiments on several costly, moderately sized games show improved sample efficiency over a range of baselines.

**Compliance With Llm Reviewing Policy:**

Affirmed.

**Final Justification:**

The overall framework is reasonable, and the rebuttal clarified several aspects of the paper while adding useful evidence on scalability and posterior uncertainty. I continue to think the paper has clear strengths.

However, my main concern remains only partially addressed. The additional examples make the covariance-based intuition clearer, but I still do not think the paper precisely characterizes the settings or assumptions under which this advantage should be expected. For example, if singleton-level effects are similar while only specific interaction effects are strong, it is unclear whether adaptive selection would still offer a substantial advantage. A more explicit characterization of the regimes in which the proposed method is expected to be most effective or be failed would strengthen the paper. As a result, I remain unconvinced overall and keep my final recommendation as weak reject.

**Key Questions For Authors:**

1. Can the authors more clearly explain how the covariance-based EIG criterion leads to meaningfully different coalition rankings in practice, and in what regimes this strategy is expected to be advantageous? While the formulation is mathematically clear, the practical mechanism by which posterior covariance drives sequential coalition selection remains difficult to assess from the current presentation.

2. How should readers interpret the practical scope of the current method beyond the moderate-size setting studied in the paper? A clearer discussion of the intended operating regime, and of where further scalability advances are still necessary, would help sharpen the paper’s practical claims

3. Can the authors provide more direct evidence on the usefulness of the posterior uncertainty over Shapley values? In particular, it would strengthen the paper to show whether these uncertainty estimates are well calibrated and practically useful for decision-making or stopping criteria, beyond improvements in final MSE.

**Limitations:**

yes

**Strengths And Weaknesses:**

Strengths

1. The paper proposes a clean and original reframing of Shapley estimation as a goal-oriented Bayesian experimental design problem, leveraging the fact that Shapley values are a linear transformation of the coalition-value vector. This leads to a principled EIG objective directly targeted at the final quantity of interest.

2. The paper derives a closed-form EIG objective and develops a tractable computation strategy under the GP surrogate, which makes the approach practically meaningful.

3. The work provides a promising methodological foundation for extending adaptive Shapley estimation to potentially larger player settings in the future.

Weaknesses

1. The practical role of the covariance-based EIG criterion remains insufficiently clear. While the formulation is mathematically well motivated, the paper does not yet provide enough intuition for how the posterior covariance structure leads to meaningfully different sequential coalition selections in practice. A small toy example or a more explicit explanation of when and why this criterion is advantageous would substantially improve the paper.

2. The empirical scope remains limited to moderately sized games. The experiments explicitly focus on $p\leq20$, and the paper itself identifies scalable GP variants, approximate EIG optimization, and batched design as future work. As a result, while the method is interesting within its intended regime, its current scope is still limited relative to broader large-scale SHAP estimation settings.

3. The Bayesian uncertainty-quantification aspect is not validated as directly as the paper’s framing suggests. The paper argues that the GP surrogate provides posterior uncertainty over Shapley values that could support decision-making or stopping criteria, but the empirical evaluation focuses primarily on final MSE and does not directly assess the calibration or practical utility of these uncertainty estimates.

---

> ### Author Rebuttal · Authors · 2026-03-31
>
> We thank the reviewer for their time and engagement with our work and particularly the appreciation of the practical relevance.
>
> ### **Q2: Scalability & Intended operating regime**
>
> Please see our response of Q2 to Reviewer ogzU, as this is closely related. There we present new experiments showing that the method can already be scaled to around 100 players and thus demonstrating a much broader scope of ShaplEIG. There we also discuss scaling techniques and remaining challenges.
>
> Regarding the intended operating regime: Although the paper already includes a computational-cost analysis, we agree that the practical operating regime of ShaplEIG may not yet be sufficiently transparent. To address this, we conducted an additional runtime benchmark analyzing how the overhead scales with the number of players and sequential iterations. It shows that for games with up to roughly 32 players, the overhead per iteration remains in the range of seconds even after 100 iterations, while for games with around 100 players it increases to the range of minutes. We believe this helps clarify the intended application regime: for smaller and moderate-sized games, ShaplEIG is already practical even when value-function evaluations are relatively cheap (seconds), whereas for larger games it is only appropriate when evaluations are actually costly (minutes to hours). The benchmark results are provided under https://anonymous.4open.science/r/shapleig_rebuttal-7C47/scalability_benchmark.md and will be featured in a revised version.
>
> (For the theoretical analysis, please see the beginning of our response to Q1 of 2sbH ).
>
> ### **Q1: Covariance-based EIG Intuition**
>
> The key practical role of the covariance-based EIG criterion is that ShaplEIG does not select the coalition whose value is merely most uncertain (as in uncertainty sampling / US), but whose evaluation is expected to reduce uncertainty most strongly about the final Shapley values (SVs) by informing about many other coalitions.
>
> This is exactly where covariance matters. Under the GP surrogate, evaluating one coalition does not only reduce uncertainty locally at that coalition itself; through the posterior covariance structure, it also reduces uncertainty globally at others. Two candidate coalitions may have similar marginal posterior variance, but very different covariance structure with the remaining coalitions. In that case, uncertainty sampling would treat them similarly, whereas EIG would overcome this.
>
> In our setting, this effect is induced by the Hamming kernel. It measures similarity between coalitions based on which players are the same, and the learned lengthscales determine how strongly specific player disagreements reduce covariance. As a result, two coalition pairs with the same Hamming distance can still have very different covariance if they differ in players that are more or less influential under the surrogate, leading to different coalition rankings in practice.
>
> This also clarifies the regime in which we expect the strategy to be advantageous: settings in which different players have different influence on the value function, so that some coalition differences are more informative than others. In such cases, treating all coalition differences equally, or relying only on coalition size as in non-adaptive strategies (i.e., the current state-of-the-art methods in Shapley estimation), is suboptimal. We believe this regime is common in practice, which is consistent with the strong performance we observe across tasks.
>
> We agree that this intuition could be made clearer in the paper, and we will include it in a revised version.
>
> ### **Q3: Utility of uncertainty estimates**
>
> Although the acquisition function in ShaplEIG is based entirely on the GP posterior covariance and already demonstrates strong performance for adaptive coalition selection, we agree that the quality and practical utility of the resulting uncertainty estimates are not analyzed directly in the current version.
>
> To address this, we conducted two additional analyses. First, for a subset of the original experiments, we tracked the mean negative log predictive density (NLPD) of the SV posterior over iterations. NLPD is a standard probabilistic metric that evaluates not only the quality of the posterior mean, but also of the predictive covariance. We observe that NLPD decreases consistently over time, indicating that the posterior over SVs is improving. Second, we investigated whether the uncertainty can serve as a practical stopping criterion. Concretely, across seeds and iterations, we compared the true SV-estimation error, measured by MSE, to the (average) marginal posterior variance of the SVs. We find Pearson correlations in the range of 0.69-0.8, showing a clear positive relationship. This suggests that the posterior variance is useful for stopping decisions. The full analyses can be found under https://anonymous.4open.science/r/shapleig_rebuttal-7C47/uncertainty_estimation.md.

---

> > ### Author Rebuttal · Reviewer_yzSA · 2026-04-04
> >
> > I appreciate the authors for providing useful additional experimental results that further support the effectiveness of the proposed framework. The rebuttal also makes the basic mechanism of ShaplEIG clearer.
> >
> > However, I would still have liked a clearer characterization of the qualitative regimes in which the covariance-based EIG criterion is expected to be most beneficial and a more concrete illustration of how uncertainty over specific coalition structures is propagated and reduced under the GP posterior. Could the authors clarify this point with a specific function or case study?

---

> > > ### Author Response · Authors · 2026-04-06
> > >
> > > Thank you for your response. To make this more illustrative, we present two concrete examples showing when EIG-based coalition selection is beneficial. In both cases, we consider games with three players ($\\mathcal{P}= \\{1,2,3\\}$) and the initial design $D_0= \\{(\\emptyset, y_0), (\\{1\\}, y_{1}), (\\{2\\}, y_{2}), (\\{3\\}, y_{3}), (\\{1,2,3\\}, y_{1,2,3}) \\}$, containing the empty and full coalition and all size-one coalitions. As candidate set, we consider all size-two coalitions, i.e., $\\mathcal{X}_{\\text{cand}}= \\{\\{1,2\\}, \\{1,3\\}, \\{2,3\\}\\}$. We implemented these examples and we report the true values of intermediate quantities.
> > >
> > > ### **Symmetric Games**
> > > First, consider the symmetric value function $\\nu_{\\text{sym}}(\\mathcal{S})= \\lvert \\mathcal{S} \\rvert^2$, which contains interactions among players but is identical across all coalitions of the same size. The initial design evaluates to $\\mathcal{D}_{0, \\text{sym}}= \\{(\\emptyset, 0), (\\{1\\}, 1), (\\{2\\}, 1), (\\{3\\}, 1), (\\{1,2,3\\}, 9) \\}$.
> > >
> > > Since all single-player coalitions yield the same value, GP fitting in the initial ShaplEIG iteration produces identical lengthscales for all players, i.e., $\\ell= (0.345,0.345,0.345)^{\\top}$. As a result, the covariances among all unseen candidates are identical: $k_{\\xi}(\\{1,2\\}, \\{1,3\\})= k_{\\xi}(\\{1,2\\}, \\{2,3\\})= k_{\\xi}(\\{1,3\\}, \\{2,3\\})= 0.120$. Each pair of candidate coalitions shares exactly one player, and due to identical lengthscales, differences in player membership do not affect the covariance differently across players. Consequently, the EIG is identical across all candidates, i.e., $\\text{EIG}(\\{1,2\\})=\\text{EIG}(\\{1,3\\})=\\text{EIG}(\\{2,3\\})= 1.391$.
> > >
> > > Thus, in this setting, EIG provides no advantage over coalition sampling methods that depend solely on coalition size and are commonly used in classical Shapley value estimation.
> > >
> > > ### **Asymmetric games**
> > > We now consider an asymmetric game with value function $\\nu_{\\text{asym}}(\\mathcal{S})= 1 \\cdot [1 \\in \\mathcal{S}]+1 \\cdot [2 \\in \\mathcal{S}]+0.01 \\cdot [3 \\in \\mathcal{S}]+1 \\cdot [\\{1,2\\} \\subseteq \\mathcal{S}]$. Players 1 and 2 have identical main effects, player 3 is negligible, and there is an interaction between players 1 and 2.
> > >
> > > Given the initial design $D_{0, \\text{asym}}= \\{(\\emptyset, 0), (\\{1\\}, 1), (\\{2\\}, 1), (\\{3\\}, 0.01), (\\{1,2,3\\}, 3.01) \\}$, the fitted GP hyperparameters $\\ell= (0.269,0.269,1.306)^{\\top}$ already reflect the low relevance of player 3. Consequently, the coalition containing the two relevant players, $\\{1,2 \\}$, exhibits higher covariance with the other candidates than the other size-2 coalitions:
> > > $k_{\\xi}(\\{1,2\\}, \\{1,3\\})= k_{\\xi}(\\{1,2\\}, \\{2,3\\})= 0.162 > k_{\\xi}(\\{1,3\\}, \\{2,3\\})= 0.060$. This arises because players 1 and 2 have smaller lengthscales, so sharing one of them reduces variability more. Coalitions sharing only player 3 leave the relevant players differing and therefore have lower covariance. This is reflected in the quadratic form $\\mathbf{e}_i^{\\top} \\mathbf{Q} \\mathbf{e}_i$ (cf. Eq. 9), which depends on these covariances, and as a result the EIG is maximized for this coalition: $\\text{EIG}(\\{1,2\\})= 1.50>\\text{EIG}(\\{1,3\\})=\\text{EIG}(\\{2,3\\})= 1.455$. Intuitively, this candidate, due to its highest correlation with the remaining coalitions, is expected to yield the largest reduction in uncertainty about the Shapley values. Consequently, ShaplEIG selects $\\{1,2 \\}$ for evaluation, revealing the key interaction. This reduces the MSE of the Shapley value estimation from 0.024 to 5.485e-06. Correspondingly, the posterior variance of the GP at $\\{1,2 \\}$, initially $0.402$, is eliminated, while for the remaining coalitions $\\{1,3\\}$ and $\\{2,3\\}$ it is reduced from 0.3841 to 0.0677.
> > >
> > > In contrast, classical size-based coalition samplers choose all three candidates with equal probability. As a result, it is more likely that either $\\{1,3 \\}$ or $\\{2,3 \\}$ is selected, neither of which reveals the relevant interaction and therefore leads to a substantially smaller reduction in MSE (from 0.024 to 8.75e-4).
> > >
> > > ### **Summary**
> > > This example demonstrates that in asymmetric games, where some players are highly relevant to the value function while others have negligible effect, and interactions occur primarily among the relevant players, ShaplEIG can exploit this structure to identify informative coalitions. In contrast, classical coalition sampling schemes often fail to do so, as they treat all players identically. We believe this setting is common, particularly in large games, where it is unlikely that all players contribute symmetrically to the value function. We hope this answer, together with our answers on the scalability, intended operating regime, as well as the utility of uncertainties, convinces you that our paper is a valuable contribution that should be accepted at ICML.

---

### Decision · Program_Chairs · 2026-04-30

**Decision:**

Accept (regular)

**Comment:**

This paper proposes the use of Bayesian experimental design (BED) to sequentially choose coalitions when estimating Shapley values.

Reviews generally praised its clear and well-motivated problem, strong conceptual contribution, methodological soundess, empirical performance, and the clear presentation and writing of the paper.  Concerns were raised about scalability limitations, computational overhead, intuition for why and when the method provides benefits, dependence on the GP surrogate, and some potentially missing baselines.  Though many of the concerns were resolved during the rebuttals, some where still left after the rebuttals and discussion.  In particular, the one reviewer still arguing for rejection did so on the basis that they did not feel that "the paper precisely characterizes the settings or assumptions under which this advantage should be expected", with other reviewers feeling that they did not have major concerns outstanding (though there definitely are some unavoidable limitations in terms of scaling and computational overhead).

I do not personally fully agree with the remain reviewer's concerns about when and why the method will be beneficial: I believe that the authors provide a solid explanation for this in their rebuttal and follow up.  Though I do think that a demonstrative example / some plots to qualitatively show the coalitions chosen would benefit the paper here and provide some more intuition, I think this is very fixable in the camera-ready version of the paper and not a reasonable basis for rejecting the paper.

All reviewers are in agreement with a number of important strengths of the work and having read it myself I also concur that it would be an interesting and solid contribution to ICML.  I thus confidently recommend that it is accepted into the program.

I further wanted to note a few points the authors may wish to address in the camera ready version of the paper
- As previously mentioned, I do think reviewer yzSA makes some valid points about providing more intuition for the benefits and would encourage the authors to both add in their discussion on this and try to also add some simple intuitive example that can show how the methods chooses coalitions qualitatively and why this is helpful.
- Related to the above, the EIG being used can be interpeted as an instance of an expected predictive information gain (or equivalently a transductive EIG) as it is explictly targeting information on mappings of the GPs predictions at some other input points.  This crucially explains why it is working differently to simple uncertainty sampling (noting the EIG of the GPs _parameters_ is actually equivalent to uncertainty sampling) and thus a critical part of why the methods works.  This should be explicitly discussed and relevant papers on predictive EIGs and transductive active learning cited accordingly, in particular, Prediction-oriented Bayesian active learning. Bickford Smith et al, AISTATS 2023
- The approach can be viewed as a special case of active testing (Active testing: Sample-efficient model evaluation. Kossen et al. ICML 2021; Active Surrogate Estimators: An Active Learning Approach to Label-Efficient Model Evaluation; Kossen et al. NeurIPS 2022) because eq (1) can be viewed as an expectation of a "loss function" over S.  This should be explicitly acknowledged in the paper.  Moreover, it provides a mechanism to perform the required diabising the reviewers have commented on.  Namely, if one relaxed the selection of coalitions to be probabilistic (i.e. using the EIG values to formulate a proposal rather than greedily choosing the maximum), then the LURE estimator (Farquhar et al 2021) can be used apply corrective weights that yield an unbiased estimator.  I don't think it is necessary for the paper to actually use an unbiased estimator in this way, but it would be nice to note that it can be done if one desires this.
- Equation (4) has the potential to be a little misleading as written and I initially thought that it was incorrect.  The EIG is always invariant to invertible transformations of one of the variables, but the equation as written makes it look like it would decrease as the determinant of A increases.  This is actually not the case as the constant in x "C" turns out to depend on A in a way that cancels out this dependency on invertible mappings on A itself.  This would be good to make clearer to aid intuition for the reader.